# VPS39-deficiency observed in type 2 diabetes impairs muscle stem cell differentiation via altered autophagy and epigenetics

Cajsa Davegårdh [1,15], Johanna Säll [1,15], Anna Benrick[2,3,15], Christa Broholm[4], Petr Volkov[1], Alexander Perfilyev [1], Tora Ida Henriksen[5], Yanling Wu[2], Line Hjort[4,6], Charlotte Brøns[4], Ola Hansson [7,8], Maria Pedersen[5], Jens U. Würthner[9], Klaus Pfeffer[10], Emma Nilsson[1], Allan Vaag[11], Elisabet Stener-Victorin [12], Karolina Pircs[13], Camilla Scheele[5,14] & Charlotte Ling [1✉]

Insulin resistance and lower muscle quality (strength divided by mass) are hallmarks of type 2 diabetes (T2D). Here, we explore whether alterations in muscle stem cells (myoblasts) from individuals with T2D contribute to these phenotypes. We identify VPS39 as an important regulator of myoblast differentiation and muscle glucose uptake, and VPS39 is downregulated in myoblasts and myotubes from individuals with T2D. We discover a pathway connecting VPS39-deficiency in human myoblasts to impaired autophagy, abnormal epigenetic reprogramming, dysregulation of myogenic regulators, and perturbed differentiation. VPS39 knockdown in human myoblasts has profound effects on autophagic flux, insulin signaling, epigenetic enzymes, DNA methylation and expression of myogenic regulators, and gene sets related to the cell cycle, muscle structure and apoptosis. These data mimic what is observed in myoblasts from individuals with T2D. Furthermore, the muscle of $Vps39^{+/-}$ mice display reduced glucose uptake and altered expression of genes regulating autophagy, epigenetic programming, and myogenesis. Overall, VPS39-deficiency contributes to impaired muscle differentiation and reduced glucose uptake. VPS39 thereby offers a therapeutic target for T2D.

[1] Epigenetics and Diabetes Unit, Department of Clinical Sciences, Lund University Diabetes Centre, Lund University, Scania University Hospital, Malmö, Sweden. [2] Department of Physiology, Institute of Neuroscience and Physiology, Sahlgrenska Academy, University of Gothenburg, Gothenburg, Sweden. [3] School of Health Sciences, University of Skövde, Skövde, Sweden. [4] Diabetes and Bone-metabolic Research Unit, Department of Endocrinology, Rigshospitalet, Copenhagen, Denmark. [5] The Centre of Inflammation and Metabolism and the Centre for Physical Activity Research, Rigshospitalet, University of Copenhagen, Copenhagen, Denmark. [6] Department of Obstetrics, Rigshospitalet, Copenhagen, Denmark. [7] Genomics, Diabetes and Endocrinology Unit, Department of Clinical Sciences, Lund University, Malmö, Sweden. [8] Finnish Institute of Molecular Medicine, University of Helsinki, Helsinki, Finland. [9] ADC Therapeutics, Biopole, Epalinges, Switzerland. [10] Institute of Medical Microbiology and Hospital Hygiene, Heinrich Heine University Düsseldorf, Düsseldorf, Germany. [11] Steno Diabetes Center Copenhagen, Gentofte, Denmark. [12] Department of Physiology and Pharmacology, Karolinska Institutet, Stockholm, Sweden. [13] Laboratory of Molecular Neurogenetics, Department of Experimental Medical Science, Wallenberg Neuroscience Center and Lund Stem Cell Center, Lund University, Lund, Sweden. [14] Novo Nordisk Foundation Center for Basic Metabolic Research, Faculty of Health and Medical Sciences, University of Copenhagen, Copenhagen, Denmark. [15] These authors contributed equally: Cajsa Davegårdh, Johanna Säll, Anna Benrick. ✉email: charlotte.ling@med.lu.se

Aging populations and a sedentary lifestyle are leading to an increased prevalence of type 2 diabetes (T2D). T2D is characterized by chronic hyperglycemia caused by insulin resistance of target tissues and impaired insulin secretion. Skeletal muscle is the primary organ responsible for insulin-stimulated glucose uptake and T2D is associated with lower muscle strength and quality (strength divided by mass) contributing to glucose intolerance[1,2]. Individuals with T2D display impaired mitochondrial function, abnormal lipid deposition, and metabolic inflexibility in their muscle[3,4].

Skeletal muscle is regenerated and maintained by muscle stem cells (satellite cells) that are activated in response to, e.g., injury and exercise[5,6]. Activated muscle stem cells (myoblasts) then proliferate, and a subset starts to differentiate and fuse into new myotubes or existing myofibers in a process called myogenesis[7]. Myogenesis is influenced by several extracellular factors, including cytokines and hormones, and intrinsically controlled by myogenic regulatory factors (MRFs) (MYOD1, MYF5, MYOG, and MYF6) and myocyte enhancer factors 2 (MEF2) (MEF2A, MEF2C, and MEF2D)[8]. However, additional myogenic regulators that may be altered in individuals with T2D remain to be identified.

Although impaired myogenesis and muscle regeneration have been observed in rodent models of diabetes[3,9], it is not well established whether the abnormalities in the muscle of individuals with T2D exist already at stem cell and progenitor stages[10]. Nevertheless, muscle stem cells derived from individuals with T2D retain some diabetic phenotypes, e.g. impaired glucose uptake and lipid oxidation, after in vitro differentiation[11,12]. This implies cellular memory of the diabetic environment. However, it remains unknown whether epigenetic modifications, such as DNA methylation, in the muscle stem cells of individuals with T2D contribute to this phenotype.

Epigenetic modifications are important during development and regulate cell specificity, expression, and chromatin stability. Environmental factors and disease, including exercise, diet, aging, and T2D, influence DNA methylation in human muscle and other tissues[13–17]. We recently found abnormal epigenetic and transcriptional changes during the differentiation of myoblasts from individuals with obesity versus non-obese[18]. Nevertheless, the methylome of myoblasts from individuals with T2D has not been studied to date.

To identify previously unrecognized candidates that contribute to the abnormalities seen in muscle from patients with T2D, we analyzed the genome-wide expression and DNA methylation in primary human myoblasts and myotubes from individuals with T2D and controls. We identified VPS39 as one of the genes associated with T2D in human myoblasts and myotubes, and VPS39 was downregulated in these cells. Using gene silencing experiments in human myoblasts and a mouse model, we provide evidence that VPS39 is a previously unrecognized regulator of human myogenesis and muscle glucose uptake.

## Results

**Expression landscapes of myoblasts and myotubes from T2D individuals are distinct from controls, and identify VPS39 as a previously unrecognized candidate regulating myogenesis.** To dissect transcriptional and epigenetic differences between T2D and control muscle cells, we isolated human muscle stem cells (satellite cells) from vastus lateralis from 14 controls with normal glucose tolerance (NGT) and 14 individuals with T2D. Their clinical characteristics are described in Table 1. Individuals with T2D showed impaired glucose control based on HbA1c measurements, oral glucose tolerance tests (OGTT), homeostatic model assessment of insulin resistance (HOMA-IR), and beta-cell

**Table 1 Clinical characteristics of human donors of muscle cells.**

| | Controls | Type 2 diabetes | *p*-value |
|---|---|---|---|
| n | 14 | 14 | |
| Gender (male/female) | 7/7 | 7/7 | |
| Age | 54.2 ± 6.8 | 58.1 ± 6.6 | 0.14 |
| Years of disease | | 4.8 ± 4.0 | |
| Height (m) | 1.7 ± 0.1 | 1.7 ± 0.1 | 0.95 |
| Weight (kg) | 74.5 ± 14.2 | 80.3 ± 13.4 | 0.28 |
| BMI (kg/m²) | 24.7 ± 2.4 | 26.6 ± 3.1 | 0.07 |
| Hip circumference (cm) | 100.6 ± 4.7 | 100.6 ± 8.7 | 1.00 |
| Waist circumference (cm) | 85.9 ± 10.2 | 96.3 ± 8.7 | 0.01 |
| Waist/hip ratio | 0.9 ± 0.1 | 1.0 ± 0.1 | 0.01 |
| Body lean mass/BW (%) | 68.4 ± 8.7 | 64.4 ± 9.7 | 0.26 |
| Body fat mass/BW (%) | 27.4 ± 8.4 | 31.1 ± 9.1 | 0.27 |
| Android fat mass/BW (%) | 2.3 ± 0.7 | 3.2 ± 0.9 | 0.009 |
| Gynoid fat mass/BW (%) | 5.6 ± 2.2 | 5.3 ± 2.0 | 0.74 |
| Android/Gynoid fat ratio | 46.1 ± 18.0 | 65.3 ± 21.3 | 0.02 |
| HbA1c (%)[a] | 5.5 ± 0.2 | 6.7 ± 1.1 | 0.002 |
| Glucose 0 h (mmol/L) | 4.8 ± 0.6 | 9.1 ± 3.7 | 0.001 |
| Glucose 2 h (mmol/L)[b] | 5.2 ± 1.3 | 18.2 ± 6.6 | $4.7 \times 10^{-6}$ |
| Insulin 0 h (pmol/L) | 32.1 ± 13.3 | 64.1 ± 41.9 | 0.02 |
| Insulin 2 h (pmol/L)[b] | 230.1 ± 155.3 | 371.5 ± 298.9 | 0.13 |
| HOMA-IR (%) | 1.0 ± 0.5 | 3.6 ± 2.3 | 0.001 |
| HOMA-B (%) | 74.4 ± 26.7 | 44.2 ± 36.6 | 0.02 |
| C-peptide 0 h (pmol/L) | 614.1 ± 144.4 | 817.6 ± 349.9 | 0.06 |
| P-Cholesterol-total (mmol/L) | 5.5 ± 1.0 | 4.5 ± 0.7 | 0.01 |
| P-Cholesterol-HDL (mmol/L) | 1.7 ± 0.5 | 1.3 ± 0.4 | 0.02 |
| P-Cholesterol-LDL (mmol/L) | 3.3 ± 0.9 | 2.5 ± 0.7 | 0.02 |
| Systolic blood pressure (mmHg) | 134.3 ± 12.0 | 140.8 ± 12.7 | 0.17 |
| Diastolic blood pressure (mmHg) | 85.6 ± 9.5 | 90.8 ± 6.3 | 0.10 |
| Pulse (beats/min) | 59.5 ± 12.4 | 68.6 ± 10.8 | 0.05 |
| VO₂ max (mL/min/kg) | 32.6 ± 11.4 | 26.4 ± 7.6 | 0.10 |

Data are presented as mean ± SD. Statistics were calculated using a two-tailed *t*-test.
*HbA1c* glycated hemoglobin, *BW* body weight.
[a]Measured after 1 week without treatment.
[b]Measured after a 2 h oral glucose tolerance test.

function (HOMA-β) analyses. After isolation, satellite cells were expanded and differentiated from myoblasts into myotubes. Cells were harvested both as proliferating myoblasts (<50% confluent) and as differentiated myotubes from all participants (Fig. 1a).

We then tested whether T2D is associated with altered expression of previously unrecognized and known regulators of muscle regeneration in human myoblasts. We performed genome-wide expression analysis of myoblasts obtained from 13 controls and 13 individuals with T2D (Fig. 1b). Based on a false discovery rate (FDR) below 5% ($q < 0.05$), we identified 577 unique genes with differential expression in myoblasts from individuals with T2D versus controls (Supplementary Data 1, Sheet A). These included several genes that had not previously been studied in human myoblasts but with identified functions in other cell types or species that suggest that they may also have a role in human muscle cells, e.g., VPS39, TDP1, and MAEA[19–25], as well as genes previously implicated in muscle regeneration, e.g., FBN2, TEAD4, and STAT3[26,27] (Fig. 1c–d). This highlights differences in expression in myoblasts from individuals with T2D and healthy individuals.

DNA methylation controls cell-specific expression and myogenesis[18]. Therefore, we proceeded to relate DNA methylation to expression in human myoblasts. Using Infinium 450K BeadChips, we analyzed DNA methylation in myoblasts from 14 controls and 14 individuals with T2D, and filtered 10,992 CpG sites annotated to the 577 unique genes that exhibited differential expression in myoblasts from individuals with T2D versus controls. We then studied the correlations between methylation and expression of these 577 genes since methylation may regulate gene expression. We identified 331 differentially expressed genes that displayed nominal correlations between expression and

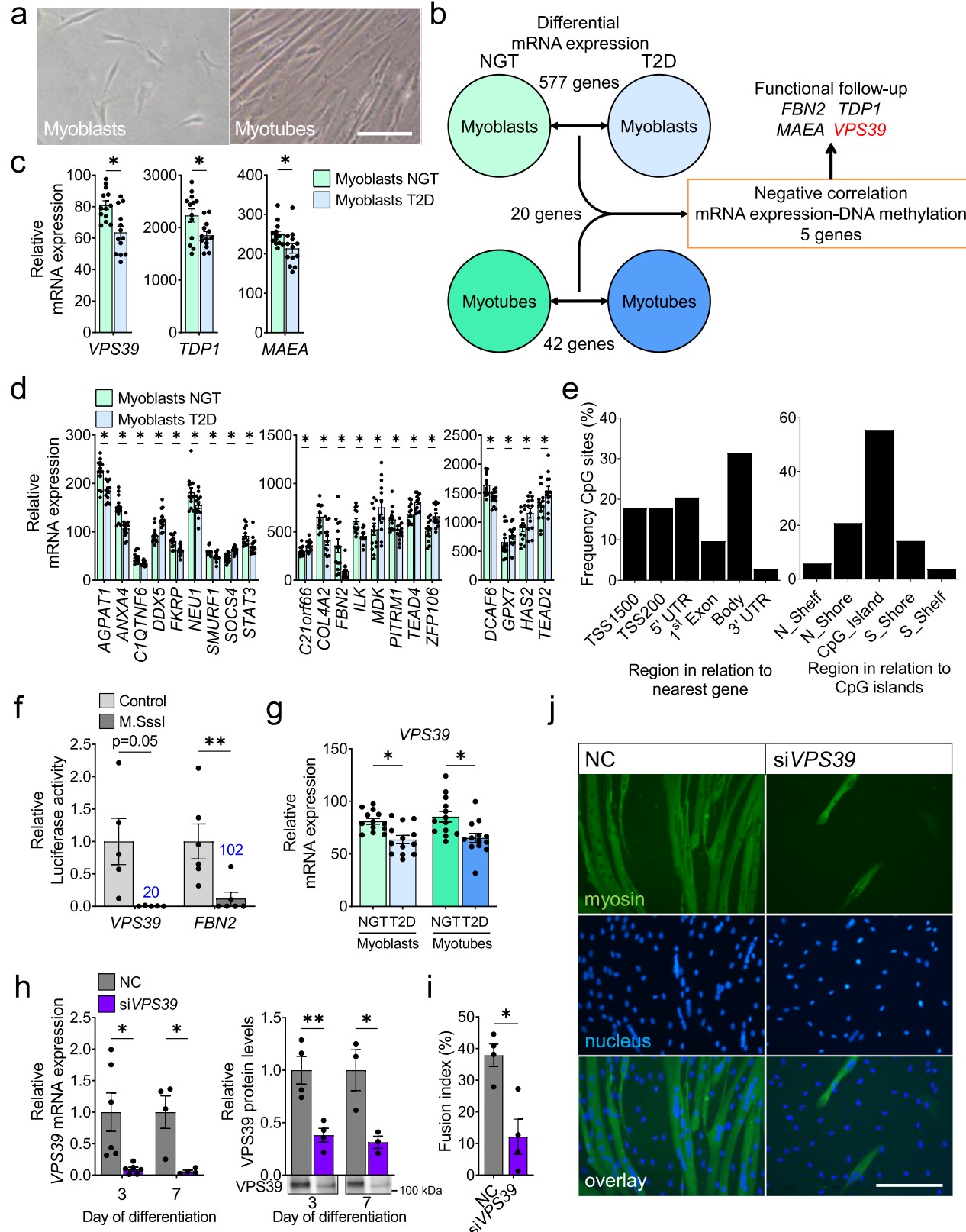

methylation of one or more CpG site ($p < 0.05$, Supplementary Data 2). These included *VPS39*, *TDP1*, *MAEA*, and *FBN2*. A large proportion of the sites with negative correlations between methylation and expression are located close to transcription start sites (TSS) (TSS200 and 5'UTR) ($p$-chi$^2$ = 0.001 compared to all analyzed sites, Fig. 1e). These data suggest that DNA

methylation may control expression in human myoblasts. To directly confirm that DNA methylation regulates the transcriptional activity, we used luciferase reporter assays to study two of the identified genes, *VPS39* and *FBN2*, presented in Fig. 1c–d and Supplementary Data 2. Indeed, higher methylation of the *VPS39* and *FBN2* promoters directly led to reduced transcriptional

**Fig. 1 Differential gene expression in myoblasts and myotubes from 13 individuals with type 2 diabetes versus 13 controls, and identification of VPS39 as a regulator of human myogenesis. a** Representative light microscope images of human proliferating myoblasts and differentiated myotubes. Images were taken for 14 individuals with type 2 diabetes and 14 controls. Scale bar 100 μm. **b** Schematic overview of analyses performed in human myoblasts and myotubes from individuals with type 2 diabetes (T2D) vs. controls (NGT, normal glucose tolerance). **c–d** mRNA expression (microarray) of genes previously not described in relation to myogenesis (*VPS39*, *TDP1*, and *MAEA*) (**c**), and genes known to be involved in muscle regeneration (**d**) that exhibit differential expression in myoblasts from individuals with T2D (blue bars) vs. NGT (green bars). $n = 13$ individuals per group. **\****q < 0.05* for T2D vs. NGT. For exact *q*-values see Supplementary Data 1, Sheet A. **e** Frequency distribution (%) of CpG sites ($n = 832$) in relation to gene regions (left panel) and CpG islands (right panel). CpG sites for which a negative correlation between DNA methylation and expression of annotated genes was established are included. TSS, transcription start site; TSS200 and TSS1500, proximal promoter, defined as 1–200 bp (base pairs) or 201–1500 bp upstream of the TSS, respectively; UTR, untranslated region; CpG island, 200 bp (or more) stretch of DNA with a C + G content of > 50% and an observed/expected CpG ratio of > 0.6; Shore, regions flanking CpG islands, 0–2000 bp; Shelf, regions flanking island shores, 0–2000 bp. **f** Reporter gene transcription measured by luciferase activity (firefly/renilla-ratio) after in vitro methylation with M.SssI (dark gray bars) or mock-methylation (Control, light gray bars) of the *VPS39* and *FBN2* promoters cloned into a CpG-free vector and transfected into C2C12 myoblasts. $n = 5$ (*VPS39*) and $n = 6$ (*FBN2*) independent experiments. Numbers in blue above the bars represent the number of target CpG sites in the respective promoter sequence. The Control for each promoter is set to 1. **\*\****p < 0.01* for M.SssI vs. Control. $p = 0.0501$ (*VPS39*), $p = 0.0068$ (*FBN2*). **g** mRNA expression (microarray) of *VPS39* in myoblasts and myotubes from individuals with T2D (blue bars) and NGT (green bars). $n = 13$ individuals per group. **\****q < 0.05* for T2D vs. NGT. For exact *q*-values see Supplementary Data 1, Sheets A and B. **h** Knockdown efficiency of *VPS39* mRNA (left panel, $n = 6$ [Day 3] and $n = 4$ [Day 7] independent experiments) and VPS39 protein (right panel, $n = 4$ [Day 3] and $n = 3$ [Day 7] independent experiments) after siRNA silencing of VPS39 (si*VPS39*, purple bars) throughout cell differentiation. Negative control (NC, gray bars) at each time point is set to 1. Representative blots are shown. **\****p < 0.05*, **\*\****p < 0.01* for si*VPS39* vs. NC. $p = 0.0302$ (*VPS39* mRNA Day 3), $p = 0.0379$ (*VPS39* mRNA Day 7), and $p = 0.004$ (VPS39 protein Day 3), $p = 0.0389$ (VPS39 protein Day 7). **i–j** Assessment of myotube formation (fusion index) at day 7 of differentiation in si*VPS39* and NC (**i**). $n = 4$ independent experiments, **\****p < 0.05* for si*VPS39* vs. NC, $p = 0.0266$. **j** Representative images from the assay, showing reduced myotube formation after VPS39-silencing. Scale bar 200 μm. Bars represent mean values and error bars display SEM (**c–d**, **f–i**). Statistical significance determined by linear regression adjusted for age, BMI and sex for T2D vs. NGT (**c–d**, **g**). *P*-values were adjusted for multiple comparisons with false discovery rate (FDR) analysis (**c–d**, **g**). Statistical significance determined by paired two-tailed *t*-test (**f**, **h–i**).

---

activity of the reporter genes, supporting an epigenetic regulation of expression in myoblasts from individuals with T2D (Fig. 1f).

We next asked whether genes that showed differential expression in myoblasts from individuals with T2D versus control individuals, also showed altered expression after differentiation into multinucleated myotubes (Fig. 1a). We performed genome-wide expression analysis of myotubes from the same 13 individuals with T2D and 13 control individuals, also included for analysis of expression in myoblasts. The analysis revealed 42 unique genes that were differentially expressed in myotubes from individuals with T2D versus controls at FDR below 5% ($q < 0.05$, Fig. 1b and Supplementary Data 1, Sheet B). Notably, 20 of these genes, including *VPS39*, *TDP1*, *MAEA*, and *FBN2*, were among those also differentially expressed in the myoblasts (Fig. 1b, g, Supplementary Fig. 1a–c and Supplementary Data 1, Sheet A).

To identify new regulators of muscle regeneration and function, we asked whether any of the genes with reduced expression in both myoblasts and myotubes from individuals with T2D versus controls, and with an inverse correlation between their expression and DNA methylation, also have a functional role in human myogenesis. We mainly focused on genes that had not been previously studied in human muscle cells, but with known functions in other cell types or species that would suggest an impact also on myogenesis[19–26]. Based on these criteria, we selected four genes (*VPS39*, *TDP1*, *MAEA*, and *FBN2*) for functional follow-up experiments (Fig. 1b, g, Supplementary Fig. 1a–c and Supplementary Data 2). To model the situation seen in myoblasts and myotubes from individuals with T2D, we silenced these four genes by using siRNA in human myoblasts from healthy individuals throughout cell differentiation. Knockdown was confirmed at both an early stage of differentiation and after differentiation into myotubes (Fig. 1h and Supplementary Fig. 1d–f). Next, we analyzed the fusion index to examine whether gene silencing affected human myotube formation. Silencing of VPS39 resulted in an almost complete lack of human myotube formation (Fig. 1i–j). Silencing of MAEA or FBN2 did not affect the fusion index, while silencing of TDP1 resulted in a modest reduction in fusion index (Supplementary Fig. 1g–i). These

observations suggest that VPS39 is a putative regulator of human myogenesis.

**VPS39 controls human myoblast function and differentiation via autophagy and epigenetic mechanisms.** *VPS39* encodes a protein called Vam6/Vps39-like protein and is a previously unrecognized regulator of human myogenesis. In other tissues, VPS39 is part of the complex that mediates fusion of autophagosomes with lysosomes (HOPS complex) and it has been found in mouse models of myotonic dystrophy type 1[20,21,25]. Because of severe effects of VPS39-silencing on myotube formation (Fig. 1i–j), we decided to dissect its role in the differentiation and function of human muscle cells.

To investigate the role of VPS39 in human myogenesis, we silenced VPS39 in myoblasts and monitored the effects of silencing on protein levels, gene expression, DNA methylome, and cell physiology as the cells differentiated (see Fig. 2a for experimental set up). VPS39-silencing resulted in reduced VPS39 protein levels at both an early stage of differentiation (day 3) and after differentiation into myotubes (day 7) (Fig. 1h). To understand the mechanisms underlying the profound effect of VPS39 knockdown on myotube formation, we compared gene expression in si*VPS39* versus control myoblasts at day 3 of differentiation. The analysis revealed that 2635 unique genes were differentially expressed in VPS39-silenced versus control cells ($q < 0.05$), including *VPS39* itself (Supplementary Data 3, Sheet A). The expression data clearly indicated that VPS39-silenced cells did not exit the cell cycle and start to differentiate (Fig. 2b–d, and Supplementary Data 3, Sheet B). For example, gene sets related to muscle structure and function, as well as key myogenic transcription factors (TFs) and muscle-specific genes had significantly lower expression (Fig. 2b–c). On the other hand, DNA replication and cell cycle gene sets had higher expression (Fig. 2d). Interestingly, several gene sets known to play a role in autophagy were upregulated in VPS39-silenced cells, e.g., mTOR-signaling pathway, p53-signaling pathway, and lysosome (Fig. 2d). In addition, the expression of numerous epigenetic enzymes was altered in VPS39-silenced cells (Fig. 2e). These are more than

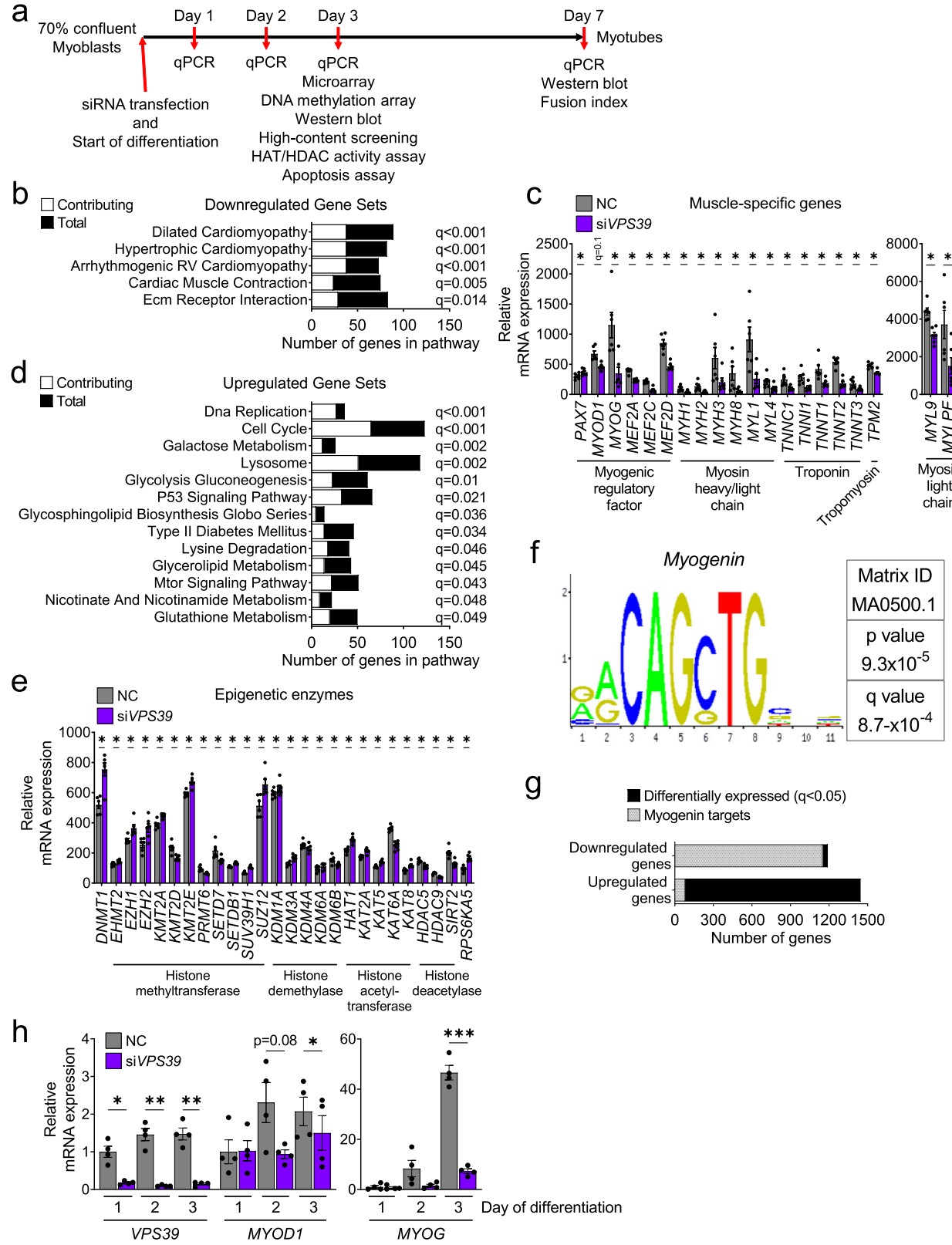

expected by chance based on the search terms epigenetics and histone, and a chi$^2$-test ($p$-chi$^2 < 0.05$).

Considering the importance of specific TFs for the regulation of myogenesis[8], we used the bioinformatics prediction tools PSCAN[28] and JASPAR 2016[29] to search for TF binding motifs that were enriched in promoter regions of downregulated and upregulated genes, after VPS39 knockdown in human myoblasts (Supplementary Data 3, Sheets C and D). One motif enriched for downregulated genes was that of myogenin (MYOG), whose expression was also lower in VPS39-silenced cells (Fig. 2c, f). Remarkably, 97% of the downregulated genes contained binding motif(s) for myogenin in their promoter region, whereas only

**Fig. 2 VPS39 knockdown results in reduced expression of key myogenic genes in human myoblasts. a** Study design for VPS39-silencing and analyses performed in human myoblasts and myotubes. HAT histone acetyltransferase, HDAC histone deacetylase. **b–e** Microarray expression analysis in VPS39-silenced myoblasts (siVPS39) and negative control (NC) at day 3 of differentiation. Gene set enrichment analysis (GSEA) enriched gene sets (FDR < 5%) that were downregulated (**b**) or upregulated (**d**) (see also Supplementary Data 3, Sheet B). Bars represent the number of differentially expressed genes contributing to each gene set (white bars), and the total number of genes in each gene set (black bars). mRNA expression (microarray) of myogenic regulatory factors and muscle-specific genes (**c**), and genes encoding epigenetic enzymes (**e**) in siVPS39 (purple bars) and NC (gray bars). n = 6 independent experiments. *q < 0.05 for siVPS39 vs. NC. For exact q-values see Supplementary Data 3, Sheet A. RV right ventricular. **f** Using PSCAN analysis, the binding motif for myogenin (Myog), presented here, was found to be significantly enriched in the promoter region of genes downregulated after VPS39-silencing. For full PSCAN analysis, see Supplementary Data 3, Sheet C. **g** The number of down- or upregulated genes with the myogenin binding motif in their promoter region (gray bars) in relation to all differentially expressed genes (black bars) for siVPS39 vs. NC. **h** Relative expression (qPCR) of VPS39, MYOD1, and MYOG at day 1 (24 h), day 2 (48 h), and day 3 (72 h) after the start of siVPS39 transfection and differentiation. n = 4 independent experiments. NC at day 1 for each gene is set to 1. *p < 0.05, **p < 0.01, ***p < 0.001 for siVPS39 vs. NC. p = 0.0114 (VPS39 Day 1), p = 0.0033 (VPS39 Day 2), p = 0.0022 (VPS39 Day 3), and p = 0.0819 (MYOD1 Day 2), p = 0.0211 (MYOD1 Day 3), and p = 0.0009 (MYOG Day 3). Bars represent mean values and error bars display SEM (**c**, **e**, **h**). Statistical significance was determined by paired two-tailed t-test (**c**, **e**, **h**). P-values were adjusted for multiple comparisons with false discovery rate (FDR) analysis (**b**, **c**, **d**, **e**, **f**).

5.7% of the upregulated genes had this motif (Fig. 2g and Supplementary Data 3, Sheets C–F). Myogenin is a key regulator of myogenesis[8], and the expression of several genes encoding TFs that activate MYOG transcription[30] was reduced in VPS39-silenced cells. These TFs include MYOD1, MEF2s (MEF2A, MEF2C, and MEF2D), and E-box proteins (TCF3, TCF4, and TCF12) (Fig. 2c and Supplementary Data 3, Sheets E and F). Hence, VPS39, directly or indirectly, regulates myogenesis by disrupting MYOG expression.

To delineate the primary and secondary effects of VPS39-deficiency on myogenesis, we examined the expression of VPS39, MYOD1, and MYOG on day 1, 2, and 3 of the differentiation in VPS39-silenced human myoblasts (Fig. 2h). VPS39 expression was downregulated already after 1 day, while MYOD1 was reduced after 2 days and MYOG after 3 days (Fig. 2h). These data suggest that the typically high and/or upregulated expression of MYOD1 and MYOG at an early stage of differentiation is partially repressed or delayed in VPS39-silenced myoblasts. Moreover, MEF2C and myosin protein levels were markedly reduced in VPS39-silenced cells at day 7 of differentiation (Supplementary Fig. 1j–k), further demonstrating the key role of VPS39 in human myogenesis.

In view of our microarray results (Fig. 2b–e), we hypothesized that reduced VPS39 levels in myoblasts might alter autophagy and thereby cell metabolism. This may result in altered activity of epigenetic enzymes followed by changes in epigenetic marks and the expression of myogenic TFs. Subsequently, myogenesis would be impaired. We proceeded to explore this hypothesis.

Rodent data support that proper autophagy is important for myogenesis[31], but this has not been verified in human muscle cells. Moreover, VPS39 is part of the complex mediating fusion of autophagosomes with lysosomes, but the role of VPS39 for the regulation of autophagy in human muscle cells remain to be elucidated[20,21]. To investigate the importance of autophagy also during human myogenesis, human myoblasts were treated with Bafilomycin A1 (Baf-A1) for 3 h per day during the first 3 days of differentiation, and key autophagy markers (LC3B, p62, LAMP1, and LAMP2)[32] were studied using both automated high-content screening (HCS) and Western blot analyses (see Supplementary Fig. 2a for experimental setup). Baf-A1 inhibits the fusion of autophagosomes with lysosomes[32]. Indeed, Baf-A1 altered autophagy (Supplementary Fig. 2b–d) and reduced the expression of myogenic markers (Supplementary Fig. 2e), similar to what we observed in VPS39-silenced cells (Fig. 2c, h). In addition, MYOD1 protein levels were significantly decreased (Supplementary Fig. 2f). These observations further demonstrate the importance of autophagy during human myogenesis.

Next, we investigated the effects of VPS39 on basal autophagy in human myoblasts using both HCS and Western blot analyses

for key markers of autophagy — LC3B, p62, LAMP1, and LAMP2. As shown in Fig. 3a–c, VPS39-silenced cells displayed increased amount and size of autophagosomes, determined by quantification of LC3B spot number and area, and increased p62, a marker of selective autophagy[32]. p62 protein levels are inversely correlated with autophagic activity[32], indicating impaired autophagic activity in VPS39-silenced cells. Moreover, the spot number and area of lysosomal markers LAMP1 and LAMP2 were also increased in the HCS analysis (Fig. 3a–c). In addition, the protein levels of LC3B-II and p62 (detected by Western blot) were increased in VPS39-silenced cells, whereas LAMP1 and LAMP2 were not altered (Fig. 3d). The discrepancies in LAMP1/2 levels between the HCS and Western blot assays may depend on that these two separate methods detect different aspects of the protein dynamics, and that HCS is able to detect more subtle changes in LAMP1/2 levels by measuring spot number and spot area per cell compared to Western blot. Together, these results support that knockdown of VPS39 alters basal autophagy in human myoblasts, similar to what was observed after inhibition of autolysosome formation (Supplementary Fig. 2b–d).

To further characterize the role of VPS39 for the regulation of the autophagic process, we performed comprehensive autophagy flux analyses[33]. Human myoblasts were treated with Baf-A1 under both basal and starvation (serum- and amino acid-free) conditions to induce autophagy, and key markers of autophagy were monitored using the two methods described above. As determined by Western blot, both LC3B-II and p62 protein levels were increased in VPS39-silenced cells compared to control (Fig. 3e–f). As expected, LC3B-II was increased upon starvation and was further increased in response to Baf-A1-treatment due to an accumulation of autophagosomes (Fig. 3e). p62 was increased in response to Baf-A1-treatment in both control and VPS39-silenced cells, but only control cells displayed a significant decrease in p62 upon starvation (Fig. 3f). The autophagic flux (calculated as LC3B-II levels in Baf-A1-treated compared to control cells) was decreased in VPS39-silenced cells, and only control cells displayed an increased autophagic flux upon starvation (Fig. 3g and Supplementary Fig. 2g). LAMP1/2 protein levels were not different between the genotypes, and not altered in response to any of the treatments (Supplementary Fig. 2h–i). In line with the results on protein level, p62 spot number and area were increased overall in VPS39-silenced cells, and only control cells displayed significantly decreased p62 upon activation of autophagy (Fig. 3h–j). The reduction in p62 in response to starvation, determined by both Western blot and HCS, was significantly larger in control cells indicating an impaired autophagic activity in the VPS39-silenced cells (Supplementary Fig. 2j–l). Moreover, both LAMP1 (Fig. 3k–m) and LAMP2 (Fig. 3n–p) spot number and area were

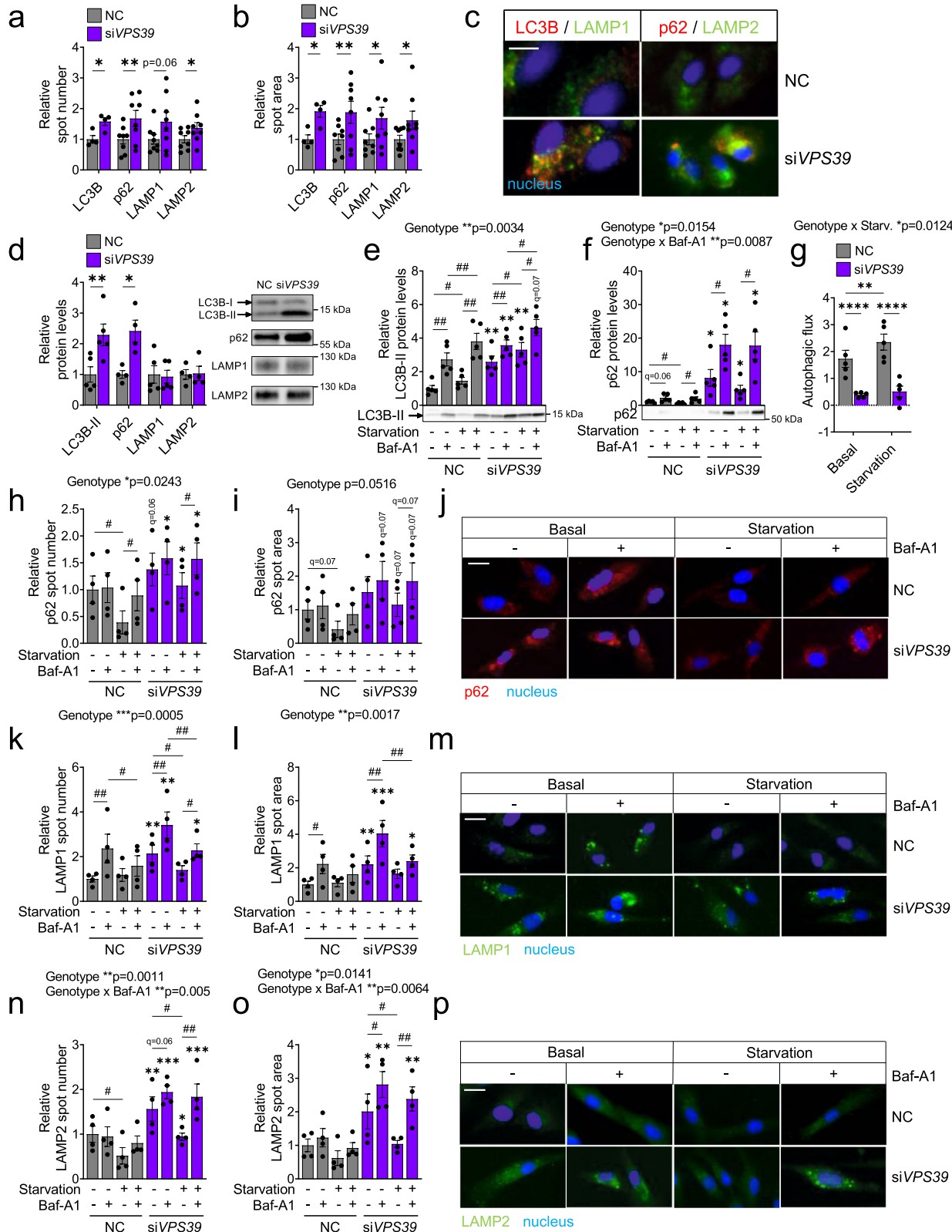

significantly increased in VPS39-silenced cells. These results support that VPS39 is required for functional autophagy in human myoblasts and that VPS39-silencing is associated with a reduced autophagic flux, likely due to defects in the late stages of the autophagy process. Furthermore, VPS39 knockdown alone mimicked effects on altered basal autophagy and impaired myogenesis observed in Baf-A1-treated myoblasts.

The autophagic process has been linked to metabolic remodeling and myoblast differentiation in rodents[31,34]. We proceeded to examine whether VPS39-silenced human muscle cells have altered activation of key proteins involved in metabolic pathways i.e. glucose uptake and glycogen synthesis. VPS39-silenced cells exhibited decreased insulin-induced Akt phosphorylation, at both serine 473 (Ser473) and threonine 308 (Thr308)

**Fig. 3 VPS39-silencing impairs autophagic flux in human myoblasts. a–c** High-content screening (HCS) analysis using spot detection application to identify autophagy markers LC3B, p62, LAMP1, and LAMP2 in VPS39-silenced myoblasts (siVPS39, purple bars) and negative control (NC, gray bars) at day 3 of differentiation. $n = 4$ (LC3B) and $n = 8$ (p62, LAMP1/2) independent experiments. Graphs show a relative number of detected spots per cell (**a**) and area per spot (**b**) for each marker. NC is set to 1. *$p < 0.05$, **$p < 0.01$ for siVPS39 vs. NC. $p = 0.0274$ (**a**, LC3B), $p = 0.0034$ (**a**, p62), $p = 0.0606$ (**a**, LAMP1), $p = 0.0112$ (**a**, LAMP2), and $p = 0.035$ (**b**, LC3B), $p = 0.0059$ (**b**, p62), $p = 0.0399$ (**b**, LAMP1), $p = 0.0317$ (**b**, LAMP2). **c** Representative images from the analyses in (**a–b**), showing immunostaining of LC3B (left panel, red) and LAMP1 (left panel, green), and p62 (right panel, red) and LAMP2 (right panel, green). Scale bar 20 μm. **d** Protein levels of LC3B-II, p62, LAMP1, and LAMP2 in siVPS39 (purple bars) and NC (gray bars) myoblasts at day 3 of differentiation. $n = 5$ (LC3B-II, p62, LAMP1) and $n = 4$ (LAMP2) independent experiments. NC is set to 1. Representative blots are shown. *$p < 0.05$, **$p < 0.01$ for siVPS39 vs. NC. $p = 0.0062$ (LC3B-II), $p = 0.0137$ (p62). **e–p** Autophagic flux measurements in siVPS39 (purple bars) and NC (gray bars) myoblasts at day 3 of differentiation in both the basal state and after starvation (3 h) to induce autophagy, and in the absence or presence of the lysosomal inhibitor Bafilomycin A1 (Baf-A1, 100 nM). NC in the basal, vehicle-treated state is set to 1. **e–g** Protein levels (Western blot) of LC3B-II (**e**) and p62 (**f**). $n = 5$ independent experiments. Representative blots are shown. #$q < 0.05$, ##$q < 0.01$ for comparisons between treatments within each genotype, and *$q < 0.05$, **$q < 0.01$ for siVPS39 vs. NC for each treatment. For exact $q$-values see Supplementary Table 1. **g** Autophagic flux calculated as LC3B-II protein levels in Baf-A1-treated vs. vehicle-treated cells under basal and starvation conditions for each genotype (see also Supplementary Fig. 2g). **$p < 0.01$, ****$p < 0.0001$ (Fisher's LSD test). $p = 0.000078$ (Basal; siVPS39 vs. NC), $p = 0.000022$ (Starvation: siVPS39 vs. NC), $p = 0.0016$ (NC: Basal vs. Starvation). Starv., starvation (**h–p**) HCS analysis using spot detection application to identify p62 (**h–j**), LAMP1 (**k–m**), and LAMP2 (**n–p**). $n = 4$ independent experiments. Graphs show the relative number (**h**, **k**, **n**) and area (**i**, **l**, **o**) of detected spots per cell for each marker. #$q < 0.05$, ##$q < 0.01$ for comparisons between treatments within each genotype, and *$q < 0.05$, **$q < 0.01$, ***$q < 0.001$ for siVPS39 vs. NC for each treatment. **j**, **m**, **p** Representative images from the analyses, showing immunostaining of p62 (**j**), LAMP1 (**m**), and LAMP2 (**p**). Scale bar 20 μm. Bars represent mean values and error bars display SEM (**a–b**, **d–i**, **k–l**, **n–o**). Statistical significance determined by paired two-tailed $t$-test (**a–b**, **d**). The effects of genotype, starvation, and Baf-A1-treatment stated above the graphs were calculated with repeated measures three-way ANOVA (**e–f**, **h–i**, **k–l**, **n–o**) or two-way ANOVA (**g**). $P$-values were adjusted for multiple comparisons with false discovery rate (FDR) analysis (**e–f**, **h–i**, **k–l**, **n–o**).

compared to control cells (Fig. 4a). The phosphorylation of downstream Akt substrates TBC1D4 that controls GLUT4 translocation[35], and glycogen synthase kinase 3 (GSK3) that regulates glycogen synthase activity[36], was significantly and nominally decreased, respectively, in insulin-stimulated VPS39-silenced cells versus control (Fig. 4b–c). Phosphorylation of TBC1D4 is associated with increased glucose uptake and phosphorylation of GSK3 is associated with activation of glycogen synthesis. These results suggest that the overall activity of the insulin signaling pathway is perturbed in VPS39-silenced myoblasts, and that some metabolic pathways may be altered.

Metabolic changes can affect the activity of epigenetic enzymes in muscle stem cells[37,38]. We observed that the expression of genes encoding epigenetic enzymes was altered in VPS39-silenced myoblasts (Fig. 2e), and we therefore postulated that the activity of these enzymes was also altered. To test this, we analyzed the activity and protein levels of epigenetic enzymes in VPS39-silenced cells. The overall activity of histone acetyl transferases (HAT) was reduced, while the overall histone deacetylase (HDAC) activity in nuclear extracts was not significantly affected after VPS39 knockdown (Fig. 4d). In line with the expression analysis, the DNA and histone methyltransferases DNMT1, DNMT3B, and EZH2 protein levels were significantly higher in VPS39-silenced cells (Fig. 4e), which may reflect key mechanisms contributing to impaired differentiation in myoblasts with reduced VPS39 levels[39–42]. In addition, the nuclear protein levels of HAT1 were nominally reduced and p300 were increased, while no differences were found in the cytoplasm (Fig. 4e). HDAC4 and HDAC5 levels were also divergently regulated (Fig. 4e), as is often the case with class II HDACs[43,44].

We next asked whether the changes seen in epigenetic enzymes were associated with epigenetic alterations in VPS39-silenced human muscle cells. We observed nominally altered DNA methylation at 5045 CpG sites annotated to 72% (1889) of the genes that exhibited differential expression after VPS39 knockdown (Fig. 4f and Supplementary Data 4, $p < 0.05$). The number of sites with nominal methylation differences were significantly more than expected by chance ($p$-chi$^2 < 0.01$). Pathway analysis of these 1889 genes revealed enrichment of processes as the cell cycle, cell death, muscle cell differentiation, and cytoskeleton organization (Fig. 4g). Specific genes of

importance during myogenesis whose DNA methylation was altered in VPS39-silenced cells included *MEF2*s, myosin heavy and light chains (*MHC* and *MLC*), and genes encoding proteins important for myocyte function[45,46] (Supplementary Data 4). In addition, we observed that acetylation of histone 3 (ac-H3) was elevated in VPS39-silenced myoblasts at day 3 of differentiation (Fig. 4h), which may be due to increased p300 and reduced HDAC5 levels (Fig. 4e). Global ac-H3 levels are expected to decrease during myoblast differentiation[47], which we clearly see in our control cells (Fig. 4i). Specific acetylation of H3 was significantly decreased during differentiation in control cells whereas this response was altered in VPS39-silenced cells that displayed increased levels of H3 acetylation compared to control at both day 3 and 7 of differentiation (Fig. 4i). Together, these data strengthen our hypothesis that reduced VPS39 levels affect epigenetic enzymes and result in an altered epigenome.

To further dissect the mechanisms underlying the impaired differentiation of VPS39-silenced myoblasts, and since the connection between autophagy and apoptosis has been documented[48], we measured Caspase 3/7 activity as an estimate of cellular apoptosis. Caspase 3/7 activity was increased in VPS39-silenced cells (Fig. 4j). This is in line with both our microarray and HCS data for VPS39-silenced cells showing that the expression of pro-apoptotic genes, e.g., *CASP3* and *BAX* was higher (Supplementary Data 3, Sheet A), and cell nuclei size was smaller in these cells (Fig. 4k).

Collectively, the data presented above support a model whereby low VPS39 levels in myoblasts impair autophagy, which may result in disturbed homeostasis and alterations of epigenetic enzymes and the epigenome (Supplementary Fig. 3). Consequently, the expression of key myogenic regulatory factors (MRFs and MEFs) and muscle-specific genes is reduced, and a reduced proportion of myoblasts differentiate into myotubes. Instead, apoptosis is increased.

**VPS39-deficiency in mice leads to reduced glucose uptake and altered gene expression in muscle.** To further elucidate the mechanisms whereby reduced VPS39 expression may contribute to impaired muscle function and T2D, we studied mice heterozygous for a germ-line deletion of *Vps39* (*Vps39*$^{+/-}$ mice)[49]. Heterozygous mice were used because total VPS39-deficiency is embryonically lethal[49]. Further, we reasoned that the reduction of

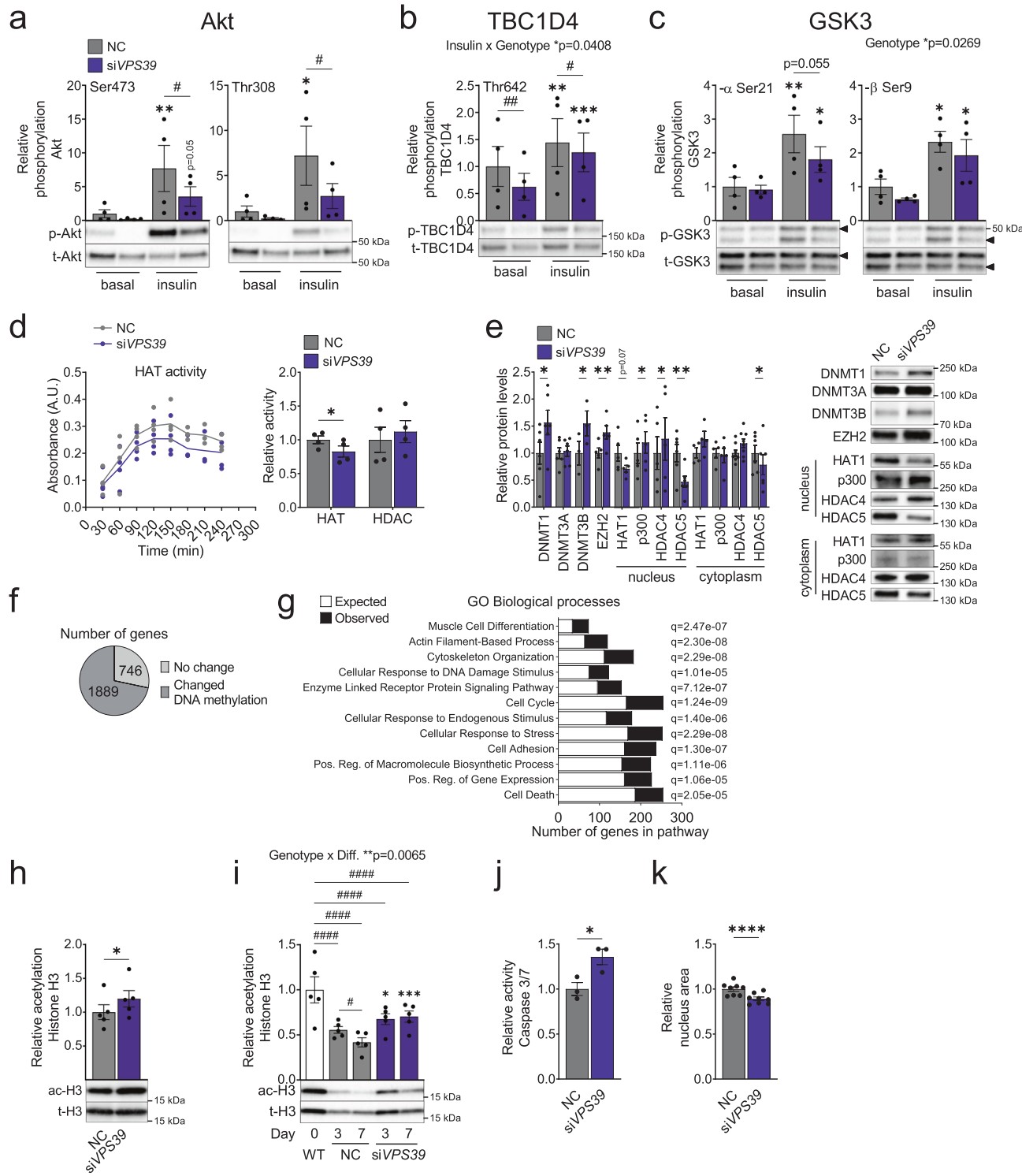

*Vps39* expression in *Vps39*[+/−] mice would mimic the situation in humans with T2D. Skeletal muscle from *Vps39*[+/−] mice contained lower *Vps39* levels compared to wild-type (WT) littermates (Fig. 5a), confirming that they would be a suitable model to study the role of VPS39 in muscle dysregulation. We observed no differences in body weight or body composition between WT and *Vps39*[+/−] mice (Supplementary Fig. 4a–c). We then examined glucose homeostasis in *Vps39*[+/−] mice. First, an OGTT was used to examine glucose tolerance in vivo (Fig. 5b). We observed that the fold change in plasma glucose levels was significantly higher in *Vps39*[+/−] than in WT mice during the first 15 min of glucose

challenge (Fig. 5c), with no difference in insulin levels in *Vps39*[+/−] males but a compensatory increased fold change in insulin secretion in *Vps39*[+/−] females (Supplementary Fig. 4d–e). Next, we measured tissue-specific glucose uptake using a tracer, i.e., by oral administration of [3]H-deoxy-glucose together with D-glucose sufficient to induce an insulin response in the mice (Fig. 5d and Supplementary Fig. 4f). Glucose uptake was significantly reduced in extensor digitorum longus (EDL) muscle of *Vps39*[+/−] versus WT mice (Fig. 5d), demonstrating that VPS39-deficiency is associated with glucose intolerance and impaired muscle function in mice, which is in line with what is seen in patients with T2D. To

**Fig. 4 Reduction of VPS39 levels alters the epigenome and insulin signaling in human myoblasts. a–c** Specific phosphorylation (intensity of phosphorylation divided by total levels for each corresponding protein) for Akt at Ser473 (**a**, left panel) and Thr308 (**a**, right panel), TBC1D4 at Thr642 (**b**), GSK3-α at Ser21 (**c**, left panel) and GSK3-β at Ser9 (**c**, right panel) in the basal state and after insulin stimulation (100 nM, 30 min) in VPS39-silenced myoblasts (siVPS39, purple bars) and negative control (NC, gray bars) at day 3 of differentiation. $n = 4$ independent experiments. NC in the basal state is set to 1. Representative blots are shown. #$p < 0.05$, ##$p < 0.01$ for siVPS39 vs. NC for each treatment, and *$p < 0.05$, **$p < 0.01$, ***$p < 0.001$ for basal vs. insulin (Fisher's LSD test). For exact $p$-values see Supplementary Table 1. **d** Kinetics of histone acetyltransferase (HAT) activity measurement (left panel). HAT (area under the curve) and histone deacetylase (HDAC) (fixed point) activity in nuclear extracts from siVPS39 (purple) and NC (gray) myoblasts at day 3 of differentiation (right panel). $n = 4$ independent experiments. NC is set to 1. *$p < 0.05$ for siVPS39 vs. NC. $p = 0.0126$ (HAT). **e** Protein levels (Western blot) of epigenetic enzymes from siVPS39 (purple bars) and NC (gray bars) myoblasts at day 3 of differentiation. Protein levels were measured in whole-cell lysates, or in nuclear and cytosolic fractions as indicated. $n = 6$ (DNMT1, DNMT3A, HDAC5 [nucleus and cytoplasm]), $n = 3$ (DNMT3B), $n = 5$ (EZH2, HAT1 [nucleus], p300 [nucleus and cytoplasm], HDAC4 [nucleus]), $n = 4$ (HAT1 [cytoplasm]), $n = 8$ (HDAC4 [cytoplasm]). NC is set to 1. Representative blots are shown. Statistical analysis was performed on log2-transformed values. *$p < 0.05$, **$p < 0.01$ for siVPS39 vs. NC. $p = 0.0259$ (DNMT1), $p = 0.0374$ (DNMT3B), $p = 0.0081$ (EZH2), $p = 0.0697$ (HAT1 [nucleus]), $p = 0.0309$ (p300 [nucleus]), $p = 0.0493$ (HDAC4 [nucleus]), $p = 0.001$ (HDAC5 [nucleus]), $p = 0.0463$ (HDAC5 [cytoplasm]). DNMT, DNA methyltransferase, HAT histone acetyltransferase, HDAC histone deacetylase. **f** The number of genes with altered DNA methylation at one or more CpG site (dark gray) or no change (light gray) among the 2635 genes with differential expression in siVPS39 vs. NC myoblasts. **g** The number of observed (black bars) and expected (white bars) genes for a selection of GO cellular processes enriched among genes with differential DNA methylation and gene expression in siVPS39 vs. NC myoblasts. Bars sorted by ratio (observed/expected). GO gene ontology. **h–i** Western blot analysis of acetylated histone 3 (ac-H3) levels related to the total amount of H3 in siVPS39 (purple bars) and NC (gray bars) myoblasts at day 3 of differentiation (**h**), and at days 0, 3, and 7 of differentiation (**i**). $n = 5$ independent experiments. NC/WT is set to 1. Representative blots are shown. *$p < 0.05$ for siVPS39 vs. NC in (**h**). $p = 0.012$. #$q < 0.05$, ####$q < 0.0001$ for comparisons between time points within each genotype, and *$q < 0.05$, ***$q < 0.001$ for siVPS39 vs. NC at each time point in (**i**). For exact $q$-values see Supplementary Table 1. Diff. differentiation. **j–k** Apoptosis measured as Caspase 3/7 activity (**j**, $n = 3$ independent experiments) and nucleus size (area of the DAPI-stain measured in the HCS assay) (**k**, $n = 8$ independent experiments) in siVPS39 (purple bars) and NC (gray bars) at day 3 of differentiation. NC is set to 1. *$p < 0.05$, ****$p < 0.0001$ for siVPS39 vs. NC. $p = 0.0259$ (**j**), $p = 0.000051$ (**k**). Bars (in (**a–c**, **d**), right panel, and (**e**, **h–k**)) or points (in (**d**), left panel) represent mean values and error bars display SEM (**a–e**, **h–k**). The effects of genotype, and insulin-treatment or differentiation stated above the graphs were calculated with repeated measures two-way ANOVA (**a–c**, **i**). $P$-values were adjusted for multiple comparisons with false discovery rate (FDR) analysis (**g**, **i**). Statistical significance determined by paired two-tailed $t$-test (**d–e**, **h**, **j–k**).

dissect the cause of this in vivo perturbation, we analyzed gene expression in skeletal muscle of Vps39$^{+/−}$ mice. Microarray analysis revealed 1641 nominally differentially expressed genes in the muscle of Vps39$^{+/−}$ versus WT mice ($p < 0.05$, Supplementary Data 5, Sheet A). To better understand the biological relevance of these expression differences, and relate them to our human data, we searched the gene ontology (GO) terms for these 1641 genes using four search terms: autophagy, epigenetics and histones, muscle (excluding cardiac and smooth muscle), as well as oxidative phosphorylation and respiratory chain. We then used chi2-tests to examine an overrepresentation of differentially expressed genes belonging to these search terms versus all analyzed genes. The analysis revealed an overrepresentation of genes associated with epigenetics and histones, and a nominal significant enrichment of genes associated with muscle (Fig. 5e and Supplementary Data 5, Sheets A and B). Some differentially expressed genes annotated to these search terms/biological processes are presented in Fig. 5f. Moreover, the mRNA and protein levels of autophagy protein 5 (ATG5) and DNMT3B correlated positively in the muscle of the mice (Supplementary Fig. 4g). We proceeded to perform a gene set enrichment analysis (GSEA) on the complete expression data set in Vps39$^{+/−}$ versus WT mice[50]. This analysis revealed three significant pathways, including the proteasome, ribosome and spliceosome (Supplementary Data 5, Sheet C).

We conclude that mimicking the VPS39-deficiency observed in muscle cells from individuals with T2D using a mouse model (Vps39$^{+/−}$) results in impaired glucose uptake in muscle and altered expression of genes affecting autophagy, epigenetic programming, muscle development, and metabolism, highlighting the possible role for VPS39 in muscle pathology.

**Markers for autophagy and epigenetic enzymes are altered in myoblasts and myotubes from individuals with T2D.** We next asked whether the observations in VPS39-silenced human myoblasts and Vps39$^{+/−}$ mice are also reflected in individuals with T2D. Higher HOMA-IR (Table 1) supported the notion of insulin

resistance and an impaired muscle function in individuals with T2D. First, we tested if LAMP2 protein levels were altered during differentiation in muscle cells from individuals with T2D versus controls. Based on the overall two-way ANOVA (*$p < 0.05$ for NGT vs. T2D), the average LAMP2 levels were higher throughout differentiation in individuals with T2D compared to controls (Fig. 6a), which agreed with observations in VPS39-silenced cells (Fig. 3a–c) and are generally in line with previous findings[51].

Next, we studied epigenetic enzymes in muscle cells from individuals with T2D and control individuals. We analyzed the protein levels of all three DNMTs during differentiation (Fig. 6b). Protein levels of the DNMTs responsible for de novo methylation, DNMT3A and an alternative isoform of DNMT3B, were increased during differentiation only in muscle cells from individuals with T2D. The alternative isoform of DNMT3B was also higher in myotubes (day 7) from individuals with T2D versus control individuals (Fig. 6b). siRNA-mediated DNMT3B-silencing confirmed that the alternative and larger-mass (~120 kDa) isoform was DNMT3B (Supplementary Fig. 5a), potentially SUMOylated DNMT3B[52]. DNMT1 levels were reduced as differentiation progressed in both groups (Fig. 6b), which is expected when cells exit the cell cycle[39]. In agreement with cells from individuals with T2D, both DNMT3B and DNMT1 levels were decreased at day 7 versus day 3 in VPS39-silenced cells (Fig. 6b and Supplementary Fig. 5b).

In general, although different methods have been used, several of the data for cells from individuals with T2D were similar to our results for VPS39-silenced human myoblasts and Vps39$^{+/−}$ mice, suggesting a model whereby reduced VPS39 levels alter autophagy and epigenetic enzymes, thereby negatively affecting myogenesis and muscle function (Supplementary Fig. 3).

**Individuals with T2D show abnormal DNA methylation changes during myogenesis.** Having established that epigenetic enzymes responsible for de novo DNA methylation are upregulated during myogenesis only in cells from individuals with T2D

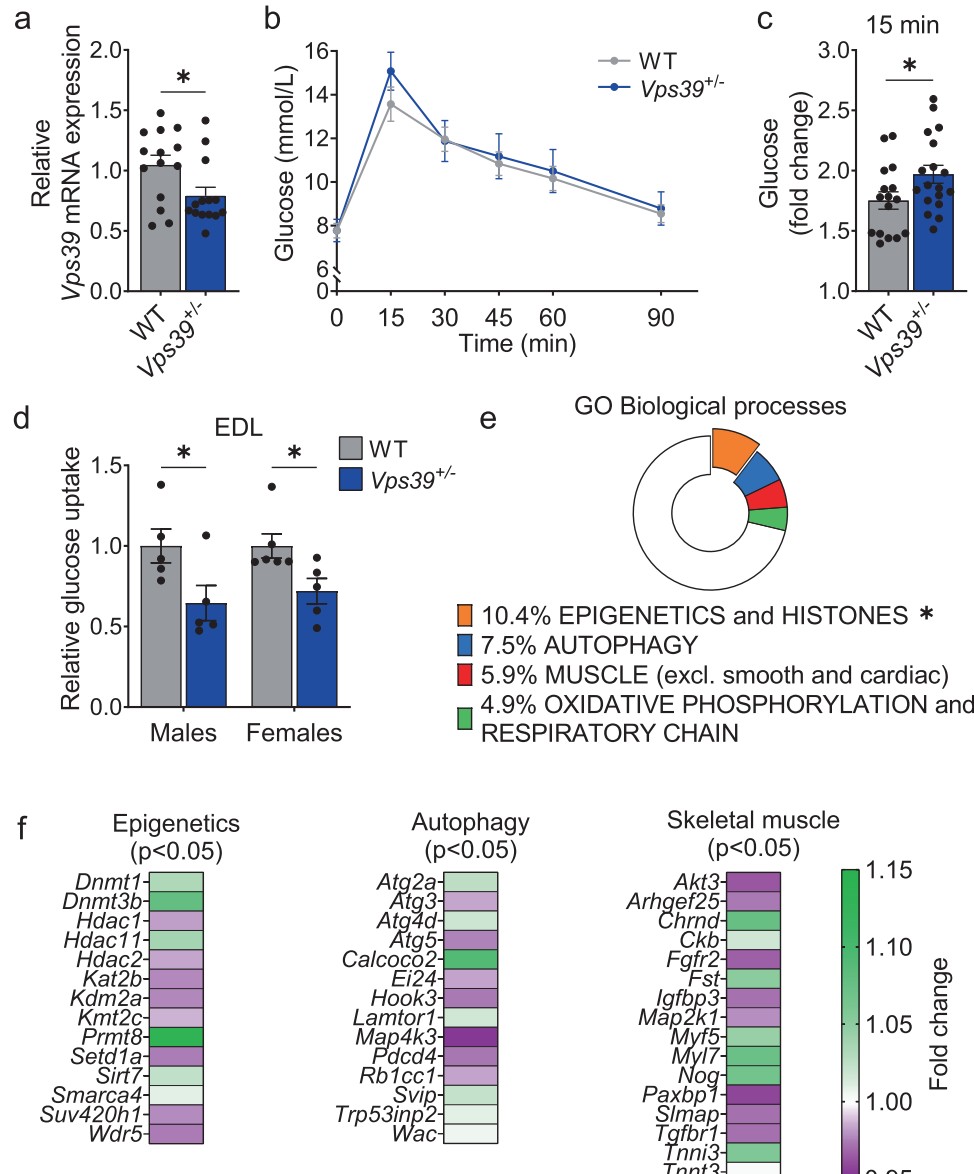

**Fig. 5 Vps39-deficiency alters glucose metabolism and gene expression related to epigenetics, autophagy, and muscle function in mouse skeletal muscle. a** mRNA expression (qPCR) of *Vps39* in skeletal muscle (tibialis anterior) from heterozygous (*Vps39*⁺/⁻, $n = 14$, including six males and eight females, blue bars) and wild type (WT, $n = 14$, including seven males and seven females, gray bars) mice. **★**$p < 0.05$ for *Vps39*⁺/⁻ vs. WT. $p = 0.0247$. **b–c** Oral glucose tolerance test (OGTT) in *Vps39*⁺/⁻ mice ($n = 18$, including ten males and eight females, blue points/bars) and WT mice ($n = 16$, including ten males and six females, gray points/bars). **b** Blood glucose levels (mmol/L) at 0–90 min during the OGTT. **c** Fold change in blood glucose levels during the first 15 min of the OGTT (glucose levels at 15 min relative 0 min). **★**$p < 0.05$ for *Vps39*⁺/⁻ vs. WT. $p = 0.0451$. **d** Relative glucose uptake in the extensor digitorum longus (EDL) muscle from *Vps39*⁺/⁻ and WT mice during 45 min after an oral glucose load. Glucose uptake in muscle was measured using 2-[1,2-³H(N)]-Deoxy-D-glucose tracer and normalized to tissue weight. Males: $n = 5$ WT and $n = 5$ *Vps39*⁺/⁻, females: $n = 6$ WT and $n = 5$ *Vps39*⁺/⁻. WT mice are set to 1. **★**$p < 0.05$ for *Vps39*⁺/⁻ vs. WT. $p = 0.0474$ (males), $p = 0.0307$ (females). **e–f** mRNA expression analysis (microarray) in skeletal muscle (tibialis anterior) from *Vps39*⁺/⁻ vs. WT mice ($n = 12$ per genotype, including six males and six females per genotype). **e** Frequency of selected GO terms ("Epigenetics and Histones" [orange], "Autophagy" [blue], "Muscle" [red] and "Oxidative phosphorylation and Respiratory chain" [green]) among the differentially expressed genes ($p < 0.05$ for *Vps39*⁺/⁻ vs. WT). Chi²-tests were used to analyze overrepresentation of differentially expressed genes belonging to a GO term compared with all analyzed genes. **★**$p$-chi² $< 0.05$ compared to all analyzed genes. **f** Heatmap showing the fold change in expression for some selected differentially expressed genes ($p < 0.05$ for *Vps39*⁺/⁻ vs. WT) related to the GO terms in (**e**) ("Epigenetics", "Autophagy" or "Muscle"), based on GO annotation or previously published research. Genes with upregulated expression in *Vps39*⁺/⁻ mice displayed in green and downregulated expression displayed in purple. For exact $p$-values see Supplementary Data 5, Sheet A. GO gene ontology. Bars (in (**a**, **c–d**)) or points (in (**b**)) represent mean values and error bars display SEM (**a–d**). Statistical significance determined by unpaired two-tailed *t*-test (**a**, **c–d**). Differential gene expression between *Vps39*⁺/⁻ and WT mice was analyzed by unpaired two-tailed *t*-tests (**e–f**).

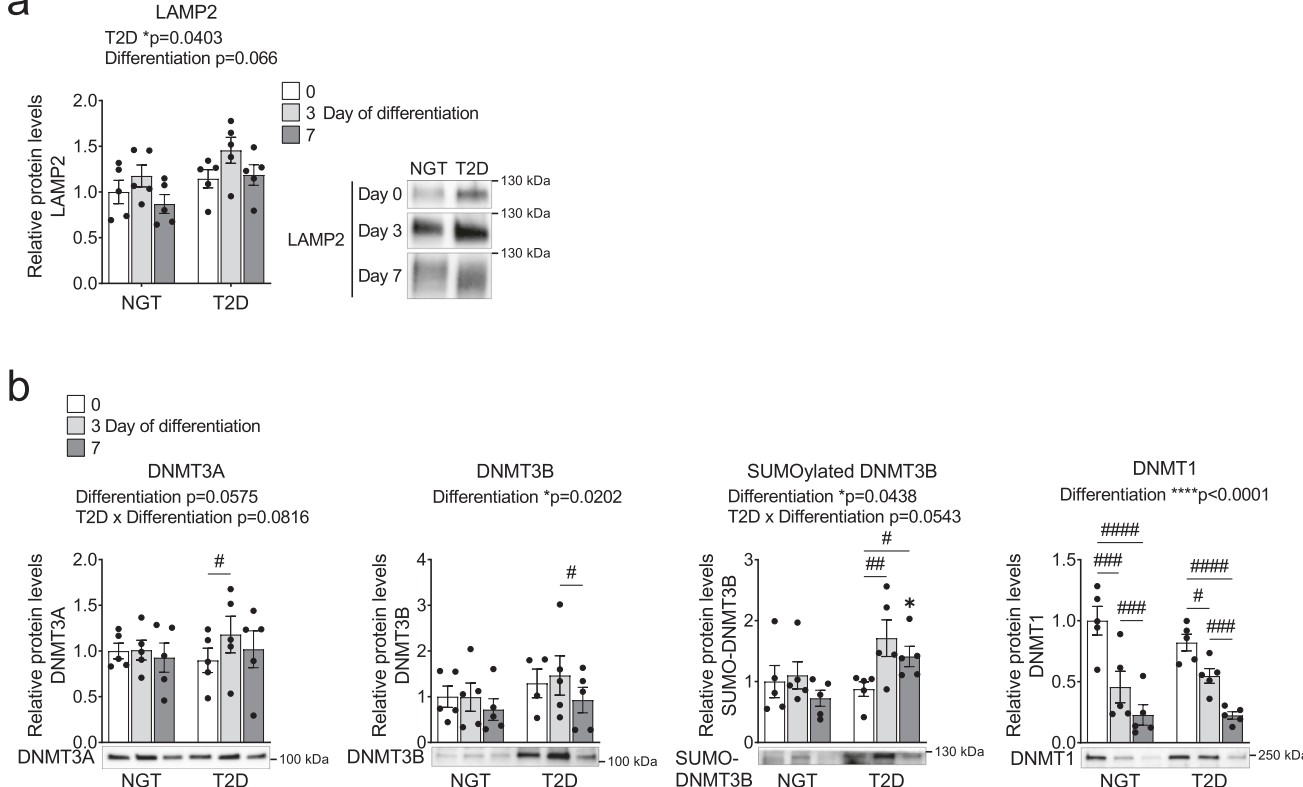

**Fig. 6 Impact of T2D on LAMP2 and DNMTs during myogenesis in human muscle cells. a–b** Protein levels (Western blot) of LAMP2 (**a**), and DNMT3A, DNMT3B, SUMOylated DNMT3B, and DNMT1 (**b**) in muscle cells from individuals with type 2 diabetes (T2D) and controls (NGT normal glucose tolerance) at day 0 (white bars), 3 (light gray bars), and 7 (dark gray bars) of differentiation. $n = 5$ individuals per group (except for DNMT3B, T2D: Day 0 where $n = 4$). NGT at day 0 is set to 1. Representative blots are shown. #$q < 0.05$, ##$q < 0.01$, ###$q < 0.001$, ####$q < 0.0001$ for comparisons between time points within each group, and *$q < 0.05$ for T2D vs. NGT at each time point. DNMT3A: $q = 0.0305$ (T2D: Day 0 vs. 3), and DNMT3B: $q = 0.0323$ (T2D: Day 3 vs. 7), and SUMO-DNMT3B: $q = 0.0073$ (T2D: Day 0 vs. 3), $q = 0.0157$ (T2D: Day 0 vs. 7), $q = 0.0237$ (Day 7: T2D vs. NGT), and DNMT1: $q = 0.0003$ (NGT: Day 0 vs. 3), $q = 0.0009$ (NGT: Day 3 vs. 7), $q = 0.0000005$ (NGT: Day 0 vs. 7), $q = 0.0457$ (T2D: Day 0 vs. 3), $q = 0.0003$ (T2D: Day 3 vs. 7), $q = 0.0000097$ (T2D: Day 0 vs. 7). Bars represent mean values and error bars display SEM (**a–b**). The effects of T2D and differentiation stated above the graphs were calculated with two-way ANOVA, or mixed-effects model (DNMT3B), with repeated measures in the factor "Differentiation" (Day 0, 3, and 7). DNMT3B, SUMOylated DNMT3B, and DNMT1 protein values were log2-transformed before statistical analysis. Non-logarithmic values are presented in the graphs. P-values were adjusted for multiple comparisons with false discovery rate (FDR) analysis (**a–b**).

but not in controls, we asked whether diabetes is associated with abnormal epigenetic changes during myoblast differentiation. To study if epigenetic changes during myogenesis are different in individuals with T2D versus controls, we separately compared DNA methylation levels at 458,475 CpG sites in myoblasts versus myotubes from 14 individuals with T2D and 14 control individuals (Fig. 7a and Table 1). We first used an unsupervised principal component analysis (PCA) to correlate the top principal components of the methylation data in myoblasts and myotubes with T2D. T2D correlated significantly with the fourth principal component in both myoblasts and myotubes ($p = 0.0001$ and $p = 0.001$, respectively), suggesting that T2D influence the DNA methylome.

We next calculated the average methylation level for all analyzed CpG sites in different gene regions, and regions based on their location in relation to CpG islands[53]. Myoblasts from individuals with T2D showed higher levels of methylation in regions distant from CpG islands (in shelves and open sea) compared to controls (Supplementary Data 6, Sheet A). Moreover, the average methylation increased significantly in the shelves and open sea during differentiation of myoblasts only in controls, while it was already high and did not increase further in myoblasts from individuals with T2D (Supplementary Data 6, Sheet A).

We then studied methylation changes of individual CpG sites during differentiation. Interestingly, DNA methylation in myoblasts versus myotubes changed significantly at twice as many individual sites in cells from individuals with T2D compared to controls (113,947 versus 49,973 CpG sites, $q < 0.05$) (Fig. 7b and Supplementary Data 6, Sheets B and C). This was in line with our recent study demonstrating that subjects with obesity had abnormal methylation changes during myogenesis compared with non-obese[18]. Many sites changed methylation only in cells from either individuals with T2D or control individuals (Fig. 7c and Supplementary Data 6, Sheets D and E), and methylation changes at 39 sites showed opposing patterns before versus after differentiation in individuals with T2D and controls (Table 2).

Together, these detected abnormal epigenetic changes during myogenesis in cells from individuals with T2D may partly be due to the impaired regulation of DNMT3A and DNMT3B, reduced VPS39 levels, and alterations in autophagy (Supplementary Fig. 3).

**Expression changes during differentiation of myoblasts into myotubes are altered in individuals with T2D.** Next, we tested whether T2D also affects the expression changes that take place

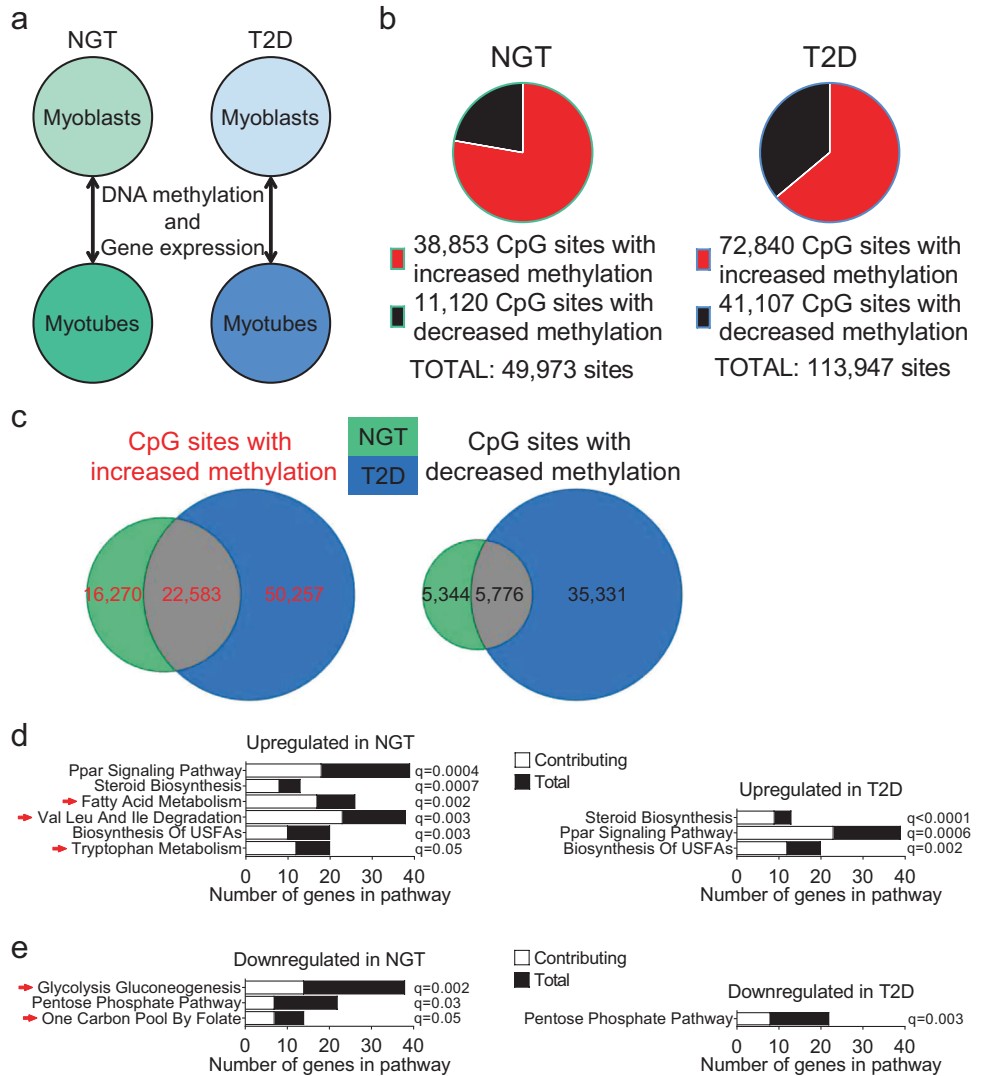

**Fig. 7 Differential changes in DNA methylation and gene expression during differentiation of human muscle cells from individuals with T2D versus controls. a** Schematic representation of the analyses performed to compare the myogenic process in cells from individuals with type 2 diabetes (T2D) and controls (NGT, normal glucose tolerance). **b** Number of CpG sites with significantly ($q < 0.05$) increased (red) or decreased (black) DNA methylation before vs. after muscle cell differentiation in individuals with NGT (left panel) and T2D (right panel) (see also Supplementary Data 6, Sheets B and C). $n = 14$ individuals per group. **c** Overlap of the CpG sites with significantly ($q < 0.05$) increased (left panel) or decreased (right panel) methylation in individuals with NGT and T2D in (**b**). **d–e** Significantly enriched gene sets (FDR < 5%) related to glucose, fatty acid, and amino acid metabolism based on gene set enrichment analysis (GSEA) of microarray expression data comparing myoblasts vs. myotubes from individuals with NGT and T2D, respectively (see also Supplementary Fig. 5f–g). Gene sets that were upregulated (**d**) or downregulated (**e**) before vs. after muscle cell differentiation. Bars represent the number of differentially expressed genes contributing to each gene set (white bars), and the total number of genes in each gene set (black bars). Red arrows indicate gene sets regulated only in individuals with NGT. $n = 13$ individuals per group. Val valine, Leu leucine, Ile isoleucine, USFA unsaturated fatty acid.

during myogenesis. We found that the expression of 7086 and 6681 genes changed in "before versus after differentiation" comparisons of muscle cells from individuals with T2D and controls, respectively (Supplementary Data 1, Sheets C and D). Most of these genes overlapped between the two groups, yet a large number changed expression in only one of the groups (Supplementary Fig. 5c). These included several genes related to AMPK/mTOR-signaling and lipid metabolism (Supplementary Fig. 5d). Interestingly, the expression of *MLST8* was regulated in the opposite direction in cells from individuals with T2D versus control individuals (Supplementary Fig. 5e). *MLST8* encodes a component of the mTOR complex 1 (mTORC1) and mTORC2, which are major regulators of autophagy[54]. We then performed a GSEA to gain biological understanding of the transcriptional

differences that occur in cells from individuals with T2D and controls during myogenesis. Many gene sets were regulated during differentiation in both groups. However, some gene sets related to amino acid and fatty acid metabolism were upregulated only in the controls (Fig. 7d and Supplementary Fig. 5f). Furthermore, gene sets related to glycolysis and one carbon pool by folate were downregulated only in controls (Fig. 7e and Supplementary Fig. 5g). A noteworthy finding is that only controls, but not individuals with T2D, showed the expected changes with increased expression of gene sets related to amino acid and fatty acid metabolism, as well as decreased expression of genes linked to glycolysis during myogenesis (Fig. 7d–e and Supplementary Fig. 5f–g). It is well established that "normal" myoblasts shift from glycolytic metabolism towards a use of amino acids and

**Table 2 CpG sites with differential DNA methylation in opposite directions before versus after differentiation of myoblasts from 14 controls (normal glucose tolerance (NGT)) and 14 individuals with type 2 diabetes (T2D).**

| | | Region in relation to | | | NGT | | | | | T2D | | | | |
|---|---|---|---|---|---|---|---|---|---|---|---|---|---|---|
| Target ID | Chromosome | Nearest gene[a] | Gene region | CpG island region | Myoblasts | Myotubes | Absolute difference (%) | p-value | q-value | Myoblasts | Myotubes | Absolute difference (%) | p-value | q-value |
| cg05133314 | 1 | *GPR153* | TSS1500 | S_Shore | 11.0 ± 1.0 | 10.0 ± 1.0 | −1.0 | 0.0040 | 0.0427 | 10.6 ± 1.3 | 12.6 ± 2.9 | 2.0 | 0.0107 | 0.0432 |
| cg09158487 | 12 | *HNRNPA1; HNRPA1L-2; CBX5* | TSS1500; TSS1500; TSS200 | Island | 8.4 ± 0.6 | 7.8 ± 0.6 | −0.6 | 0.0031 | 0.0376 | 7.5 ± 1.0 | 8.3 ± 0.6 | 0.9 | 0.0052 | 0.0267 |
| cg10245072 | 4 | *TLR6* | 3'UTR | | 87.7 ± 2.9 | 84.2 ± 3.1 | −3.5 | 0.0031 | 0.0376 | 87.5 ± 2.5 | 88.6 ± 2.3 | 1.1 | 0.0012 | 0.0110 |
| cg0389371 | 15 | *AKAP13* | 5'UTR | | 90.4 ± 3.4 | 88.8 ± 3.6 | −1.5 | 0.0040 | 0.0427 | 89.7 ± 5.6 | 91.4 ± 3.8 | 1.7 | 0.0085 | 0.0369 |
| cg14059340 | 17 | *C17orf97* | 1stExon | Island | 6.5 ± 0.7 | 5.7 ± 0.6 | −0.8 | 0.0017 | 0.0294 | 6.0 ± 0.6 | 6.7 ± 0.6 | 0.7 | 0.0031 | 0.0189 |
| cg15622783 | 6 | *NT5DC1* | TSS200 | Island | 3.2 ± 0.4 | 2.9 ± 0.3 | −0.4 | 0.0052 | 0.0482 | 3.0 ± 0.2 | 3.2 ± 0.2 | 0.2 | 0.0052 | 0.0267 |
| cg20188676 | 12 | | | | 67.5 ± 3.3 | 62.3 ± 3.9 | −5.2 | 0.0009 | 0.0229 | 67.6 ± 10.3 | 72.6 ± 9.0 | 5.0 | 0.0012 | 0.0110 |
| cg00428492 | 3 | *GTPBP8* | TSS1500 | N_Shore | 95.7 ± 0.8 | 96.5 ± 0.7 | 0.8 | 0.0031 | 0.0376 | 96.2 ± 0.8 | 95.6 ± 0.7 | −0.6 | 0.0067 | 0.0314 |
| cg00549566 | 7 | *ELMO1* | 5'UTR | Island | 6.8 ± 0.5 | 7.9 ± 0.9 | 1.1 | 0.0017 | 0.0294 | 7.5 ± 1.5 | 6.7 ± 1.2 | −0.8 | 0.0023 | 0.0159 |
| cg01195526 | 2 | *SP110* | TSS1500 | | 79.4 ± 1.6 | 83.6 ± 3.4 | 4.2 | 0.0006 | 0.0205 | 83.3 ± 2.9 | 81.4 ± 3.6 | −1.9 | 0.0067 | 0.0314 |
| cg01941585 | 12 | *HMGA2* | Body | Island | 96.1 ± 0.7 | 97.0 ± 0.7 | 0.9 | 0.0006 | 0.0205 | 95.9 ± 1.1 | 94.9 ± 2.1 | −1.1 | 0.0107 | 0.0432 |
| cg04406574 | 8 | *MYST3* | TSS1500 | S_Shore | 87.3 ± 8.5 | 90.1 ± 6.2 | 2.8 | 0.0017 | 0.0294 | 91.2 ± 2.4 | 87.8 ± 3.1 | −3.4 | 0.0031 | 0.0189 |
| cg04459360 | 6 | | | N_Shelf | 87.4 ± 4.0 | 90.0 ± 2.4 | 2.7 | 0.0009 | 0.0229 | 90.9 ± 2.9 | 88.3 ± 3.9 | −2.6 | 0.0052 | 0.0267 |
| cg04568923 | 17 | *COX10* | Body | Island | 93.1 ± 2.7 | 95.2 ± 1.3 | 2.1 | 0.0040 | 0.0427 | 95.2 ± 1.8 | 91.9 ± 5.8 | −3.2 | 0.0052 | 0.0267 |
| cg04977002 | 13 | *UPF3A; UPF3A* | 1stExon; 1stExon | Island | 5.6 ± 0.4 | 6.5 ± 0.5 | 0.9 | 0.0006 | 0.0205 | 6.2 ± 1.0 | 5.6 ± 1.0 | −0.6 | 0.0067 | 0.0314 |
| cg05076820 | 7 | *LOC100124692* | Body | Island | 24.6 ± 8.2 | 28.2 ± 8.7 | 3.5 | 0.0040 | 0.0427 | 25.5 ± 8.4 | 21.1 ± 6.0 | −4.3 | 0.0067 | 0.0314 |
| cg05221370 | 7 | *LRRN3; IMMP2L* | 5'UTR; Body | | 41.1 ± 10.0 | 46.9 ± 10.1 | 5.8 | 0.0009 | 0.0229 | 41.8 ± 12.7 | 37.9 ± 12.2 | −3.9 | 0.0085 | 0.0369 |
| cg07203362 | 2 | | | | 61.8 ± 12.2 | 68.6 ± 12.3 | 6.7 | 0.0052 | 0.0482 | 68.1 ± 19.4 | 64.4 ± 20.8 | −3.7 | 0.0085 | 0.0369 |
| cg08638395 | 1 | *XPR1* | TSS200 | Island | 3.1 ± 0.5 | 3.6 ± 0.3 | 0.5 | 0.0023 | 0.0334 | 3.4 ± 0.2 | 3.1 ± 0.2 | −0.2 | 0.0040 | 0.0225 |
| cg09180070 | 8 | | | | 72.0 ± 9.0 | 77.0 ± 6.7 | 5.0 | 0.0002 | 0.0161 | 72.2 ± 28.5 | 67.3 ± 31.5 | −4.9 | 0.0002 | 0.0049 |
| cg09291982 | 3 | *ZBTB20* | Body | Island | 96.8 ± 0.6 | 97.5 ± 0.4 | 0.7 | 0.0031 | 0.0376 | 97.8 ± 0.7 | 96.7 ± 1.4 | −1.0 | 0.0067 | 0.0314 |
| cg12076823 | 11 | *RPS6KA4* | Body | S_Shore | 83.5 ± 7.1 | 85.9 ± 6.1 | 2.4 | 0.0031 | 0.0376 | 87.2 ± 1.9 | 85.3 ± 2.4 | −1.9 | 0.0085 | 0.0369 |
| cg12891678 | 1 | | | | 40.7 ± 14.0 | 44.0 ± 12.2 | 3.3 | 0.0023 | 0.0334 | 46.0 ± 10.8 | 40.8 ± 9.7 | −5.2 | 0.0012 | 0.0110 |
| cg13044475 | 13 | *THSD1* | Body | N_Shelf | 93.0 ± 0.6 | 94.3 ± 1.1 | 1.3 | 0.0017 | 0.0294 | 94.2 ± 1.9 | 92.9 ± 1.9 | −1.3 | 0.0085 | 0.0369 |
| cg13186228 | 20 | *STX16* | TSS1500 | Island | 14.1 ± 2.8 | 17.2 ± 4.2 | 3.0 | 0.0052 | 0.0482 | 21.3 ± 14.9 | 18.4 ± 14.2 | −2.8 | 0.0067 | 0.0314 |
| cg13454978 | 8 | | | | 37.2 ± 15.6 | 41.6 ± 16.8 | 4.5 | 0.0052 | 0.0482 | 44.9 ± 27.9 | 41.9 ± 27.2 | −2.9 | 0.0052 | 0.0267 |
| cg13474639 | 9 | *MRPL50; ZNF189* | Body; TSS1500 | N_Shore | 27.9 ± 13.0 | 32.3 ± 13.1 | 4.4 | 0.0040 | 0.0427 | 30.9 ± 8.5 | 25.5 ± 5.7 | −5.4 | 0.0052 | 0.0267 |
| cg13635560 | 11 | *LRRC32; LRRC32* | 5'UTR; 1stExon | N_Shore | 13.4 ± 6.7 | 15.9 ± 7.8 | 2.5 | 0.0004 | 0.0178 | 15.4 ± 9.8 | 12.3 ± 8.3 | −3.1 | 0.0052 | 0.0267 |
| cg14299369 | 15 | *TGM5* | TSS200 | N_Shore | 84.6 ± 5.2 | 86.8 ± 5.1 | 2.2 | 0.0006 | 0.0205 | 87.7 ± 2.3 | 83.9 ± 1.9 | −3.8 | 0.0004 | 0.0059 |
| cg15935791 | 1 | *LOC100030240; RASAL2* | Body; TSS1500 | | 6.0 ± 0.9 | 6.6 ± 0.7 | 0.6 | 0.0023 | 0.0334 | 6.8 ± 0.9 | 6.0 ± 0.7 | −0.8 | 0.0107 | 0.0432 |
| cg15948245 | 7 | *DAB1* | 5'UTR | N_Shore | 67.7 ± 1.2 | 68.9 ± 1.4 | 1.1 | 0.0009 | 0.0229 | 70.2 ± 2.4 | 68.7 ± 2.0 | −1.5 | 0.0067 | 0.0314 |
| cg18863090 | 1 | | | | 31.9 ± 16.5 | 35.8 ± 17.4 | 3.9 | 0.0040 | 0.0427 | 36.3 ± 18.5 | 30.3 ± 18.7 | −6.0 | 0.0023 | 0.0159 |
| cg20574381 | 11 | *ST5* | Body | S_Shelf | 85.8 ± 4.0 | 88.3 ± 3.8 | 2.5 | 0.0017 | 0.0294 | 83.4 ± 6.8 | 79.2 ± 9.0 | −4.3 | 0.0107 | 0.0432 |
| cg20998885 | 20 | *C20orf79* | 1stExon | | 90.0 ± 2.6 | 91.5 ± 2.2 | 1.5 | 0.0012 | 0.0260 | 92.5 ± 1.4 | 90.7 ± 1.7 | −1.8 | 0.0067 | 0.0314 |
| cg21391046 | 4 | | | | 85.5 ± 7.6 | 88.8 ± 6.2 | 3.4 | 0.0009 | 0.0229 | 87.2 ± 5.8 | 84.0 ± 8.0 | −3.2 | 0.0085 | 0.0369 |
| cg21889054 | 14 | *NUMB* | 5'UTR | Island | 88.1 ± 4.3 | 91.1 ± 3.0 | 2.9 | 0.0009 | 0.0229 | 92.0 ± 3.3 | 90.2 ± 3.3 | −1.8 | 0.0031 | 0.0189 |
| cg23763197 | 11 | *H2AFX; H2AFX* | 3'UTR; 1stExon | S_Shore | 4.3 ± 0.2 | 4.9 ± 0.4 | 0.6 | 0.0017 | 0.0294 | 5.3 ± 0.8 | 4.4 ± 0.6 | −1.0 | 0.0012 | 0.0110 |
| cg27294837 | 13 | *POU4F1* | TSS1500 | | 54.6 ± 21.2 | 60.8 ± 18.7 | 6.2 | 0.0012 | 0.0260 | 56.5 ± 28.8 | 50.2 ± 29.9 | −6.3 | 0.0067 | 0.0314 |
| cg27559724 | 3 | *RFC4;RFC4* | TSS200; TSS1500 | Island | 21.4 ± 8.6 | 23.5 ± 8.7 | 2.1 | 0.0052 | 0.0482 | 23.5 ± 9.4 | 21.4 ± 9.2 | −2.0 | 0.0004 | 0.0059 |

[a]Gene names are written in italics according to conventional formatting guidelines.

fatty acids in myotubes[37] and our expression data strongly support the existence of metabolic differences during differentiation of myoblasts from individuals with T2D versus controls.

Finally, we tested if *VPS39* expression in human skeletal muscle biopsies correlates with measures of insulin sensitivity analyzed in vivo with a euglycemic hyperinsulinemic clamp, and whether the expression is reduced in biopsies from individuals with T2D versus control individuals in a cohort previously described[4]. Interestingly, *VPS39* expression in muscle biopsies correlated positively with glucose uptake (*M*-value, $r = 0.7015$, $p = 0.0052$) in individuals with T2D (Supplementary Fig. 6a), and there was a non-significant trend to reduced *VPS39* expression in muscle from individuals with T2D versus controls (Supplementary Fig. 6b).

Altogether, the human and rodent data demonstrate that VPS39 plays a key role in myogenesis and muscle glucose metabolism. We show that T2D is associated with reduced *VPS39* levels, which contribute to impaired autophagy, higher DNMT levels, and abnormal epigenetic and expression changes during myogenesis.

## Discussion

This study aimed to explore the molecular mechanisms behind the memory seen in muscle stem cells from individuals with T2D, which may contribute to an altered myogenic potential, reduced muscle function, and hyperglycemia[10,12,55]. We found lower *VPS39* expression in myoblasts and myotubes from individuals with T2D. VPS39 is a previously unrecognized regulator of human muscle regeneration and function. VPS39-deficiency in human muscle stem cells gave rise to impaired autophagy, perturbed insulin signaling, and, based on mRNA microarray data, appeared to alter the well-known metabolic switch from glycolysis in myoblasts towards fatty acid oxidation and use of amino acids in myotubes[18,37]. This resulted in abnormal regulation of epigenetic enzymes and the epigenome, and aberrant expression changes of myogenic regulators as well as gene sets related to the cell cycle, apoptosis, and glucose metabolism. In turn, myoblast differentiation was poor and apoptosis increased. Although we suggest an order of events in Supplementary Fig. 3, it is possible that the progression from VPS39-deficiency to impaired myogenesis in T2D is slightly different. Moreover, we confirmed several alterations seen in VPS39-silenced cells in both muscle cells from individuals with T2D and VPS39-deficient mice ($Vps39^{+/-}$). $Vps39^{+/-}$ mice exhibit glucose intolerance and expression changes of genes affecting epigenetics, autophagy, and metabolism in muscle. On the basis of the present data, we propose a model for T2D in which low VPS39 levels contribute to impaired muscle regeneration and insulin resistance through metabolic and epigenetic changes.

To find support for the role of VPS39 in human muscle regeneration and T2D, we silenced VPS39 in human myoblasts in vitro and in mice in vivo. First, we demonstrated that VPS39-silenced human myoblasts fail to differentiate, supporting a key role for this protein in muscle regeneration. In line with these results, our microarray data demonstrated that VPS39-silenced cells have decreased expression of key myogenic TFs and gene sets related to muscle structure. Moreover, upregulated genes support that VPS39-silenced cells do not exit the cell cycle and seem to rely on glycolysis. Indeed, it is well established that myoblasts rely glycolysis, while there is a shift from using carbohydrates towards using amino acids and fatty acids during differentiation to myotubes[18,37]. In addition, Ryall et al. showed that the metabolic shift during myogenesis is also associated with changes in epigenetic enzymes and histone modifications[37]. Interestingly, we found differential expression of numerous genes encoding epigenetic enzymes that regulate DNA methylation and

histone modifications in VPS39-silenced cells. We also found differential expression of gene sets related to autophagy, such as lysosome, and mTOR-, and p53-signaling pathways in these cells. Next, we demonstrated that functional autophagy is important for proper human myogenesis, and that autophagy is impaired in VPS39-silenced human myoblasts. To investigate which step of the autophagic process was altered, we measured autophagic flux using Baf-A1-treatment with and without starvation. VPS39-silenced cells have increased amounts of autophagy markers LC3B-II, p62, LAMP1, and LAMP2 under basal conditions, and activation of autophagy in response to starvation was impaired in these cells. These data clearly indicate an alteration in the autophagic flux after VPS39 knockdown. The lack of relative induction of LC3B-II protein levels after any of the treatments, coupled with increased p62 and LAMP1/2, indicate a late autophagy impairment, potentially by a defect in lysosomal degradation due to the increased LAMP1/2 levels already at basal conditions that increase further after Baf-A1-treatment. On the other hand, using different cells and methods, Pols et al. have previously suggested that VPS39-deficiency in HeLa cells impairs, or delays, the fusion of endosomes with endosomes or lysosomes by following BSA-gold to LAMP1-positive compartments[21]. We also found lower insulin-stimulated phosphorylation of key proteins involved in glucose uptake and metabolism, i.e., Akt and TBC1D4, in VPS39-silenced human myoblasts, suggesting an altered glucose metabolism in these cells[55,56]. We further showed that VPS39 knockdown alters HAT activity, and protein levels of numerous epigenetic enzymes including increased levels of DNMT1 and DNMT3B. It should be noted that while mRNA expression of *HAT1* was increased, the nuclear protein level was nominally decreased in VPS39-silenced cells. This is of no surprise, since inconsistencies between mRNA and protein levels are often seen[57,58]. Importantly, DNMT1 levels are known to decrease when rodent myoblasts differentiate and withdraw from the cell cycle[39]. Subsequently, we analyzed DNA methylation genome-wide in VPS39-silenced cells, and identified epigenetic alterations including differential methylation of muscle-specific genes, e.g., *MEF2*s, *MHC*s, and *MLC*s, which may contribute to the reduced expression of these genes and subsequently poor differentiation. Ultimately, knockdown of VPS39 resulted in diminished myogenesis and a higher rate of apoptosis compared to controls.

We then tested if myoblasts and myotubes from individuals with T2D resemble the phenotypes seen in VPS39-silenced muscle cells. In agreement, a marker of autophagy was altered in muscle cells from individuals with T2D. Interestingly, the expression of several genes related to autophagy and mTOR-signaling were regulated differently during myogenesis in individuals with T2D compared with controls. For example, *MLST8*, encoding a component of mTORC1 and mTORC2, which regulates lysosome function and autophagy[54], was upregulated in individuals with T2D and downregulated in control individuals during myogenesis. Autophagy is suggested to be a leading cause of reduced muscle mass and quality during aging and metabolic diseases[59,60], and the myogenic potential of satellite cells decreases with age[61]. mTOR-signaling is central for the regulation of metabolic pathways, myogenesis, and autophagy[54,62], and regulated by the availability of amino acids[63]. In addition, mTORC1-signaling is aberrant in the skeletal muscle from individuals with T2D[64,65], and the activity of the HOPS complex was recently suggested to be regulated by mTORC1[66]. Interestingly, the HOPS complex may also regulate the endocytic pathway[64]. GLUT4 is the main glucose transporter in skeletal muscle cells, and GLUT4 storage vesicles traffic in endocytic and exocytic compartments[56]. GLUT4 translocation was found to be impaired in muscle from individuals with T2D[67]. Future studies may test

whether lower levels of VPS39, a subunit of the HOPS complex, also affect the endocytic pathway, and thereby GLUT4 translocation and glucose uptake in muscle. This could be an additional mechanism contributing to the reduced glucose uptake in the muscle of individuals with T2D. In support of this, we found reduced glucose uptake in muscle from $Vps39^{+/-}$ mice, lower insulin-stimulated Akt and TBC1D4 phosphorylation in VPS39-silenced human muscle cells, as well as a positive correlation between insulin-stimulated glucose uptake and $VPS39$ expression in human skeletal muscle.

An interesting finding in our study was that gene sets related to glycolysis were downregulated, while fatty acid and amino acid metabolism gene sets were upregulated, during differentiation of myoblasts to myotubes only in controls, and not in individuals with T2D. Moreover, in accordance with the epigenetic defects seen in VPS39-silenced cells, individuals with T2D had higher DNMT levels and global DNA methylation in myoblasts, as well as aberrant remodeling of the DNA methylome during myogenesis. Notably, twice as many methylation changes occurred during differentiation of myoblasts from individuals with T2D compared to controls, based on analysis with the 450K array. Importantly, we have previously both technically and biologically validated DNA methylation data generated with the 450K array in human samples[13]. We have recently shown that obesity epigenetically reprograms muscle stem cells in a similar way[18]. This abnormal methylation pattern may be explained by the increased expression of de novo DNMTs (DNMT3A and -B) in muscle cells from individuals with T2D. In addition, the gene set "one carbon pool by folate" was downregulated only in controls, and not in individuals with T2D, during myogenesis. This pathway may also contribute to the abnormal methylation changes seen in individuals with T2D during myogenesis, since these genes encode proteins that regulate the generation of the methyldonor S-Adenosyl methionine, essential for DNA methylation. Overall, the results seen in myoblasts and myotubes from individuals with T2D resemble the data seen in VPS39-silenced cells. Our data support a model where low VPS39 levels contribute to impaired muscle regeneration in individuals with T2D.

To further study the role of VPS39 in vivo, we examined a mouse model for VPS39-deficiency ($Vps39^{+/-}$). Heterozygous mice were chosen because they have reduced, but not lacking, expression of VPS39, which resembles the situation seen in muscle cells from individuals with T2D. $Vps39$ expression was reduced by 26% in muscle from $Vps39^{+/-}$ mice, while it was reduced by 21.4% and 23.6% in myoblasts and myotubes from individuals with T2D, respectively. Indeed, $Vps39^{+/-}$ mice showed glucose intolerance and decreased glucose uptake in muscle. These mice also exhibited differential expression of genes encoding proteins that affect autophagy, epigenetic programming, muscle function, and oxidative phosphorylation. In agreement, the expression of OXPHOS genes is reduced in muscle from individuals with T2D[4]. Together, we demonstrate that $Vps39^{+/-}$ mice have metabolic phenotypes similar to a human muscular model of T2D.

T2D is a complex disease where both numerous genetic and non-genetic factors affect the pathogenesis and subsequently most likely also some of the results in the present study. We found differential expression of several genes between the T2D and control groups, and although we focused this study on dissecting the role of VPS39 in muscle cells, other genes do also contribute to the muscle dysfunction and insulin resistance seen in T2D. Moreover, to reduce the risk of random findings due to inter-individual variation we included as many individuals as technically possible in this study. We found nominally reduced VPS39 expression in muscle biopsies from individuals with T2D versus control individuals. This result could possibly be explained

by that these biopsies were taken exclusively from elderly men (≈66 years) and that muscle biopsies contain several other cell types such as endothelial cells, smooth muscle cells, immune cells, nerve cells, and fibroblasts apart from myoblasts and myotubes.

In conclusion, our data collected from human and mouse converge into a model where VPS39-deficiency leads to impaired autophagy and metabolism, epigenetic alterations, insufficient upregulation of myogenic regulators, and thereby poor myogenesis and glucose intolerance. Based on this we propose that the lower $VPS39$ expression in individuals with T2D could be one mechanism that contributes to a dysfunctional skeletal muscle phenotype and hyperglycemia.

## Methods

**Study participants**. Individuals included in this study are a subset of a previously described cohort[12,68,69]. Individuals with T2D ($n = 14$) were selected to obtain a group with a similar gender composition, age, and BMI as in the control group ($n = 14$) previously described[18] (Table 1). To determine clinical characteristics, venous blood samples were collected in fasting state following 1 week without any prescribed medication. A standardized OGTT was performed, anthropometric measurements obtained, body composition estimated by using dual-energy X-ray absorptiometry (Lunar iDXA, GE Healthcare, Madison, WI) and VO₂max assessed using a single-stage sub-maximal model test (the Aastrand test).

**Ethics statement**. The study was approved by the local ethics committee (National Committee on Health Research Ethics (DNVK) KF 01-141/04), and followed the principles of the Helsinki declaration. All study participants had provided informed written consent before any experiments.

**Isolation of primary human muscle stem cells, and myoblast culture**. After an overnight fast, human skeletal muscle biopsies were obtained under local anesthesia from the vastus lateralis muscle using a biopsy needle with suction. Muscle stem cells were isolated and cultured as described in detail previously[11,18]. Isolated myoblasts were cultured in growth medium (HAM's/F10 supplemented with 20% fetal bovine serum (FBS), 1% penicillin/streptomycin, and 1% Amphotericin B) until 70% confluent. To induce differentiation, cells were incubated in differentiation medium 1 (DMEM with 1.0 g/L glucose supplemented with 10% FBS and 1% penicillin/streptomycin) for 2 days, followed by differentiation medium 2 (DMEM with 4.5 g/L glucose supplemented with 2% horse serum and 1% penicillin/streptomycin) until cells were fully differentiated, as determined by visual confirmation of myotube formation (>3 nuclei per myotube in ~70% of cells) on day 7. The myogenic purity of a subset of cells from the cohort was previously analyzed by flow cytometry[18]. All cells were positive for CD56, which is expressed on myogenic cells, and negative for the endothelial and hematopoietic markers CD31 and CD45[70]. This indicated that pure myogenic cells have been obtained. Cells were tested negative for mycoplasma.

**Nucleic acid extraction for RNA and DNA arrays**. DNA and RNA from the cultured muscle cells were extracted using DNeasy Blood and Tissue kit (Qiagen #69504, Hilden, Germany) and Trizol (Invitrogen, ThermoFisher Scientific, Waltham, MA, USA), respectively. RNA from VPS39-silenced cells was extracted using RNeasy mini kit (Qiagen #74104) or miRNeasy mini kit (Qiagen #217004), with DNase digestion step using RNase-free DNase set (Qiagen #79254). Whole mouse tibialis anterior muscle was homogenized in TissueLyser II, and RNA extracted using RNeasy Fibrous Tissue mini kit (Qiagen #74704). RNA extractions were followed by RNeasy MinElute Cleanup kit (Qiagen #74204). The quantity and purity of nucleic acids were determined using a NanoDrop 1000 spectrophotometer (NanoDrop Technologies, Wilmington, DE, USA). RNA integrity was determined using the Bioanalyzer system (Agilent, Santa Clara, CA, USA).

**mRNA expression arrays**. mRNA expression data for myoblasts and myotubes from 13 individuals with T2D and 13 control individuals were obtained using HumanHT-12 Expression BeadChip (Illumina, San Diego, CA, USA) which targets 28,688 well-annotated transcripts. Here, we included 13 out of 14 individuals with T2D, and 13 out of 14 controls based on good quality RNA. The assay procedure followed the manufacturer's recommendations. Probes with a mean detection p-value > 0.01 for more than 60% of the samples were filtered out. Data were background-corrected, log2-transformed, quantile-normalized, and batch-corrected using COMBAT[71]. COMBAT was not used for the comparison of expression data between myoblasts and myotubes from the same individuals since these samples were placed on the same chip. The expression levels presented in tables and figures are non-logarithmic values.

mRNA expression data for VPS39-silenced myoblasts and controls ($n = 6$ per group) were obtained using GeneChip™ Human Gene 2.0 ST Assay (Applied Biosystems), and expression data for the mouse muscle ($n = 6$ per sex and genotype, for a total of 24 mice) were obtained using Clariom™ S Assay Mouse

(Applied Biosystems) according to the manufacturer's recommendations. Expression Console Software v1.4.1.46 was used for quality analyses, and data were processed using Robust Multi-array Analysis (RMA) and log2-transformed. Only probes annotated to gene symbols (31,135 probes for human cells and 22,206 for mouse muscle) were used in downstream analyses.

**DNA methylation arrays**. DNA methylation was analyzed in myoblasts and myotubes from 14 individuals with T2D and 14 control individuals using Infinium HumanMethylation450 BeadChip (Illumina). Details of this analysis are described in the Supplementary Methods.

DNA methylation was analyzed in VPS39-silenced myoblasts and controls (n = 6 per group) 3 days after transfection with siRNA and at the start of differentiation using Infinium MethylationEPIC BeadChip kit (Illumina) that covers ~850,000 CpG sites. Details of this analysis are described in the Supplementary Methods.

**Luciferase reporter assay**. Promoter fragments (1500 bp) upstream of the FBN2 or VPS39 transcription start sites (TSS) were cloned into a CpG-free luciferase reporter vector (pCpGL-basic)[72] by GenScript (GenScript USA Inc., Piscataway, NJ, USA). The resultant plasmids were in vitro methylated using the methyltransferase M.SssI (New England Biolabs, Frankfurt, Germany) (2.5 U/µg DNA), or mock-methylated (Control).

C2C12 myoblasts were cultured in 96-well plates in DMEM medium with 4.5 g/L glucose supplemented with 10% horse serum, and co-transfected with 50 ng (FBN2) or 150 ng (VPS39) methylated plasmid DNA or mock-methylated plasmids together with 4 ng Renilla luciferase control reporter vector (pRL-CMV vector; Promega, Madison, WI, USA) using FuGene HD (Promega). Luciferase activity was measured 48 h later using dual-luciferase reporter assay (E1910, Promega) and a GloMax Discover Multimode microplate reader (Promega) according to the manufacturer's instructions. Briefly, the cells were lysed in 100 µl PBL-buffer, agitated on an orbital shaker for 25 min (600 rpm). Luminescence was measured in 3.5 µl of lysate using 65 µl each of Assay Reagent II, and Stop and Glo reagent. An average firefly/Renilla signal was calculated from triplicates for each experiment and condition (three technical replicates for each experiment, and each experiment is done with n = 5–6).

**Transfection with siRNA**. Primary human myoblasts (at 70% confluence) were transfected with ON-TARGETplus human set of four siRNA molecules (Dharmacon, Lafayette, CO, USA) targeting VPS39 (LQ-014052-01), TDP1 (LQ-016112-00), MAEA (LQ-012095-00) or FBN2 (LQ-011656-00), or negative control (Non-targeting plus #D-001810-10). siRNAs were mixed with Opti-MEM reduced serum medium (Gibco, ThermoFisher Scientific, #31985-062) and Lipofectamine RNAi-MAX (Invitrogen, #13778-075) (≈0.8 µl/cm², ≈0.3% [v/v]) and incubated for 20 min at room temperature. siRNA/Lipofectamine in Opti-MEM was then diluted 1:5 (corresponding to a final concentration of 50 nM) in differentiation medium 1 without antibiotics, and added to the cells. Myoblasts were then cultured and differentiated as described above. For siRNA-mediated knockdown of DNMT3B (ON-TARGETplus SMARTpool, L-006395-00, Dharmacon) cells were harvested on day 5 of differentiation (n = 1). For these knockdown experiments, "n" is the number of independent experiments performed. Each analysis used cells from at least three non-diabetic individuals, but in some cases myoblasts from the same individual may have been used at different cell passages, or different time points. The myoblasts have been isolated from muscle biopsies of non-diabetic individuals at Lund University and Copenhagen University (n = 8; sex (male/female): 5/3; age (years ± SD): 40 ± 18; BMI (kg/m² ± SD): 23.9 ± 1.7).

**qPCR**. Extracted RNA was converted to cDNA by using QuantiTect reverse transcriptase kit (Qiagen #205311). qPCR was performed using pre-designed TaqMan gene expression assays (Applied Biosystems, ThermoFisher Scientific), or SYBRgreen primers (DNA Technology A/S Risskov, Denmark). Details of this analysis are described in the Supplementary Methods, and a list of all primers used is supplied in Supplementary Table 2.

**Fusion index**. Human myoblasts were cultured and differentiated in 6-well plates as described above. On day 7 of differentiation, myotubes were fixed with 2% paraformaldehyde for 15 min, permeabilized with 0.5% Triton X-100 and blocked with 3% BSA. Fixed cells were stained with anti-myosin antibody for 1 h at room temperature, and subsequently incubated with a secondary antibody (Supplementary Table 3) for 30 min, followed by nucleus staining with 4′,6-diamidine-2′-phenylindole dihydrochloride (DAPI). The cells were washed two-three times with PBS between all incubation steps. Stained cells were placed in Hank's Balanced Salt Solution (HBSS) and imaged within 24 h using an EVOS FL fluorescent microscope (ThermoFisher Scientific). Fusion index was calculated from six images per sample at ×10 magnification per well using pixel-based co-localization in ImageJ.

**Pathway analysis**. The GSEA preranked module[50] was used for pathway analysis of gene expression data to find enriched KEGG pathways. The expression of all analyzed transcripts on the array was ranked according to t-statistics by using t-tests to compare data for the controls versus individuals with T2D, and a paired t-

test to compare data for muscle cells before versus after differentiation. The analyses were performed with the highest occurrence for genes with multiple probes. Pathways with 1–500 transcripts were considered. FDR was used to adjust for multiple testing. Methylation data were merged with gene expression data based on gene symbol annotation. Webgestalt[73] was then used to find enriched GO biological processes among the differentially methylated and expressed genes in VPS39-silenced cells. All analyzed genes on the expression array were used as reference. FDR <5% was applied.

Calculations of GO biological processes in the mouse expression data set were based on the frequency of occurrences of the following search terms (the annotation was provided by Affymetrix): autophagy, muscle (excluding terms containing only "cardiac muscle" and "smooth muscle"), epigenetics and histones, as well as oxidative phosphorylation and respiratory chain. One gene could be annotated to more than one term and counted multiple times. Frequencies were calculated based on the total count of terms for all GO biological processes among all the analyzed genes and the differentially expressed genes (p < 0.05).

**PSCAN**. PSCAN Web Interface[28] together with JASPAR[74] were used to find enriched transcription binding motifs 0–1000 bp upstream of transcription start sites of the differentially expressed genes. PSCAN is a software tool that scans promoter sequences from co-regulated genes, looking for over-represented motifs describing the binding specificity of known TFs, thus providing quick hints on which factors could be responsible for the patterns of expression observed. FDR-corrected significance threshold was used. We highlighted TFs that were also differentially expressed in VPS39-silenced muscle cells.

**Bafilomycin A1 (Baf-A1)-treatment and autophagic flux analyses**. Primary myoblasts were treated with 100 nM Baf-A1 for 3 h per day during 4 consecutive days, starting when the culture medium was switched to differentiation medium 1. For autophagic flux analyses, primary myoblasts (day 3 of differentiation) were treated with 100 nM Baf-A1 for 3 h under both basal condition (standard culture media, differentiation medium 2), and serum- and amino acid-free starvation condition (HBSS). Control cells were treated with vehicle only (1 µl/mL DMSO). The treatments did not affect cell viability, as determined by cell number (see Supplementary Fig. 2m–n).

**Western blot**. Cells were lysed in ice-cold RIPA buffer (50 mM Tris-HCl pH 7.4–7.6, 150 mM NaCl, 2 mM ethylenediaminetetraacetic acid (EDTA), 1% Triton X-100, 0.5% Na-deoxycholate, and 0.1% sodium dodecyl sulfate (SDS)) containing protease inhibitor cocktail, phosphatase inhibitor cocktail 2 and -3 (Sigma Aldrich, St. Louis, MO, USA). For analysis of phosphorylations, cells were lysed in ice-cold lysis buffer (50 mM Tris-HCl pH 7.5, 1 mM EDTA, 1 mM EGTA, 50 mM NaF, 1 mM Na-orthovanadate, 5 mM Na-pyrophosphate, 1% [w/v] NP-40, 0.27 M sucrose) containing cOmplete protease inhibitor cocktail (Roche, 1 tablet per 50 mL), and 1 mM dithiothreitol (DTT). Nuclear content and cytoplasmic fractions were extracted using Nuclear Extraction Kit (ab113474, Abcam, Cambridge, UK). Protein concentrations were determined with the BCA method (Pierce BCA Protein Assay Kit, ThermoScientific #23225). Total protein (5–30 µg) was mixed with sample buffer (60 mM Tris-HCl pH 6.8, 2% [w/v] SDS, 10% [v/v] glycerol, 2% [v/v] 2-mercaptoethanol), separated by gel electrophoresis (Mini-PROTEAN or Criterion TGX Stain-Free Precast Gradient Gels, Bio-Rad) and transferred to LF PVDF membranes (0.45 µm, Bio-Rad). Membranes were blocked, incubated with primary antibody overnight at +4 °C, and secondary antibody for 1 h at room temperature (Supplementary Table 3). Proteins were detected with enhanced chemiluminescence in a ChemiDoc MP (Bio-Rad), and quantified based on total protein normalization with Stain-Free gels in Image Lab version 6.0.1 (V3 Western Workflow, Bio-Rad, Hercules, CA, USA). Quantifications of post-translational modifications (acetylation and phosphorylation) were instead normalized to the total levels of each corresponding protein after stripping and reprobing the membrane. Membranes probed with acetylation- or phosphorylation-specific antibodies were stripped in mild stripping buffer (15 g/L glycine, 1 g/L SDS and 1% Tween-20, pH 2.2), washed twice in PBS, and then twice in Tris-buffered saline supplemented with 0.05% [w/v] Tween-20 (TBST), followed by blocking and re-probing with primary antibody. To be able to compare unpaired samples run on separate gels and blotted to separate membranes, we included a loading reference sample in an outer lane of each gel. The quantified intensities for all samples were then normalized to the loading reference before calculating the relative values.

**High-content screening (HCS)**. Primary myoblasts were cultured in 24-well plates, and transfected with siRNA or treated with Baf-A1 (100 nM) as described above. On day 3 of differentiation, cells were fixed with 4% paraformaldehyde, permeabilized with 0.25% Triton X-100, blocked, and stained with primary antibody overnight at +4 °C, followed by incubation with a secondary antibody for 2 h at room temperature (Supplementary Table 3), and thereafter with DAPI. Target proteins were detected using spot detection application on a Cellomics ArrayScan VTI HCS Reader (ThermoFisher Scientific). First, the area (cytoplasm) surrounding DAPI stained objects (nuclei) were defined by a ring. Then, the number and area of spots for each labeled protein inside the defined rings were detected. Data from 200 fields per well, with at least 2000 nuclei, were obtained. All

conditions were performed in duplicate wells for each experiment ($n = 4$ per condition).

**Epigenetic enzyme activity assays.** Cells were seeded and transfected with siRNA as described above. Nuclear content and cytoplasmic fractions were extracted from siRNA-transfected myoblasts (day 3 of differentiation) using Nuclear Extraction kit (ab113474, Abcam) without the addition of DTT. Enzymatic activity was measured in 30 µg (HAT) and 10 µg (HDAC) nuclear extracts using HAT and HDAC Activity Assay kits, respectively (K332 and K330, BioVision, Milpitas, CA, USA) according to the manufacturer's instructions. Absorbance and fluorescence were measured on a GloMax Discover Multimode Microplate reader (Promega). All samples were analyzed in duplicates.

**Apoptosis assay.** Primary myoblasts were cultured in 96-well plates, and transfected with siRNA as described above. Apoptosis was measured at day 3 of differentiation using Apo-ONE Homogenous Caspase-3/7 assay (Promega, #G7792) according to the manufacturer's instructions. The cells were incubated with Apo-One Caspase-3/7 reagent overnight before sample fluorescence was measured. All samples were analyzed in quadruplicate.

**VPS39 heterozygous mice.** Wild-type (WT) and $Vps39^{+/-}$ mice[49], on a C57BL/6J background, were maintained under standard housing conditions with a 12-h light/dark cycle at the animal facility, Sahlgrenska Academy, University of Gothenburg. The absence of $Vps39$ expression in a homozygous embryo leads to embryonic lethality[49], and therefore WT and $Vps39^{+/-}$ male and female mice were studied. In $Vps39^{+/-}$ mice, exon 2 (bases 21–56, 36 bp) is disrupted by the insertion of a neomycin-EGFP cassette[49]. Mice had ad libitum access to water and normal chow. All experiments were performed with the permission of the Animal Ethics Committee of the University of Gothenburg (169-15), in accordance with the legal requirements of the European Community (Decree 86/609/EEC).

Genotyping was performed at weaning. Tissue samples were incubated at 55 °C overnight in DirectPCR lysis buffer (#102-T, Viagen) and Proteinase K solution (Direct PCR #25530049, 0.2 mg/mL, Invitrogen), and then centrifuged at $14,000 \times g$ for 3 min. Then, 2 µL of DNA was mixed with 18 µL mastermix (HotstarTaq MMx #1010023, Qiagen), and amplified with the following primers for WT mice: FW 5′-tcggaaggatgttggtgagt-3′ and BW 5′-ggggggtagtctttaacagaatg-3′, and for $Vps39^{+/-}$ mice: FW 5′-gccctcgatatcaagctt and BW 5′-ggggggtagtctttaacagaatg-3′[49]. An OGTT was carried out for 4-5-month old male and female mice fasted for 5 h ($n = 18$ for $Vps39^{+/-}$, and $n = 16$ for WT mice). The baseline glucose value was measured before mice received 2.5 g/kg body weight of D-glucose (25%, Sigma-Aldrich) dissolved in tap water by gavage. Blood glucose was measured after 15, 30, 45, 60, and 90 min, using a glucose analyzer (Contour XT, Bayer). A blood sample was taken after 0 and 15 min for insulin measurements, and analyzed by ELISA (Mercodia, 10-1247-01).

Tissue-specific glucose uptake was measured using 2-[1,2-$^3$H (N)]-Deoxy-D-glucose tracer (NET328A001MC, 1 mCi/mL, PerkinElmer). Male and female WT and $Vps39^{+/-}$ mice ($n = 5–6$ per group and sex) that had been fed a standard chow, were fasted for 5 h, and then received 0.25 mL tracer solution containing $^3$H-2-deoxyglucose at $23 \times 10^6$ d.p.m. $\times$ ml$^{-1}$ in 25% D-glucose by gavage. Animals were sacrificed 45 min after tracer administration. The tissue samples were weighed (~50–100 mg), and homogenized in 1 ml of 2:1 chloroform:methanol solution using TissueLyzer (Qiagen). Samples were kept at $+4$ °C overnight, then 0.5 mL 1 M CaCl$_2$ was added and the samples were centrifuged at $800 \times g$ for 20 min. The upper phase was then transferred to scintillation vials containing 5 mL Ultima Gold scintillation fluid (PerkinElmer). Plasma and total tissue $^3$H activities were determined using liquid scintillation spectrometry.

**VPS39 expression data from human muscle biopsies.** $VPS39$ mRNA expression in human muscle biopsies was analyzed by Affymetrix HG-U133A microarray in individuals with T2D and NGT individuals[4]. Muscle biopsies were taken from vastus lateralis in fasted state. Insulin sensitivity was determined with a 2-h euglycemic hyperinsulinemic clamp. The rate of glucose uptake ($M$-value) was calculated from the infusion rate of glucose, and the residual rate of endogenous glucose production measured by the tritiated glucose tracer during the clamp.

**Statistical analyses.** Data in graphs are presented as means ± SEM. Clinical characteristics between individuals with T2D and NGT were analyzed using two-tailed $t$-tests. We used linear regression analyses and adjusted for age, BMI, and sex to compare DNA methylation and mRNA expression array data between individuals with NGT and T2D (FDR below 5%, $q < 0.05$, was applied). Wilcoxon signed-rank tests were used for comparison of DNA methylation and gene expression array data between myoblasts and myotubes from the same individuals (FDR below 5%, $q < 0.05$, was applied). Distributions were analyzed with chi$^2$-tests. Paired $t$-tests were used in siRNA knockdown experiments comparing NC and si$VPS39$ cells. FDR was applied when analyzing array expression data after VPS39-silencing, except for DNA methylation data where nominal $p$-values are presented, and a chi$^2$-test was used to examine if the number of nominal differences in methylation was more than expected by chance. Protein levels throughout differentiation, or after insulin stimulation, were analyzed with repeated measures two-

way ANOVA followed by tests for each time point/treatment between groups, and between time points/treatments within each group (Fisher's LSD test). Autophagic flux data were analyzed with a repeated measures three-way ANOVA (factors "Genotype", "Starvation" and "Baf-A1"), followed by tests for each treatment between groups, and between treatments within each group. The three-way ANOVA enabled us to study the overall effects of two different treatments (starvation and Baf-A1) and genotype (si$VPS39$), as well as perform multiple comparisons between individual groups, in a single analysis. Three-way ANOVAs, and two-way ANOVAs comparing more than $2 \times 2$ groups, were adjusted for multiple comparisons (FDR below 5%, $q < 0.05$, was applied). Mouse data were analyzed using unpaired $t$-tests comparing WT and $Vps39^{+/-}$ mice. GO-terms enrichment was calculated using chi$^2$-tests. Two-sided statistical tests were used throughout, unless otherwise stated.

**Reporting summary.** Further information on research design is available in the Nature Research Reporting Summary linked to this article.

## Data availability

mRNA expression and DNA methylation data that support the findings of this study have been deposited in the NCBI Gene Expression Omnibus (GEO). mRNA expression and DNA methylation data from VPS39-silenced (si$VPS39$) and control human myoblasts are available in GEO with accession numbers GSE157345 and GSE166587, respectively. mRNA expression data from skeletal muscle from $Vps39^{+/-}$ and control mice are available in GEO with accession number GSE157342. mRNA expression and DNA methylation data from human myoblasts and myotubes, comparing individuals with T2D versus controls in either myoblasts or myotubes, are available in GEO with accession numbers GSE166467 and GSE166652, respectively. Moreover, mRNA expression and DNA methylation data from myoblasts and myotubes, comparing myoblasts versus myotubes in either individuals with T2D or controls, are available in GEO with accession numbers GSE166502 and GSE166787, respectively. Source data are provided with this paper.

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

## Acknowledgements

We thank Swegene Center for Integrative Biology at Lund University (SCIBLU) Genomics Facility, and Maria Sterner and Malin Neptin at Lund University Diabetes Center (LUDC) for help with DNA methylation and/or mRNA expression analyses, Anna Hammerberg, and the Cellomics Platform at MultiPark at Lund University for help with HCS analyses, as well as the Transgenic Core Facility at Lund University for help with embryo transfer of *Vps39*$^{+/-}$ embryos. The LAMP1 and LAMP2 antibodies, developed by August, J.T./Hildreth, J.E.K. (The Johns Hopkins University School of Medicine) was obtained from the Developmental Studies Hybridoma Bank (DSHB), created by the NICHD of the NIH and maintained at The University of Iowa, Department of Biology, Iowa City (USA). This work was supported by grants from the Swedish Foundation for Strategic Research IRC15-0067, Swedish Research Council, Region Skåne (ALF), Knut and Alice Wallenberg Foundation, Novo Nordisk Foundation, EFSD/Lilly Fellowship, Söderberg Foundation, The Swedish Diabetes Foundation, Diabetes Wellness Sweden, Påhlsson Foundation, The Royal Physiographic Society of Lund, EXODIAB (2009-1039), and Linné grant (B31 5631/2006). The Centre of Inflammation and Metabolism (CIM) is supported by a grant from the Danish National Research Foundation (DNRF55). The Centre for Physical Activity Research (CFAS) is supported by a grant from Trygfonden. Novo Nordisk Foundation Center for Basic Metabolic Research is an independent Research Center, based at the University of Copenhagen (Denmark), and partially funded by an unconditional donation from the Novo Nordisk Foundation (http://www.cbmr.ku.dk/) (Grant number NNF18CC0034900).

## Author contributions

Conceptualization, C.D., C.B., and C.L.; Methodology, C.D., J.S., A.B., C.B., E.N., E.S.V., C.S., and C.L.; Software, C.D., P.V., and A.P.; Formal analysis, C.D., J.S., and A.P.; Investigation, C.D., J.S., A.B., C.S., T.H., L.H., M.P., E.N., K.P., and Y.W.; Resources, A.B., Cha.B., O.H., J.U.W., K.P., A.V., E.S.V., K.P., C.S., and C.L.; Writing – Original draft, C. D., and C.L.; Writing, C.D., J.S., A.B., and C.L.; Review & Editing, all authors; Visualization, C.D., and J.S.; Supervision, E.S.V., K.P., C.S., and C.L.; Funding acquisition, C.D., A.B., C.B., C.S., and C.L.; Equal contributions, C.D., J.S., and A.B.

## Funding

## Competing interests

The authors declare no competing interests.
