## [Peer Review File · Nature Communications]

Reviewers' Comments:

Reviewer #1:

Remarks to the Author:

Davegardh et al describe downregulation of VPS39 in human muscle stem cell derived from type 2 diabetes (T2D) and nondiabetic persons. Using gene-silencing experiments in vitro in myoblasts and in a mouse model, they conclude that VPS39-deficiency leads to impaired autophagy, which results in altered epigenetic programming, impaired muscle regeneration and reduced glucose uptake.

In general, epigenetic muscle reprogramming has gained increasing interest over the last years. In this context, the authors identified a possibly relevant regulator of myogenesis and glucose uptake. The study includes a broad range of experiments, but several data sets do not seem necessary or relevant making the reading of the manuscript quite tedious. Thus, the manuscript would benefit from focusing on the key data and reduction of data presentation (e.g. Figure 1B,K, 2C,F,G, 3C, 5C, S2 and kinetics of HAT activity measurement in Figure 3A do not seem necessary). Moreover, the organization of the data is confusing: data from patients, cells and mice should be presented separately.

Major comments:

- 1) Overall, one key information seems missing: were the myoblast populations cultivated from one individual each and then served for individual data points or pooled (also the n per experiment varies so much that it is unclear which set of experiments is representative for a group!). There may be large intragroup variations introducing a high risk to identify differences, which are not necessarily due to a difference between diabetes vs. normal glucose tolerance status, but due to other variables, e.g. VO₂max albeit not different varies substantially within groups.
- 2) The authors claim that there has been found no role for VPS39 in muscle, while Siboni et al. Cell Rep 2015 detected VPS39 in muscle of myotonic dystrophy. Please cite and comment!
- 3) p. 7 and Figure 1 E: what was the basis for selection certain proteins for displaying the correlation between expression and methylation data, there has to be a rational, as Table S2 shows these and other data anyhow, so please clarify or delete!
- 4) The authors repeatedly use the word metabolic in the context of state, homeostasis or memory without any experimental proof in the myoblasts, these terms should therefore be deleted unless substantiated by experimental data.
- 5) The authors show that mRNA levels of HAT1 are higher in siVPS39 cells, but the protein levels are lower when compared to control. How do you explain this? HAT1 is a type of B-HAT, which are located in the cytoplasm, but HAT1 protein has been measured only in nuclear extracts and not in cytoplasmic fraction. Although p300 protein is mainly localized in the nucleus, it can be found also in the cytosol. Please evaluate protein levels in cytoplasmic fraction as well.
- 6) The study states that "acetylation of H3 could be potentially due to increased p300 and decreased HDAC5". What is the role of HAT1 and HDAC4 in H3 acetylation?
- 7) It is unclear why for the DNA methyltransferase only the protein levels are shown and data about DNMT activity (e.g. methyltransferase activity colorimetric assay) are not provided (as for HAT and HDAC).
- 8) I.315: it cannot be concluded that "all data obtained in cells from T2D individuals were in line with results for VPS39-deficient cells", since a different analysis had been performed. Indeed the authors show that siVPS39 cells have higher expression of DNMT3B compared to NC at day 3 but they don't show the protein levels in siVPS39 and NC at day 0 or 7. Instead DNMT3B levels in T2D are not significantly different from NGT at day 3, but only lower at day 7 compared to day 0. In order to conclude that data for cells from T2D are in line with data for siVPS39 cells, the authors must provide data on DNMT3B levels in VPS39-deficient cells at day 0, 3 and 7.
- 9) One key issue is the suggestion that T2D generally have lower muscle mass, which however may mainly relate to lower physical activity, this would need to be tested in the donors of the muscle biopsies.
- 10) Furthermore, the discussion is quite speculative, e. g. by stating that these data support that the switch from glycolytic fuels towards lipid oxidation during myogenesis is deficient in T2D individuals, although no flux analyses were performed. Please delete!

11) Last but not least, this reviewer did not find any proof that VPS39 expression and/or function is abnormal in skeletal muscle of humans prior to cell culturing and differentiation. Without this information, the data describe an artificial situation.

Minor comments:

- 1) l. 109, T2D have impaired glucose control always, but these patients had good to excellent glycemic control! Furthermore, the OGTT defines mainly fasting and postload glycemia not insulin resistance - so please completely rephrase this sentence or split into 2-3 correct sentences.
- 2) l. 145, what is the definition of T2D muscle pathology? I would suggest to delete this term.
- 3) Figure 3G, 3H and 3I: please provide data for male and female mice separately, as it is for figure 3J and S2.
- 4) In 95 please change "deficiencies seen in muscle" to "alterations/abnormalities seen in muscle"
- 5) For some experiments 14 individuals are used, for other 13 individuals. Please comment.
- 6) If the supplementary tables (S1, S2 and S3) show a complete list of genes, then figures 1C, 1D, 1E, 2C are unnecessary. Please remove the figures and it is enough to mention the most interesting/relevant genes (e.g. VPS39, TDP1, FBN2, etc) in the text (lines 121-122, 130)
- 7) Please specify, which primary human myoblasts have been used for the in vitro experiments? Commercial cell lines?
- 8) Please change "VPS39-deficient" into "VPS39-silenced"
- 9) How did you evaluate the nuclei size? Electron microscopy? Please comment.
- 10) In 327: Table 1 does not show any methylation data!
- 11) In 554: did you also use plasmids control (unmethylated)?
- 12) Please provide catalog numbers for each chemical used (DMEM, FBS, Pen/Strep), the components of sample buffer and in table S7 the solution used to dilute the secondary antibodies for immunohistochemistry
- 12) Labeling must be consistent in the paper (e.g. body fat-free mass or body lean mass). In Figure 1I, specify on the y axis "Relative mRNA expression VPS39". In Figure 1F: change the y axis title to "Frequency CpG sites (%)"
- 13) Is there a negative correlation between expression and methylation of CpG sites in genes with lower expression in NGT myoblast? e.g. DDX5, SOCS4, MDK, TEAD4 according to figure 1D are downregulated in NGT myoblasts, do they have a higher methylation?

Reviewer #2:

Remarks to the Author:

The manuscript by Davegardh and colleagues uses myoblast samples isolated from T2D patients and healthy individuals to determine transcriptional and epigenetic differences. While this approach is not novel (Mundry et al. 2017), the authors have identified VPS39 as a novel gene to have a role in human myogenesis. The authors found VPS39 to be downregulated in T2D, and that this downregulation affects DNA methylation. They propose a model in which VPS39 deficiency leads to alterations in autophagy, metabolism and epigenetics, which in turn trigger downregulation of myogenic regulators. Although the authors provide some compelling data, I feel the manuscript falls short in providing convincing demonstration to support their conclusions. Below major concerns:

1- Autophagy results are not convincing. Results from IF and western blot do not correlate in Baf-1 treated myoblasts. To convincingly demonstrate that there are differences in autophagy and specifically in what stage, the authors need to show data for siVPS39 and T2D myoblasts and T2D myotubes in the context of Baf-1 or another autolysosome inhibitor and assay the autophagic flux. Data showing autophagy only in the basal state does not provide enough information of impaired autophagy given that autophagy is a dynamic process. Several reviews propose the best practices to properly assess autophagy (Mizushima et al. 2010 and Klionsky et al. 2016), and the authors should take these as reference.

2- The data on LAMP1 in siVPS39 does not agree with previous report showing that VPS39 knockdown leads to decrease in LAMP-1 levels (Pols et al. 2012). The authors need to discuss this.

3- In terms of relevance, additional experiments in mice would be important to determine if autophagy is also affected in the context of VPS39 downregulation. These studies would contribute to a better understanding of the role of VPS39 in autophagy and muscle.

4- The connection between decreased VPS39 and epigenetics is not clear. The authors suggest that misregulated autophagy leads to changes in the metabolic state that in turn affect epigenetic enzymes causing impairment in myogenesis and muscle function. Although, the results support an effect on methylation upon VPS39 deficiency, current data on autophagy does not provide strong evidence that the misregulation of autophagy is the cause of changes in metabolic state.

5- To convincingly demonstrate that elevated H3ac in VPS39 deficient myoblast has a negative effect of myotube differentiation, the authors need to show H3ac levels during differentiation in control samples.

6- p300 quantification and blot do not correlate. The quantification observed in Figure 3B does not correlate with the representative image shown on Figure S1H.

Minor comments:

-Line 215: this observation is not new. The importance of autophagy in myogenesis has been previously documented in several papers.

-Line 218: The text reads: ..."the levels of lysosomal markers LAMP1 and LAMP2 were only modestly affected.", but they are not significant at the protein level, and only LAMP2 seems to show an increase by IF when analyzing relative intensity/spot area.

-Line 248: there is a typing error: "and since ta connection".

Reviewer #3:

Remarks to the Author:

In this study, Cajsa Davegårdh et al. identified VPS39 as a novel regulator of muscle regeneration and function. For T2D individuals, the expression of VPS39 was lower in myoblasts and myotubes, which led to impaired autophagy and subsequently caused abnormal epigenetic reprogramming, dysregulation of myogenic regulators and perturbed differentiation. Furthermore, the muscle of Vps39^{+/-} mouse which could mimic the situation in human with T2D displayed reduced glucose uptake and altered expression of genes regulating autophagy, epigenetic programming and myogenesis. In conclusion, the study provided interesting insights into the connection between autophagy and T2D. A few suggestions were provided to further improve the quality of this work and reliability of the observations in this study.

1. The authors used Infinium 450K BeadChips to analyze DNA methylation in myoblasts of 14 control and 14 T2D individuals and filtered 10,992 CpG sites annotated to the 577 unique genes that exhibited differential expression in myoblasts from diabetics versus controls. They studied negative correlations between DNA methylation and expression of these genes since DNA methylation is a known repressive mark. They identified 245 differentially expressed genes that displayed negative correlations between expression and methylation of one or more CpG sites. These included VPS39, TDP1, MAEA and FBN2 (Figure 1E). Considering this paper mainly studied VPS39, in order to rigorously test the results, the authors are suggested to choose another method to verify the methylation of VPS39 promoter.

2. The authors mainly used qPCR to identify the knockdown of proteins (VPS39, TDP1, MAEA and FBN2), western-blot is required to verify the level of the proteins.

3. The authors are suggested to add internal controls to all western-blot.

4. The authors are suggested to use western-blot to compare the level of TFs (MYOD1 and MEF2s) in control and T2D.

5. Figure 2J, the authors are suggested to provide a better resolution picture.
6. To test the importance of autolysosome formation during human myogenesis, the authors treated human myoblasts with Bafilomycin A1 (Baf-A1) for 3 hours per day during the first 3 days of differentiation and studied key markers of autophagosomes (LC3B), autolysosomes (p62), and lysosomes (LAMP1 and LAMP2) using both cellomics and Western blot analyses (see Figure S1D for experimental setup). Indeed, Baf-A1 impaired autophagy (Figure S1E) and the expression of myogenic markers was lower in the treated human muscle cells (Figure S1F). In order to prove the consistency of the experimental results, the authors are suggested to add the expression of myogenic markers in Figure S1E western-blot.
7. Baf A1 inhibits the fusion of autophagosomes with lysosomes, VPS39 is part of the complex mediating fusion of autophagosomes with lysosomes, so VPS39-knockdown would have a same effect as Baf A1. The authors are suggested to use cellomics and western-blot analyses (including the expression of myogenic markers) in Figure S1G.
8. Why it is described in the paper that the nuclear protein levels of p300 was increased in Figure 3B and Figure S1H, but the results showed that p300 was increased in Figure 3B and decreased in Figure S1H.
9. To verify VPS39 levels in Vps39 +/- mice to mimic the situation in human with T2D, the authors are suggested to compare the expression of VPS39 in T2D and Vps39 +/- mice.
10. The literatures are not well discussed. VPS39 participates in autophagy regulation as a component of HOPS complex, and the activity is also regulated by mTORC1 complex, which has fundamental roles in T2D. A recent study indicated that the activity of HOPS complex is regulated by mTORC1 (PMID: 30704899). In addition, HOPS complex also regulates endocytic pathway (PMID: 23645161), what could be the involvement of endocytic pathway in the connection between VPS39 and T2D?

Reviewer #4:

Remarks to the Author:

Davegardh et al studied here whether alterations in myoblasts contribute to lower muscle mass and insulin resistance in type 2 diabetes (T2D). Their major claims are that VPS39 deficiency in myoblasts impair autophagocytosis resulting in disturbed metabolic homeostasis, which in turn leads to alterations in the expression of epigenetic enzymes and thus altered epigenetic marks. Altered epigenetic programming then leads to altered expression of important transcription factors and muscle specific genes resulting in reduced myoblast differentiation and increase of apoptosis. These events contribute to the hallmarks of T2D, and suggest VPS39 as a target for T2D therapies. Demonstrating the above chain of events underlying T2D pathology in muscle cells is novel. In addition, VPS39 has not been shown before to act as an important regulator of human myogenesis.

Overall, this study is on an important and timely health topic, T2D, prevalence of which is increasing in the population. Thus the results from this study will be of interest for a broad audience from basic to clinical researchers. This study includes a large number of experiments, each resulting in a hypothesis of which the next experiment aims to test, all logically following each other, and bringing evidence together to support the full story. The manuscript is very well written and easy to follow. However, there are several issues that I feel require further clarification from the authors.

The whole story this manuscript tells is logical, however, I wonder if the authors had a prior hypothesis, which does not come up in the manuscript? Or how the authors happened to pick a gene, VPS39 (the top 92nd gene, plus the other 5 they mention), among all the 577 genes that were differentially expressed (DEG) in the myoblasts of T2D vs controls? What was the rationale for highlighting these genes already from the very beginning?

Gene expression analyses were logically followed with DNA methylation profiling of the 577 DEG in

the same cells, and the authors searched for negative correlations between expression and methylation. As it has been shown for numerous times, also by these authors, the impact of DNA methylation on gene expression is far from being straight forward. DNA methylation may increase or decrease transcription, depending for example on the genomic context. Therefore it is surprising that the authors decided to only study negative correlations between methylation and gene expression. Could the authors please clarify the basis for this choice?

The correlation analyses come with multiple testing burden, but the authors have only reported nominal p-values. Could the authors please adjust the p-values accordingly, or justify the use of nominal p-values if they disagree?

The negative correlations were observed for 245 transcripts, and the authors highlight again the same 4 genes as before, although many more were identified and a high number of other genes showed stronger correlations. This fits very nicely to the story, but the rationale for picking these 4 genes is not clear at this point. Could the authors please elaborate on this?

Next the authors explored whether the same myoblast DEGs are also DEGs in myotubes of T2D vs controls, and identified 42 DEGs, of which 20 were the same as in myoblasts. These include also the 4 genes that have been highlighted from the beginning of this manuscript. The authors conclude that these genes likely have a role in T2D muscle pathology, and selected "the 4 genes" for functional experiments. Here they aimed to find novel muscle regulators and selected genes based on the following criteria: reduced expression in both myoblasts and myotubes in T2D, inverse correlation between expression and methylation, and known functions in other cell types suggesting a potential impact on myogenesis. How many genes fulfilled these criteria?

The authors showed that silencing of VSP39 resulted in lack of myotube formation and concluded that VPS39 is a putative regulator of myoblast differentiation, hypothesis of which they went to explore further by measuring protein levels, gene expression, DNA methylation and cell physiology in VPS39 silenced vs control myoblasts differentiating into myotubes (n=6 per group). They identified 2635 DEGs including myogenic transcription factors and muscle specific genes downregulated, DNA replication, cell cycle and autophagocytosis related gene sets upregulated, and various epigenetic enzymes showing differential expression in VSP39-deficient vs control cells in early differentiation. These epigenetic enzymes that were identified here, were there more of them identified than what would be expected by chance?

PSCAN and JASPAR were used to search for transcription factor binding motifs in the DEG identified in the VPS39 deficient vs control cells. Could the authors please describe how these tools were used in the methods section of the manuscript? This would help in interpreting the predictions. How are the Z-scores and the associated p- or q-values interpreted? These prediction tools identified 97 motifs, and the authors decided to highlight the motif at top 68th position, i.e. myogenin. Rationale for this over the more significant motifs?

The authors tested if autolysosome formation is important in myogenesis and state on line 207 that this has not been studied before in muscle cells. I suppose the authors refer here to human myogenesis, as it has been studied at least in rodents (reference 37) and drosophila (Fujita et al eLife. 2017; 6: e23367, doi: 10.7554/eLife.23367). Please correct accordingly.

As a number of epigenetic enzyme gene and protein expression as well as their activity was altered in VPS39-defective cells the authors assessed DNA methylation in these cells vs controls (n=6). They observed 5045 CpG sites with differential methylation with nominal $p < 0.05$, of which almost 2/3 annotated to the DEGs observed in the same cell comparisons. Here the authors consider $p < 0.05$ as significant, although in the methods it is stated the FDR $q < 0.05$ was considered as significant. Table S4 shown that the differentially methylated genes highlighted in the manuscript body text show only tiny if any difference in the mean methylation of the CpG sites (range for all CpGs 0.4-10%, but for the highlighted genes e.g. MEF2s 1-4% difference in means,

i.e these are not within the top differentially methylated CpGs). What is the authors' argument for the p-value cut off for significance here?

I also suggest the authors mark in Table S4 those genes they refer to as "Specific genes of importance" in the body text on page 11, line 239.

The authors conclude that their data support a model where low levels of VPS39 in the myoblasts impair autophagy, which results in disturbed metabolic homeostasis and is followed by alterations of epigenetic enzymes and the epigenome. This results into lower expression of key myogenic regulators and muscle specific genes, and reduced proportion of myoblasts differentiating to myotubes and increase in the rate of apoptosis. As the manuscript is very long, it would be helpful if the authors could add the main findings supporting this chain of events into Figure 4.

The mouse model seems to mimic well human T2D in regards to glucose intolerance and reduced glucose uptake in their muscle tissue. The findings also nicely supports the results in human cells. However, the significance of the expression findings are based on $p < 0.05$, not $q < 0.05$, and it looks like none of the findings in the human sample could be replicated if the p-values were corrected for multiple testing. It is also not clear how to interpret the Table S5 sheet B listing gene ontologies. The authors claim that the analysis revealed an overrepresentation of genes associated with epigenetics and histones, and a nominal significant enrichment of genes associated with muscle. What is this statement based on? How was the overrepresentation calculated? There are some highlighted GOs with 0 or 1 differentially expressed genes (with $p < 0.05$). How should one interpret this table? Also, the GOs highlighted in the text are not among the top DEGs or GOs, based on Table S5. Can the authors please clarify these issues?

After examining the mouse model, the authors went back to the T2D individuals and identified DNA methylation differences in myoblasts vs myotubes in 14 T2D patients vs 14 controls. The authors report T2D myoblasts having higher methylation in certain gene regions with $q < 0.07$. This is again slightly confusing as the significance threshold is again different. Why is this?

Also, on page 15, line 336: "in these regions" refer to the regions that show higher mean methylation in myoblasts of T2D compared with controls, and when looking at Table S6 sheet A, this is not true. Please match the text with the results shown in Table S6 sheet A (both controls and T2D increase in some gene regions, and also in CpG island regions).

The number of differentially methylated CpGs in myoblast vs myotube and T2D vs controls are huge, and it is difficult to draw any other conclusions but that the methylation alterations were broader in T2D vs controls. The authors however report 39 CpG sites that showed opposite methylation change during myogenesis of T2D vs controls. It would have been interesting to see whether there was any overrepresented GO terms these genes belong to, which could give a hint on why they were with opposite methylation patterns.

In the methods statistics -section it is stated for all the analyses with multiple testing the significance is based on FDR $q < 0.05$, but in certain tables it is FDR $q < 0.07$ or even $p < 0.05$ which was considered as significant (same in the text referring to these results). Could the authors please give a rationale for these inconsistencies?

All in all, the level of detail provided in this manuscript regarding the methodology is appropriate and will enable other researchers to reproduce their work, and the statistical analyses performed were valid, to the best of my understanding.

Finally, I fully agree with the authors that the pathway from impaired expression of VPS39 to low muscle mass and insulin resistance in T2D makes sense. However, have the authors considered any other alternative routes their results would also support? How confident can the authors be for the causality of events they propose here? Which results support the claim that impaired

autophagy resulted in altered metabolic state? And that this disturbed metabolic state would result in expression changes in the epigenetic enzymes and epigenetic makeup of the VPS39 deficient cells, or T2D muscle cells? Which occurred first epigenetic or gene expression changes, any support for either? There were long lists of differentially expressed genes and differentially methylated CpG sites identified in myoblasts vs myotubes, and T2DM vs controls as well as during myogenesis in VPS39 deficient vs control cells. There may be other highly important genes than those highlighted throughout the manuscript among these. The authors should discuss the potential other conclusions on the chain of events their data support, as well as mention clearly the reasons for only following up certain genes (even though they were not the top significant hits) and ignoring the rest (excluding some GSEA/GO analyses these genes contributed to).

Response to Reviewers' comments NCOMMS-19-27809-T:

“Reduced VPS39 observed in type 2 diabetes impairs autophagy, epigenetics and muscle stem cell differentiation”

Reviewer #1 (Remarks to the Author):

Davegardh et al describe downregulation of VPS39 in human muscle stem cell derived from type 2 diabetes (T2D) and nondiabetic persons. Using gene-silencing experiments in vitro in myoblasts and in a mouse model, they conclude that VPS39-deficiency leads to impaired autophagy, which results in altered epigenetic programming, impaired muscle regeneration and reduced glucose uptake. In general, epigenetic muscle reprogramming has gained increasing interest over the last years. In this context, the authors identified a possibly relevant regulator of myogenesis and glucose uptake. The study includes a broad range of experiments, but several data sets do not seem necessary or relevant making the reading of the manuscript quite tedious. Thus, the manuscript would benefit from focusing on the key data and reduction of data presentation (e.g. Figure 1B,K, 2C,F,G, 3C, 5C, S2 and kinetics of HAT activity measurement in Figure 3A do not seem necessary). Moreover, the organization of the data is confusing: data from patients, cells and mice should be presented separately.

RESPONSE: We appreciate the valuable comments from reviewer 1 that have helped to improve our manuscript and we were happy that he/she believes we identified a possibly relevant regulator of myogenesis and glucose uptake. To improve the organization, we have separated figures and deleted Figure 1E. However, based on comments from the other reviewers, we had to add data to the revised ms. Moreover, before submitting the ms, the Editor at ELEVATE SCIENTIFIC proof read the text to make sure the content and structure of the ms was adequate and during the revision we have tried to further improve the ms.

Major comments:

1) Overall, one key information seems missing: were the myoblast populations cultivated from one individual each and then served for individual data points or pooled (also the n per experiment varies so much that it is unclear which set of experiments is representative for a group!). There may be large intragroup variations introducing a high risk to identify differences, which are not necessarily due to a difference between diabetes vs. normal glucose tolerance status, but due to other variables, e.g. VO₂max albeit not different varies substantially within groups.

RESPONSE: We apologize that this was unclear. Each “n” is from one individual when we compare controls and diabetics, or one animal in all animal experiments. For example, when n=14, we study myoblasts from 14 different people individually. We aimed to include as many samples/people as possible in each experiment, but since human primary myoblasts are a limited resource we had to reduce the number of human samples for the siRNA experiments. For knockdown experiments in human myoblasts, “n” is the number of experiments performed; each experiment used cells from at least 3 different individuals, but in some cases myoblasts from the same individual may have been used at different cell passages or different time points when n>3. Samples have never been pooled. Based on this comment, throughout the ms we have tried to clarify whether the number of samples is from different individuals or different experiments.

We agree that when human samples are studied, differences between people (for example genetic and non-genetic factors) may affect the results. Therefore, we aimed to keep the number of samples as big as possible, and we included both humans and a mouse model in the paper. Based on this comment, we added a discussion on page 23-24 of the revised ms.

“T2D is a complex disease where both numerous genetic and non-genetic factors affect the pathogenesis and subsequently most likely also some of the results in the present study. We found

differential expression of several genes between the diabetic and non-diabetic groups, and although we focused this study on dissecting the role of VPS39 in muscle cells, other genes do also contribute muscle dysfunction and insulin resistance seen in T2D. Moreover, to reduce the risk of random findings due to individual variation between people we included as many individuals as technically possible in this study.”

2) The authors claim that there has been found no role for VPS39 in muscle, while Siboni et al. Cell Rep 2015 detected VPS39 in muscle of myotonic dystrophy. Please cite and comment!

RESPONSE: We appreciate that the reviewer found this paper that has identified mis-splicing of Vps39 expression in a mouse model of myotonic dystrophy. However, we believe that our statement that “VPS39 is a previously unrecognized regulator of human myogenesis” is still correct. We now cite this paper on pages 7 and 9 of the revised ms and we also added “human” on pages 7, 9, 11, 20, 21.

Page 9: *“In other tissues, VPS39 is part of the complex that mediates fusion of autophagosomes with lysosomes (HOPS complex) and it has been found in mouse models of myotonic dystrophy type 1^{20,21,25}.”*

3) p. 7 and Figure 1 E: what was the basis for selection certain proteins for displaying the correlation between expression and methylation data, there has to be a rational, as Table S2 shows these and other data anyhow, so please clarify or delete!

RESPONSE: We describe on page 7 and 8 why we selected these genes.

“These included several genes that had not previously been studied in human myoblasts but with identified functions in other cell types or species that suggest that they may also have a role in human muscle cells, e.g. VPS39, TDP1, and MAEA¹⁹⁻²⁵, as well as genes previously implicated in muscle regeneration, e.g. FBN2, TEAD4 and STAT3^{26,27} (Figure 1c-d).”

“To identify new regulators of muscle regeneration and function, we asked whether any of the genes with reduced expression in both myoblasts and myotubes from T2D versus controls, and with inverse correlation between their expression and DNA methylation, also have a functional role in human myogenesis. We mainly focused on genes that had not been previously studied in human muscle cells, but with known functions in other cell types or species that would suggest an impact also on myogenesis¹⁹⁻²⁶. Based on these criteria, we selected four genes (VPS39, TDP1, MAEA and FBN2) for functional follow-up experiments (Figure 1b, 1g, Figure S1a-c and Table S2).”

But based on this comment, we have deleted Figure 1E in the revised ms.

4) The authors repeatedly use the word metabolic in the context of state, homeostasis or memory without any experimental proof in the myoblasts, these terms should therefore be deleted unless substantiated by experimental data.

RESPONSE: We appreciate this comment and have altered this phrase on several pages of the revised ms. We also performed additional experiments to explore whether VPS39-silenced human muscle cells have altered activation of key proteins involved in metabolic pathways i.e. glucose uptake and glycogen synthesis. These data have been included on page 12-13 of the revised ms:

“The autophagic process has been linked to metabolic remodeling and myoblast differentiation in rodents^{31,34}. We proceeded to examine whether VPS39-silenced human muscle cells have altered activation of key proteins involved in metabolic pathways i.e. glucose uptake and glycogen synthesis. VPS39-silenced cells exhibited decreased insulin-induced Akt phosphorylation, at both serine 473 (Ser473) and threonine 308 (Thr308) compared to control cells (Figure 4a). The phosphorylation of downstream Akt substrates TBC1D4 that controls GLUT4 translocation³⁵, and glycogen synthase

kinase 3 (GSK3) - α that regulates glycogen synthase activity³⁶, were both decreased in insulin-stimulated VPS39-silenced cells versus control (Figure 4b-c). Phosphorylation of TBC1D4 is associated with increased glucose uptake and phosphorylation of GSK3 is associated with activation of glycogen synthesis. These results suggest that the overall activity of the insulin signaling pathway is perturbed in VPS39-silenced myoblasts, and that some metabolic pathways may be altered.”

5) The authors show that mRNA levels of HAT1 are higher in siVPS39 cells, but the protein levels are lower when compared to control. How do you explain this? HAT1 is a type of B-HAT, which are located in the cytoplasm, but HAT1 protein has been measured only in nuclear extracts and not in cytoplasmic fraction. Although p300 protein is mainly localized in the nucleus, it can be found also in the cytosol. Please evaluate protein levels in cytoplasmic fraction as well.

RESPONSE: This is a valid comment. Protein levels do not always correspond/correlate to the mRNA levels and we do not find it strange that HAT1 mRNA expression is higher while nuclear protein levels are lower in siVPS39 cells. However, based on this comment, we have now measured HAT1 and p300 protein levels in the cytosolic fraction and found no significant differences between the siVPS39 and control cells. These data are added to **Figure 4e** and we revised the text on page 13.

“Additionally, the nuclear protein levels of HAT1 were nominally reduced and p300 were increased, while no differences were found in the cytoplasm (Figure 4e).”

We also discuss the difference between mRNA and protein levels on page 21 of the revised ms.

“It should be noted that while mRNA expression of HAT1 was increased, the nuclear protein level was nominally decreased in VPS39-silenced cells. This is of no surprise, since inconsistencies between mRNA and protein levels are often seen^{57,58}.”

6) The study states that “acetylation of H3 could be potentially due to increased p300 and decreased HDAC5”. What is the role of HAT1 and HDAC4 in H3 acetylation?

RESPONSE: HAT1 is involved in the rapid acetylation of newly synthesized cytoplasmic histones, which are in turn imported into the nucleus for *de novo* deposition onto nascent DNA chains. Specifically, HAT1 can acetylate soluble but not nucleosomal histone H4 at lysines 5 and 12, and to a lesser degree, histone H2A at lysine 5. The HAT1 function was recently described by Gruber et al (PMID:31278053). Although HAT1 does not seem to be important for H3 acetylation, some studies suggest it might and we cannot rule out that it may affect H3.

HDAC4 is categorized a class IIa histone deacetylase, and contributes to the regulation of histone H3 acetylation status together with other HDACs (PMID: 24579951). In order to be enzymatically active, HDAC4 requires association with other factors such as SMRT/NCoR and HDAC3 (PMID: 11804585), hence the individual contribution of HDAC4 to deacetylation of H3 is likely minor. HDAC4 activity is also regulated by its subcellular localization by binding to 14-3-3 proteins (PMID: 10869435). In immature muscle cells, HDAC4 interacts with the transcription factor MEF2 to repress transcriptional activity of muscle-specific genes (PMID: 10983972, PMID: 10487761). During differentiation, HDAC4 is sequestered in the cytoplasm resulting in increased histone acetylation and increased MEF2-dependent transcription (PMID: 10737771). Additionally, HDAC4 plays a role in histone methylation by forming complexes with heterochromatin protein 1 (HP1) that recruits histone methyltransferases to methylated histone lysines, to provide an efficient epigenetic mechanism for coupling histone deacetylation and methylation (PMID: 12242305). Therefore, HDAC4 may also affect H3 but most likely in combination with other proteins.

7) It is unclear why for the DNA methyltransferase only the protein levels are shown and data about DNMT activity (e.g. methyltransferase activity colorimetric assay) are not provided (as for HAT and HDAC).

RESPONSE: Our lab has tried to set up DNMT activity assays since 2010. We have used commercial assays and spoken to both the companies and different research groups but none of the assays we have tried to develop/use have generated reliable results. Three different people in my group have worked on this over the years and we have tried to analyze DNMT activity in both pancreatic cells and muscle cells without success. Hence, we do not find the DNMT activity assays reliable and did therefore not use it for this study.

8) I.315: it cannot be concluded that “all data obtained in cells from T2D individuals were in line with results for VPS39-deficient cells”, since a different analysis had been performed.

Indeed the authors show that siVPS39 cells have higher expression of DNMT3B compared to NC at day 3 but they don't show the protein levels in siVPS39 and NC at day 0 or 7. Instead DNMT3B levels in T2D are not significantly different from NGT at day 3, but only lower at day 7 compared to day 0.

In order to conclude that data for cells from T2D are in line with data for siVPS39 cells, the authors must provide data on DNMT3B levels in VPS39-deficient cells at day 0, 3 and 7.

RESPONSE: We appreciate this comment and have now changed the sentence on page 17:
“In general, although different methods have been used, several of the data for cells from T2D individuals were similar to our results for VPS39-silenced human myoblasts and mouse, suggesting a model whereby reduced VPS39 levels alter autophagy, epigenetic enzymes, thereby negatively affecting myogenesis and muscle function (Figure S3).”

Moreover, based on this review comment, have analyzed both DNMT3B as well as DNMT1 protein levels in VPS39-deficient cells at day 0, 3 and 7 and included these data in **Figure S5b** and added the following text on page 17:

“In agreement with T2D cells, both DNMT3B and DNMT1 levels were decreased at day 7 versus day 3 in VPS39-silenced cells (Figure 6b and S5b).”

9) One key issue is the suggestion that T2D generally have lower muscle mass, which however may mainly relate to lower physical activity, this would need to be tested in the donors of the muscle biopsies.

RESPONSE: We refer in the introduction to a paper by Parker et al (PMID:16731847), where they have studied muscle strength, mass, and quality in subjects with T2D and controls. Parker et al found reduced muscle strength and quality (strength divided by mass) in subjects with T2D compared with the controls. We do not have the possibility to study these functions in our cohort since it was collected approximately 10 years ago. However, based on this comment we made a change to the abstract and the introduction.

“Low muscle quality and insulin resistance are hallmarks of type 2 diabetes (T2D).”

“Skeletal muscle is the primary organ responsible for insulin-stimulated glucose uptake and T2D is associated with lower muscle strength and quality (strength divided by mass) contributing to glucose intolerance^{1,2}.”

10) Furthermore, the discussion is quite speculative, e. g. by stating that these data support that the switch from glycolytic fuels towards lipid oxidation during myogenesis is deficient in T2D individuals, although no flux analyses were performed. Please delete!

RESPONSE: We have now deleted this sentence in the discussion.

11) Last but not least, this reviewer did not find any proof that VPS39 expression and/or function is abnormal in skeletal muscle of humans prior to cell culturing and differentiation. Without this information, the data describe an artificial situation.

RESPONSE: Our goal was to identify genes with altered expression in myoblasts from T2D versus control individuals. However, based on this question we now also tested if *VPS39* expression is altered in skeletal muscle biopsies from T2D versus control individuals using available expression data in our group (Mootha et al Nature Genetics, PMID:12808457). We have added these data on page 19 and added methods on page 33-34 of the revised ms.

“Finally, we tested if VPS39 expression in human skeletal muscle biopsies correlates with measures of insulin sensitivity analyzed in vivo with a euglycemic hyperinsulinemic clamp, and whether the expression is reduced in biopsies from T2D versus control individuals in a cohort previously described⁴. Interestingly, VPS39 expression in muscle biopsies correlated positively with glucose uptake (M-value, $r=0.64$, $p=0.017$) in diabetics (Figure S6a) and there was a non-significant trend to reduced VPS39 expression in muscle from individuals with T2D versus control (Figure S6b).”

“VPS39 expression data from human muscle biopsies

VPS39 mRNA expression in human muscle biopsies was analyzed by microarray in T2D and NGT individuals as previously described⁴. Muscle biopsies were taken from vastus lateralis in fasted state. Insulin sensitivity was determined with a 2-h euglycemic hyperinsulinemic clamp. The rate of glucose uptake (M-value) was calculated from the infusion rate of glucose and the residual rate of endogenous glucose production measured by the tritiated glucose tracer during the clamp.”

Minor comments:

1) l. 109, T2D have impaired glucose control always, but these patients had good to excellent glycemic control! Furthermore, the OGTT defines mainly fasting and postload glycemia not insulin resistance - so please completely rephrase this sentence or split into 2-3 correct sentences.

RESPONSE: We have now deleted “insulin resistant” from this sentence.

2) l. 145, what is the definition of T2D muscle pathology? I would suggest to delete this term.

RESPONSE: In the revised ms we have deleted this term.

3) Figure 3G, 3H and 3I: please provide data for male and female mice separately, as it is for figure 3J and S2.

RESPONSE: Unfortunately, and most likely due to power issues, for some of the phenotypes we only found significant differences when we combined data from both male and female mice. However, there was a similar trend when splitting the groups based on sex. Therefore, we would like to keep the figures as they are in the main part of the manuscript. However, based on this comment, we modified the legend to clarify how many of each sex were included in each figure.

4) In 95 please change “deficiencies seen in muscle” to “alterations/abnormalities seen in muscle”

RESPONSE: We have now changed this to abnormalities.

5) For some experiments 14 individuals are used, for other 13 individuals. Please comment.

RESPONSE: Muscle cells from 14 T2D and 14 controls were included in this study. However, we only managed to generate good quality expression data from 13 T2D and 13 controls. Based on this comment, we added the following sentence on page 26:

“Here, we included 13 out of 14 diabetics and 13 out of 14 controls based on good quality RNA.”

6) If the supplementary tables (S1, S2 and S3) show a complete list of genes, then figures 1C, 1D, 1E, 2C are unnecessary. Please remove the figures and it is enough to mention the most interesting/relevant genes (e.g. VPS39, TDP1, FBN2, etc) in the text (lines 121-122, 130)

RESPONSE: We have deleted Figure 1E based on this suggestion. However, we prefer to keep the other figures in the main part of the manuscript to make it easier for the reader. We are willing to change this if this reviewer finds this very important.

7) Please specify, which primary human myoblasts have been used for the in vitro experiments? Commercial cell lines?

RESPONSE: Primary human myoblasts used for siRNA experiments and other functional experiments were from non-diabetic individuals. We have added this information on page 28 of the revised ms:

“For these knockdown experiments, “n” is the number of experiments performed; each experiment used cells from at least 3 non-diabetic individuals, but in some cases myoblasts from the same individual may have been used at different cell passages or different time points. The myoblasts have been isolated from muscle biopsies of non-diabetic individuals at Lund University and Copenhagen University (n=8; sex (male/female): 5/3; age (years±SD): 40±18; BMI (kg/m²±SD): 23.9±1.7).”

8) Please change “VPS39-deficient” into “VPS39-silenced”

RESPONSE: We have made this change throughout the revised ms.

9) How did you evaluate the nuclei size? Electron microscopy? Please comment.

RESPONSE: The nuclei size was evaluated based on the area of the DAPI-stain. We first identified the nuclei by intensity and area and morphology than verified which were true Dapi+ nuclei. High content screening analyses gave us the opportunity to analyze many cells in a completely unbiased matter. We describe this in the legend to Figure 3 and in the methods on page 31-32.

”High-content screening (HCS)

Cells were seeded in 24-well plates and transfected or treated with Baf-A1 (100 nM) as described above. On day 3 of differentiation, cells were fixed with 4 % paraformaldehyde, permeabilized with 0.25 % TritonX-100, blocked, and stained with primary antibody (Table S7) overnight at +4°C, followed by incubation with a secondary antibody for 2 hours at room temperature, and thereafter with DAPI. Target proteins were detected using spot detection application on a Cellomics ArrayScan VTI HCS Reader (ThermoFisher Scientific). First, the area (cytoplasm) surrounding DAPI stained objects (nuclei) were defined by a ring. Then, number and size of spots for each protein inside defined rings were detected. Data from 200 fields per well, with at least 2000 nuclei, were obtained. The area (cytoplasm) surrounding DAPI stained objects (nuclei) was analyzed using a Cellomics ArrayScan VTI HCS reader (ThermoFisher Scientific). The number and size of detected spots were collected for each target protein. All conditions were performed in duplicate wells for each experiment (n=4 per condition).”

10) In 327: Table 1 does not show any methylation data!

RESPONSE: Here, Table 1 refers to the individuals who we measured DNA methylation in.

11) In 554: did you also use plasmids control (unmethylated)?

RESPONSE: Yes, we used mock-methylated plasmids as control see Figure 1f, the methods on page 27-28 and the legend to Figure 1f. We apologize that this was unclear and have tried to clarify it in the revised ms.

12) Please provide catalog numbers for each chemical used (DMEM, FBS, Pen/Strep), the components of sample buffer and in table S7 the solution used to dilute the secondary antibodies for immunohistochemistry

RESPONSE: As requested by the reviewer, we now provide catalog numbers for relevant chemicals. The recipe for the sample buffer has been added to the methods section.

12) Labeling must be consistent in the paper (e.g. body fat-free mass or body lean mass). In Figure 1I, specify on the y axis "Relative mRNA expression VPS39". In Figure 1F: change the y axis title to "Frequency CpG sites (%)"

RESPONSE: Thanks for this comment. We have fixed it in the revised ms.

13) Is there a negative correlation between expression and methylation of CpG sites in genes with lower expression in NGT myoblast? e.g. *DDX5*, *SOCS4*, *MDK*, *TEAD4* according to figure 1D are downregulated in NGT myoblasts, do they have a higher methylation?

RESPONSE: Yes, there are negative correlations between expression and DNA methylation for *SOCS4*, *MDK* and *TEAD4* (but not for *DDX5*), which means that these genes have higher methylation when expression is lower (see Table S2).

Reviewer #2 (Remarks to the Author):

The manuscript by Davegardh and colleagues uses myoblast samples isolated from T2D patients and healthy individuals to determine transcriptional and epigenetic differences. While this approach is not novel (Mundry et al. 2017), the authors have identified VPS39 as a novel gene to have a role in human myogenesis. The authors found VPS39 to be downregulated in T2D, and that this downregulation affects DNA methylation. They propose a model in which VPS39 deficiency leads to alterations in autophagy, metabolism and epigenetics, which in turn trigger downregulation of myogenic regulators. Although the authors provide some compelling data, I feel the manuscript falls short in providing convincing demonstration to support their conclusions. Below major concerns:

RESPONSE: We appreciate that reviewer 2 believes we provide some compelling data and we thank the reviewer for his/her valuable comments that have helped us to improve the manuscript.

We also appreciate pointing out the study by Mudry et al. However, their study design is very different from ours. They studied DNA methylation using microarray in muscle from healthy people before and after insulin exposure. They then analyzed DNA methylation of some selected candidates in muscle from T2D individuals and controls and performed functional follow up experiments when silencing *ATP2A3* and *DAPK3*. In contrast, we studied myoblasts isolated from T2D patients and healthy individuals to determine genome-wide transcriptional and epigenetic differences before and after differentiation in order to find novel candidates that may affect myogenesis and glucose uptake in T2D. However, based on this comment, we now refer to this paper (ref 17) in the introduction of the revised ms.

1- Autophagy results are not convincing. Results from IF and western blot do not correlate in Baf-1 treated myoblasts. To convincingly demonstrate that there are differences in autophagy and specifically in what stage, the authors need to show data for siVPS39 and T2D myoblasts and T2D myotubes in the context of Baf-1 or another autolysosome inhibitor and assay the autophagic flux. Data showing autophagy only in the basal state does not provide enough information of impaired autophagy given that autophagy is a dynamic process. Several reviews propose the best practices to properly assess autophagy (Mizushima et al. 2010 and Klionsky et al. 2016), and the authors should take these as reference.

RESPONSE: We would like to thank the reviewer for this comment. Accordingly, we have now performed detailed autophagic flux measurements by using the autophagy inhibitor Bafilomycin A1 (Baf-A1) combined with serum- and amino acid-free starvation that induces autophagy. Baf-A1 acts

by inhibiting vacuolar H⁺ ATPase (V-ATPase), which prevents maturation of autophagic vacuoles by inhibiting fusion between autophagosomes and lysosomes. We had 4 experimental groups in the new experiments: non-treated, Baf-A1 treated, starved and starved+Baf-A1 treated, which were compared in both control and VPS39-silenced muscle cells at day 3 of differentiation. Autophagic flux was both assayed by Western Blot to measure the protein levels of LC3B and p62 in all conditions, and high content screening (HCS) analysis to define the number and area of the autophagy markers LC3B+, p62+, LAMP1+ and LAMP2+ stainings. Unfortunately, we were not able to perform the corresponding analyses in muscle cells from NGT and T2D individuals, as also suggested by the reviewer. These cells are stored in Copenhagen, Denmark and due to stricter ethical regulations in Denmark we were not allowed to transfer the cells to Lund University, Sweden, where the autophagic analyses were performed. Our new data are presented in Figures 3 and S2, and are described in a new paragraph in the Results section, pages 11-12.

First, we verified a significant increase of all autophagy markers already at basal condition in VPS39-silenced muscle cells (Figure 3a-d).

After starvation, when autophagy was activated, we found as expected a clear reduction of p62 and increase of LC3B-II in the control cells. On the contrary, after starvation the amount of LC3B-II protein levels and p62+ dot number and area was not changed in the VPS39-silenced cells. The absence of p62 reduction and increase of lipidated LC3B-II after autophagy activation by starvation in the VPS39-silenced cells indicated a clear alteration in the autophagy process. These data are presented on pages 11-12 of the revised ms:

“Next, we investigated the effects of VPS39 on basal autophagy in human myoblasts using both HCS and Western blot analyses for key markers of autophagy – LC3B, p62, LAMP1 and LAMP2. As shown in Figure 3a-c, VPS39-silenced cells displayed increased amount and size of autophagosomes, determined by quantification of LC3B spot number and area, and increased p62, a marker of selective autophagy³². p62 protein levels are inversely correlated with autophagic activity³², indicating impaired autophagic activity in VPS39-silenced cells. Moreover, the spot number and area of lysosomal markers LAMP1 and LAMP2 were also increased in the HCS analysis (Figure 3a-c). Additionally, the protein levels of LC3B-II and p62 (detected by Western blot) were increased in VPS39-silenced cells, whereas LAMP1 and LAMP2 were not altered (Figure 3d). The discrepancies in LAMP levels detected by HCS and WB, respectively, may depend on that these two methods have different sensitivities to be able to detect differences in LAMP1/2 levels. Together, these results support that knockdown of VPS39 alters basal autophagy in human myoblasts, similar to what was observed after inhibition of autolysosome formation (Figure S2b-d).”

To further investigate which step of the autophagic process was altered we measured autophagic flux using Baf-A1 treatment with and without starvation. We found a clear increase of LC3B-II protein levels in the control cells both with and without starvation after A1 treatment. This was expected as Baf-A1 prevents the autophagosomal lysosomal fusion therefore the autophagosomal marker LC3B-II increases. We have also demonstrated an efficient Baf-A1 treatment coupled with starvation in the control cells by detecting increased p62. p62 negatively correlates with autophagy and is selectively degraded by the autophagy pathway.

In contrast, the expression level of LC3B-II did not change in the VPS39-silenced cells after Baf-A1 treatment with and without starvation. The amount and area of endosomal/lysosomal/autolysosomal marker LAMP1/2 significantly increased after Baf-A1 treatment both with and without starvation. p62 levels further increased after stopping the autophagic flux at a later step in the VPS39-silenced cells. Altogether these data clearly indicate an alteration in the autophagic flux after VPS39 knockdown. The lack of increased LC3B-II expression after any of the treatments coupled with increased p62 and LAMP1/2 all indicate a late autophagy impairment. VPS39-silenced cells have a significantly increased amount of LC3B-II under basal conditions, which does not increase by Baf-A1 (with or without starvation), all pointing towards a decreased autophagic flux probably by a defect in

lysosomal degradation due to the increased LAMP1/2 level already at basal conditions that further increase after Baf-A1 treatment. These data are presented on pages 12 of the revised ms:

“To further characterize the role of VPS39 for regulation of the autophagic process, we performed comprehensive autophagy flux analyses³³. Human myoblasts were treated with Baf-A1 under both basal and starvation (serum- and amino acid-free) conditions to induce autophagy, and key markers of autophagy were monitored using the two methods described above. As determined by Western blot, both LC3B-II and p62 protein levels were increased in VPS39-silenced cells compared to control (Figure 3e-f). As expected, LC3B-II was increased upon starvation and was further increased in response to Baf-A1 treatment due to an accumulation of autophagosomes. p62 was increased in response to Baf-A1 treatment in both control and VPS39-silenced cells, but only control cells displayed a significant decrease in p62 upon starvation. The autophagic flux (calculated as LC3B-II levels in Baf-A1-treated compared to control cells) was decreased in VPS39-silenced cells, and only control cells displayed an increased autophagic flux upon starvation (Figure 3g). In line with results on protein level, p62 spot number and area were increased overall in VPS39-knockdown cells and only control cells displayed significantly decreased p62 upon activation of autophagy (Figure 3h-j). The reduction in p62 in response to starvation, determined by both Western blot and HCS, was significantly larger in control cells indicating an impaired autophagic activity in the VPS39-silenced cells (Figure S2h-j). Moreover, both LAMP1 (Figure 3k-m) and LAMP2 (Figure 3n-p) spot number and area were significantly increased in VPS39-silenced cells. These results support that VPS39 is required for a functional autophagy in human myoblasts and that VPS39 silencing is associated with a reduced autophagic flux, likely due to defects in the late stages of the autophagy process. Furthermore, VPS39 knockdown alone mimicked effects on altered basal autophagy and impaired myogenesis observed in Baf-A1-treated myoblasts.”

2- The data on LAMP1 in siVPS39 does not agree with previous report showing that VPS39 knockdown leads to decrease in LAMP-1 levels (Pols et al. 2012). The authors need to discuss this.

RESPONSE: We appreciate this comment and believe the reviewer refers to the paper by Pols et al published in Traffic 2013 (PMID:23167963).

However, their experiments are very different from ours. They silenced Vps39 in the HeLa cells and suggested that it is required for late endosomal-lysosomal fusion events. They distinguished endosomes and lysosomes by the presence or absence of BSA-gold as well as LAMP1 in order to follow transport of BSA-gold to late (LAMP1+) endo-lysosomal compartments in HeLa cells. Vps39 silenced cells had an increased number of BSA-/LAMP1+ endosomes and a decrease in BSA+/LAMP1+ endosomes. They also had a decreased number of BSA+/LAMP+ lysosomes. The researchers concluded that fusion of endosomes with endosomes or lysosomes is impaired or delayed when silencing Vps39.

Our study is performed in a different cell type and we used different methods (HCS) to study the autophagic process in VPS39-silenced myoblasts. We showed that VPS39-silenced muscle cells had larger and more autophagosomes, determined by quantification of the autophagosome marker LC3B. Moreover, using HCS we found increased levels of lysosomal markers LAMP1 and LAMP2 in the VPS39-silenced muscle cells.

We had referred to this reference at several places in the original version of our ms and we do now also discuss their data together with our new flux data on pages 21 of the revised ms:

“Next, we demonstrated that functional autophagy is important for proper human myogenesis, and that it is impaired in VPS39-silenced human myoblasts. To investigate which step of the autophagic process was altered, we measured autophagic flux using Baf-A1 treatment with and without starvation. VPS39-silenced cells have increased amounts of autophagy markers LC3B-II, p62, LAMP1 and LAMP2 under basal conditions, and activation of autophagy in response to starvation was impaired in these cells. These data clearly indicate an alteration in the autophagic flux after VPS39 knockdown. The lack of relative induction of LC3B-II protein levels after any of the treatments,

coupled with increased p62 and LAMP1/2, indicate a late autophagy impairment, potentially by a defect in lysosomal degradation due to the increased LAMP1/2 levels already at basal conditions that further increase after Baf-A1 treatment. On the other hand, using different cells and methods, Pols et al have previously suggested that Vps39 deficiency in HeLa cells impair or delay the fusion of endosomes with endosomes or lysosomes by following BSA-gold to LAMP1 positive compartments²¹.

3- In terms of relevance, additional experiments in mice would be important to determine if autophagy is also affected in the context of VPS39 downregulation. These studies would contribute to a better understanding of the role of VPS39 in autophagy and muscle.

RESPONSE: Based on this valid comment, we have now used Western blot analysis to analyze protein levels in skeletal muscle of some of the genes with differential mRNA expression in muscle biopsies from *Vps39*^{+/-} mice presented in **Figure 5f**. Among the genes presented in **Figure 5f**, we selected one enzyme involved in autophagy (ATG5) and one enzyme involved in epigenetic regulation (DNMT3B) for this Western blot analysis. We found that the protein and mRNA levels of these enzymes involved in autophagy and epigenetics correlated positively in muscle biopsies from *Vps39*^{+/-} and control mice further supporting a role for VPS39 in these processes. These data are presented on page 15 of the revised ms:

“Moreover, the mRNA and protein levels of autophagy protein 5 (ATG5) and DNMT3B correlated positively in muscle of the mice (Figure S4g).”

4- The connection between decreased VPS39 and epigenetics is not clear. The authors suggest that misregulated autophagy leads to changes in the metabolic state that in turn affect epigenetic enzymes causing impairment in myogenesis and muscle function. Although, the results support an effect on methylation upon VPS39 deficiency, current data on autophagy does not provide strong evidence that the misregulation of autophagy is the cause of changes in metabolic state.

RESPONSE: Throughout the revised ms, we have toned down that the “misregulation of autophagy is the cause of changes in metabolic state”. We also added the following sentence in the discussion on page 20:

“Although we suggest an order of events in Figure S3, it is possible that the progression from VPS39-deficiency to impaired myogenesis in T2D is slightly different.”

Based on this comment, we also performed new experiments and added the following data to page 12-13 of the revised ms:

“The autophagic process has been linked to metabolic remodeling and myoblast differentiation in rodents^{31,34}. We proceeded to examine whether VPS39-silenced human muscle cells have altered activation of key proteins involved in metabolic pathways i.e. glucose uptake and glycogen synthesis. VPS39-silenced cells exhibited decreased insulin-induced Akt phosphorylation, at both serine 473 (Ser473) and threonine 308 (Thr308) compared to control cells (Figure 4a). The phosphorylation of downstream Akt substrates TBC1D4 that controls GLUT4 translocation³⁵, and glycogen synthase kinase 3 (GSK3) - α that regulates glycogen synthase activity³⁶, were both decreased in insulin-stimulated VPS39-silenced cells versus control (Figure 4b-c). Phosphorylation of TBC1D4 is associated with increased glucose uptake and phosphorylation of GSK3 is associated with activation of glycogen synthesis. These results suggest that the overall activity of the insulin signaling pathway is perturbed in VPS39-silenced myoblasts, and that some metabolic pathways may be altered.”

5- To convincingly demonstrate that elevated H3ac in VPS39 deficient myoblast has a negative effect of myotube differentiation, the authors need to show H3ac levels during differentiation in control samples.

RESPONSE: Based on this valid comment, we added the requested experiment in Figure 4I of the revised ms and in line with a previous study we also see decreased H3ac levels during differentiation.

In these new experiments, we also found elevated H3ac levels at day 7 of VPS39 deficient myoblast, which have been included in Figure 4i. We changed the text on page 14:

“Global ac-H3 levels are expected to decrease during myoblast differentiation⁴⁷, which we clearly see in our control cells (Figure 4i). Specific acetylation of H3 was significantly decreased during differentiation in control cells whereas this response was altered in VPS39-silenced cells that displayed increased levels of H3 acetylation compared to control at both day 3 and 7 of differentiation (Figure 4i).”

6- p300 quantification and blot do not correlate. The quantification observed in Figure 3B does not correlate with the representative image shown on Figure S1H.

RESPONSE: We want to thank the reviewer for spotting this mistake in Figure S1H. Based on this comment, we have replaced the blot in the figure and added Western blot images used for figures to the Source data file.

Minor comments:

-Line 215: this observation is not new. The importance of autophagy in myogenesis has been previously documented in several papers.

RESPONSE: We agree that it has been shown to be important in rodent myogenesis but we are not aware of that it has been shown to be of importance in human myogenesis? We refer to this on page 11 and based on this comment we changed the sentence on line 215 (new line 203-206) slightly:

“Rodent data support that proper autophagy is important for myogenesis³¹, but this has not been verified in human muscle cells. Moreover, VPS39 is part of the complex mediating fusion of autophagosomes with lysosomes, but the role of VPS39 for regulation of autophagy in human muscle cells remain to be elucidated^{20,21}.”

Page 11: *“These observations further demonstrate the importance of autophagy during human myogenesis.”*

-Line 218: The text reads: ...“the levels of lysosomal markers LAMP1 and LAMP2 were only modestly affected.”, but they are not significant at the protein level, and only LAMP2 seems to show an increase by IF when analyzing relative intensity/spot area.

RESPONSE: We appreciate this comment. However, based on comment 1 from this reviewer, this section was rewritten and additional experiments were performed (see above and in the revised ms).

-Line 248: there is a typing error: “and since ta connection”.

RESPONSE: Thanks for spotting this, we have changed it to “the”.

Reviewer #3 (Remarks to the Author):

In this study, Cajsa Davegårdh et al. identified VPS39 as a novel regulator of muscle regeneration and function. For T2D individuals, the expression of VPS39 was lower in myoblasts and myotubes, which led to impaired autophagy and subsequently caused abnormal epigenetic reprogramming, dysregulation of myogenic regulators and perturbed differentiation. Furthermore, the muscle of Vps39^{+/-} mouse which could mimic the situation in human with T2D displayed reduced glucose uptake and altered expression of genes regulating autophagy, epigenetic programming and myogenesis. In conclusion, the study provided interesting insights into the connection between autophagy and T2D. A few suggestions were provided to further improve the quality of this work and reliability of the observations in this study.

RESPONSE: We are very happy that the reviewer believe our study provides interesting insights into the connection between autophagy and T2D. We also appreciate valuable comments that helped to improve our ms.

1. The authors used Infinium 450K BeadChips to analyze DNA methylation in myoblasts of 14 control and 14 T2D individuals and filtered 10,992 CpG sites annotated to the 577 unique genes that exhibited differential expression in myoblasts from diabetics versus controls. They studied negative correlations between DNA methylation and expression of these genes since DNA methylation is a known repressive mark. They identified 245 differentially expressed genes that displayed negative correlations between expression and methylation of one or more CpG sites. These included VPS39, TDP1, MAEA and FBN2 (Figure 1E). Considering this paper mainly studied VPS39, in order to rigorously test the results, the authors are suggested to choose another method to verify the methylation of VPS39 promoter.

RESPONSE: We agree that it is important to technically verify methods. We have therefore previously both technically and biologically verified DNA methylation data generated with the 450K BeadChips using pyrosequencing in numerous published studies for example E Hall et al Genome Biology 2014, (PMID:25517766), T Rönn et al Plos Genetics 2014 (PMID:23825961), T Dayeh et al Plos Genetics 2015 (PMID:24603685), A Olsson et al Plos Genetics 2014 (PMID:25375650), K Bacos et al Nature Communications 2015 (PMID:27029739). Based on these studies and our experience with the 450K data we are confident that also in the present study the DNA methylation data is technically sound. We discuss these studies on page 23 of the revised ms.

“Notably, twice as many methylation changes occurred during differentiation of myoblast from diabetics compared to controls based on analysis with the 450K array. Importantly, we have previously both technically and biologically validated DNA methylation data generated with the 450K in human samples¹³.”

We would have been happy to also perform pyrosequencing for technical validation in the present study but unfortunately we used all DNA from the T2D (n=14) and non-diabetic (n=14) individuals for the 450K microarray and to culture these muscle cells again for a new “batch” of DNA for technical validation would take approximately 3 months. We believe this is beyond the scope of this study. Moreover, the luciferase data presented in **Figure 1f** support a role for DNA methylation in regulation of *VPS39*.

2. The authors mainly used qPCR to identify the knockdown of proteins (VPS39, TDP1, MAEA and FBN2), western-blot is required to verify the level of the proteins.

RESPONSE: We appreciate this comment and agree that it is important to confirm knockdown on protein level. In the original submission we presented a clear reduction in VPS39 protein levels when silencing VPS39 using siRNA, both at day 3 and 7 (**Figure 2b**). For clarity, we have now moved this panel to **Figure 1h** so that knockdown of mRNA and protein levels are presented together.

Based on this comment, we have now also analyzed protein levels of TDP1, MAEA and FBN2 at day 7 of differentiation using Western blot in respective knockdown experiment. We chose to validate the knockdown efficiency only at this time point since that was when the cells were tested with the fusion index assay. In line with results on mRNA expression, MAEA and FBN2 were both significantly decreased on protein level. Unfortunately, we were not able to validate knockdown of TDP1 on protein level due to technical issues. For the Western blot analysis of TDP1 we have tested three commercially available antibodies, but neither of these antibodies gave specific bands for TDP1. However, given that the knockdown efficiency is similar on both mRNA and protein levels for the other genes/proteins studied (VPS39, MAEA and FBN2) it is likely that also TDP1 was decreased. These data are included in Figure S1d-f of the revised ms and described in the corresponding figure legend.

We also mentioned these results on page 9:

“Knockdown was confirmed at both an early stage of differentiation and after differentiation into myotubes (Figure 1h and S1d-f).”

3. The authors are suggested to add internal controls to all western-blot.

RESPONSE: As mentioned in the methods section, for Western blot normalization we used total protein normalization by the Stain-Free technology, based on the V3 workflow developed by Bio-Rad (PMID: 24429481). In principle, the Stain-Free gel is formulated with a fluorescent trihalo compound that binds covalently to tryptophan residues in proteins upon ultraviolet (UV) activation. Proteins are then visualized under UV light and the fluorescent signal quantified. For normalization, we used the signal intensity of the entire sample loaded in a lane instead of using the expression of an individual housekeeping protein (internal control). Total protein normalization using Stain-Free gels outperforms the use of housekeeping proteins (eg. β -actin, GAPDH) or other total protein stains (eg. Ponceau, Coomassie) in its sensitivity and reproducibility. Moreover, the “staining” is immediate and allows continuous monitoring of proteins in the gel or on the membrane at any step during the western blotting process (PMID: 23085117, PMID: 23747530). Based on the advantages of total protein normalization we used Stain-Free gels throughout the paper to normalize protein levels. Post-translational modifications (i.e. acetylation, phosphorylation) were instead normalized to the total levels of each corresponding protein after stripping and reprobing the membrane. To be able to compare samples run on separate gels and blotted to separate membranes, we included a loading reference sample in an outer lane of each gel. The quantified intensities for all samples were then normalized to the loading reference before calculating the relative values.

Based on this comment, excel data from the Stain-Free images of the membranes used for quantification are now provided in the source data file together with blot images used to compile the graphs.

4. The authors are suggested to use western-blot to compare the level of TFs (MYOD1 and MEF2s) in control and T2D.

RESPONSE: We thank the reviewer for this comment. We have analyzed MEF2C and MYOD1 protein levels in T2D and NGT muscle cells during differentiation using Western blot (see FIGURE 1 below). Both MEF2C and MYOD1 were regulated in a similar manner in both T2D and NGT muscle cells. MEF2C increased with increasing degree of differentiation, whereas MYOD1 was upregulated at an early stage of differentiation (day 3) and was then reversed again in the fully differentiated cells (day 7). However, due to large inter-individual variation in both groups we found no significant differences between the two groups at any of the time points. Differences in protein levels between T2D and NGT individuals may be subtle and in order to confidently conclude whether any differences exist analysis of more individuals is needed. Furthermore, neither MEF2s nor MYOD1 were among the significant differentially expressed genes in NGT and T2D myoblasts and myotubes (Table 1, sheets A-B). Due to the large amount of data in the ms, we did not include these data in the revised ms, but only present it here.

FIGURE 1. MEF2C and MYOD1 protein levels in muscle cells from individuals with type 2 diabetes (T2D) and controls (NGT). MEF2C (left panel, $n=5$ per group) and MYOD1 (right panel, $n=2-4$ per group) protein levels (Western blot) in human muscle cells during differentiation. Representative blots are shown. NGT at day 0 is set to 1. The effects of differentiation stated above the graphs were calculated with repeated measurements 2-way ANOVA. # $q < 0.05$, ## $q < 0.01$, ### $q < 0.001$, #### $q < 0.0001$ for comparisons between time points within each genotype. Values were log2-transformed before statistical analyses.

We also analyzed MEF2C and myosin protein levels in VPS39-silenced cells and found strong effects. These data are included on page 10 of the revised ms:

“Moreover, MEF2C and myosin protein levels were markedly reduced in VPS39-silenced cells at day 7 of differentiation (Figure S1j-k), further demonstrating the key role of VPS39 in human myogenesis.”

5. Figure 2J, the authors are suggested to provide a better resolution picture.

RESPONSE: We have tried to improve the resolution of this picture now presented in Figure 3.

6. To test the importance of autolysosome formation during human myogenesis, the authors treated human myoblasts with Bafilomycin A1 (Baf-A1) for 3 hours per day during the first 3 days of differentiation and studied key markers of autophagosomes (LC3B), autolysosomes (p62), and lysosomes (LAMP1 and LAMP2) using both cellomics and Western blot analyses (see Figure S1D for experimental setup). Indeed, Baf-A1 impaired autophagy (Figure S1E) and the expression of myogenic markers was lower in the treated human muscle cells (Figure S1F). In order to prove the consistency of the experimental results, the authors are suggested to add the expression of myogenic markers in Figure S1E western-blot.

RESPONSE: The expression of myogenic markers (*MYOD1*, *MYOG* and *TNNI1* mRNA and MYOD1 protein) in Baf-A1-treated cells were presented in Figure S1F in the original ms. Based on comment 1 from reviewer 2, we have changed this section of the ms and added more data to study the autophagic flux in VPS39-silenced muscle cells. Therefore, these data are now presented in Figure S2e-f of the revised ms. We hope the reviewer is fine with the order in this figure.

7. Baf A1 inhibits the fusion of autophagosomes with lysosomes, VPS39 is part of the complex mediating fusion of autophagosomes with lysosomes, so VPS39-knockdown would have a same effect as Baf A1. The authors are suggested to use cellomics and western-blot analyses (including the expression of myogenic markers) in Figure S1G.

RESPONSE: We appreciate this comment and agree with the reviewer that VPS39 knockdown should have similar effects to Baf A1. Indeed, the autophagy markers are regulated in a similar manner in Baf-A1-treated and VPS39-silenced cells. In the original ms we already used automated high content screening (HCS) and WB in both Baf-A1-treated and VPS39-silenced cells to analyze effects on basal autophagy. This data was presented in Figures S1E and 2I-J. In the revised version of the

manuscript these panels are now presented as Figures S2b-d and 3a-d. Based on comment 1 by reviewer 2, we have performed new experiments to also analyze the autophagic flux under these conditions (see Figure 3). Please see the response above and page 11-12. In the original manuscript we investigated the effect of VPS39-silencing on the mRNA expression of numerous myogenic markers at day 3 of differentiation by microarray and qPCR. This data was presented in Figure 2d and 2h (Figure 2c and 2h in the revised manuscript). We have now also analyzed the protein levels of the myogenic markers MEF2C and myosin in VPS39-silenced muscle cells during differentiation. As in Baf-A1-treated myoblasts, the expression of myogenic markers was lower in VPS39-silenced cells. These results are presented in Figure S1j-k in the revised manuscript, and described in the text on page 10.

8. Why it is described in the paper that the nuclear protein levels of p300 was increased in Figure 3B and Figure S1H, but the results showed that p300 was increased in Figure 3B and decreased in Figure S1H.

RESPONSE: We want to thank the reviewer for spotting this mistake in Figure S1H (new **Figure 4e**). Based on this comment, we have replaced the blot in the figure and added all western blot images used for the figures to the Source data file.

9. To verify VPS39 levels in Vps39 +/- mice to mimic the situation in human with T2D, the authors are suggested to compare the expression of VPS39 in T2D and Vps39 +/- mice.

RESPONSE: The average reduction in *VPS39* expression (as determined by microarray) in human T2D myoblasts and myotubes compared to NGT is 21.4% and 23.6%, respectively. The corresponding reduction in *Vps39* expression (as determined by qPCR) is 26% in skeletal muscle from *Vps39*^{+/-} compared to WT mice. Hence, the reduction in *Vps39* expression in muscle cells from T2D individuals and muscle from *Vps39*^{+/-} mice are quite similar. Based on this comment, we added this information to the discussion on page 23 of the revised ms.

“To further study the role of VPS39 in vivo, we examined a mouse model for VPS39-deficiency. Heterozygous mice were chosen because they have reduced, but not lacking, expression of VPS39, which resembles the situation seen in T2D individuals. Vps39 expression was reduced by 26% in muscle from Vps39^{+/-} mice, while it was reduced by 21.4% and 23.6% in myoblasts and myotubes from T2D individuals, respectively.”

10. The literatures are not well discussed. VPS39 participates in autophagy regulation as a component of HOPS complex, and the activity is also regulated by mTORC1 complex, which has fundamental roles in T2D. A recent study indicated that the activity of HOPS complex is regulated by mTORC1 (PMID: 30704899). In addition, HOPS complex also regulates endocytic pathway (PMID: 23645161), what could be the involvement of endocytic pathway in the connection between VPS39 and T2D?

RESPONSE: We appreciate this suggestion by reviewer 3 and have now expanded the discussion on page 22 and included the suggested references:

“We then tested if myoblasts and myotubes from T2D individuals resemble the phenotypes seen in VPS39-silenced muscle cells. In agreement, a marker of autophagy was altered in muscle cells from T2D individuals. Interestingly, the expression of several genes related to autophagy and mTOR signaling were regulated differently during myogenesis in T2D compared with control subjects. For example, MLST8, encoding a component of mTORC1 and mTORC2, which regulates lysosome function and autophagy⁵⁴, was upregulated in diabetics and downregulated in control individuals during myogenesis. Autophagy is suggested to be a leading cause of reduced muscle mass and quality during aging and metabolic diseases^{59,60}, and the myogenic potential of satellite cells decrease with age⁶¹. mTOR signaling is central for regulation of metabolic pathways, myogenesis and autophagy^{54,62} and regulated by the availability of amino acids⁶³. Additionally, mTORC1 signaling is aberrant in the skeletal muscle from T2D individuals^{64,65} and the activity of the HOPS complex was recently suggested to be regulated by mTORC1⁶⁶. Interestingly, the HOPS complex may also regulate the

*endocytic pathway*⁶⁴. GLUT4 is the main glucose transporter in skeletal muscle cells and GLUT4 storage vesicles traffic in endocytic and exocytic compartments⁵⁶. GLUT4 translocation was found to be impaired in muscle from T2D individuals⁶⁷. Future studies may test whether lower levels of VPS39, a subunit of the HOPS complex, also affect the endocytic pathway and thereby GLUT4 translocation and glucose uptake in muscle. This could be an additional mechanism contributing to the reduced glucose uptake in muscle of T2D individuals. In support of this, we found reduced glucose uptake in muscle from *Vps39*^{+/-} mice, lower insulin-stimulated Akt, TBC1D4 and GSK3 phosphorylation in VPS39-silenced muscle cells as well as a positive correlation between insulin-stimulated glucose uptake and VPS39 expression in human skeletal muscle.“

Reviewer #4 (Remarks to the Author):

Davegardh et al studied here whether alterations in myoblasts contribute to lower muscle mass and insulin resistance in type 2 diabetes (T2D). Their major claims are that VPS39 deficiency in myoblasts impair autophagocytosis resulting in disturbed metabolic homeostasis, which in turn leads to alterations in the expression of epigenetic enzymes and thus altered epigenetic marks. Altered epigenetic programming then leads to altered expression of important transcription factors and muscle specific genes resulting in reduced myoblast differentiation and increase of apoptosis. These events contribute to the hallmarks of T2D, and suggest VPS39 as a target for T2D therapies. Demonstrating the above chain of events underlying T2D pathology in muscle cells is novel. In addition, VPS39 has not been shown before to act as an important regulator of human myogenesis.

Overall, this study is on an important and timely health topic, T2D, prevalence of which is increasing in the population. Thus the results from this study will be of interest for a broad audience from basic to clinical researchers. This study includes a large number of experiments, each resulting in a hypothesis of which the next experiment aims to test, all logically following each other, and bringing evidence together to support the full story. The manuscript is very well written and easy to follow. However, there are several issues that I feel require further clarification from the authors.

RESPONSE: We thank reviewer 4 for helpful comments and we are happy that he/she finds our ms well written and of interest for a broad audience.

1. The whole story this manuscript tells is logical, however, I wonder if the authors had a prior hypothesis, which does not come up in the manuscript? Or how the authors happened to pick a gene, VPS39 (the top 92nd gene, plus the other 5 they mention), among all the 577 genes that were differentially expressed (DEG) in the myoblasts of T2D vs controls? What was the rationale for highlighting these genes already from the very beginning?

RESPONSE: Our main hypothesis/aim was to try to find novel candidates in T2D myoblasts and myotubes that contribute to deficiencies seen in T2D e.g. affecting muscle regeneration and/or glucose homeostasis. On page 8, we describe the selection criteria of candidates for functional follow-up experiments:

“To identify new regulators of muscle regeneration and function, we asked whether any of the genes with reduced expression in both myoblasts and myotubes from T2D versus controls, and with inverse correlation between their expression and DNA methylation, also have a functional role in human myogenesis. We mainly focused on genes that had not been previously studied in human muscle cells, but with known functions in other cell types or species that would suggest an impact also on myogenesis¹⁹⁻²⁶.”

There were only 5 genes that fulfilled these three criteria; *VPS39*, *TDPI*, *MAEA*, *FBN2* and *PCMTD2*. There were **i)** 577 DEGs in myoblasts, **ii)** 20 DEGs in myotubes that overlapped with the 577 DEGs in myoblasts and **iii)** 5 of these 20 genes had negative correlations between mRNA expression and DNA methylation in the myoblasts.

Among the 5 genes that fulfilled these three criteria, we used the search term “human skeletal muscle” in PubMed and found that 4 of these genes had previously not been studied in human muscle (*VPS39*, *TDPI*, *MAEA* and *PCMTD2*). While one of these genes (*FBN2*), which was also number one in Table S1 (sheet A), had previously studied in human skeletal muscle.

Next, we searched PubMed for the 4 identified genes previously not studied in human skeletal muscle to examine if their function in other cell types may indicate that they could have a role in T2D muscle. *VPS39* is part of the HOPS complex that affect autophagy, *TDP1* is a member of the phospholipase D family and a recent study showed that *SCAN1-TDP1* in mitochondria creates a pathological state that allows neurons to turn on mitophagy to rescue fit mitochondria as a mechanism of survival. *MAEA* encodes a protein that plays a role in erythroblast enucleation and in the development of mature macrophages and SNPs linked to *MAEA* showed significant association with T2D. There are only two papers published for *PCMTD2* and these do not support a potential role in muscle. We therefore selected *VPS39*, *TDPI*, *MAEA* and *FBN2* genes for functional follow-up.

It may not be fully clear why certain genes are presented in each step in the result section. We wanted to highlight some genes in each step because we don't find it very interesting to only present the number of genes and gene lists. Moreover, we tried to explain why we present certain genes, for example on page 7:

“These included several genes that had not previously been studied in human myoblasts but with identified functions in other cell types or species that suggest that they may also have a role in human muscle cells, e.g. VPS39, TDPI, and MAEA¹⁹⁻²⁵, as well as genes previously implicated in muscle regeneration, e.g. FBN2, TEAD4 and STAT3^{26,27} (Figure 1c-d).”

Nevertheless, we would be happy to delete specific genes in the different steps in the result section if reviewer 4 prefers that.

2. Gene expression analyses were logically followed with DNA methylation profiling of the 577 DEG in the same cells, and the authors searched for negative correlations between expression and methylation. As it has been shown for numerous times, also by these authors, the impact of DNA methylation on gene expression is far from being straight forward. DNA methylation may increase or decrease transcription, depending for example on the genomic context. Therefore, it is surprising that the authors decided to only study negative correlations between methylation and gene expression. Could the authors please clarify the basis for this choice?

RESPONSE: We agree with the reviewer that increased DNA methylation may also be associated with increased expression depending on the genomic location. Based on this comment, we have added positive correlations in Table S2 and changed the text on page 7-8.

“We then studied correlations between methylation and expression of these 577 genes since methylation may regulate gene expression. We identified 331 differentially expressed genes that displayed nominal correlations between expression and methylation of one or more CpG site ($p < 0.05$, Table S2). These included VPS39, TDPI, MAEA and FBN2.”

3. The correlation analyses come with multiple testing burden, but the authors have only reported nominal p-values. Could the authors please adjust the p-values accordingly, or justify the use of nominal p-values if they disagree?

RESPONSE: Based on this valid comment, we added the word “nominal” on page 8 and we performed/added a FDR analysis on the data in Table S2. Unfortunately, we are under-powered in this analysis (number of samples (n=28) are low compared to number of tests), why we followed up some the results with luciferase analysis showing direct effects of methylation on the transcriptional activity.

4. The negative correlations were observed for 245 transcripts, and the authors highlight again the same 4 genes as before, although many more were identified and a high number of other genes

showed stronger correlations. This fits very nicely to the story, but the rationale for picking these 4 genes is not clear at this point. Could the authors please elaborate on this?

RESPONSE: Please see our answer to comment 1.

5. Next the authors explored whether the same myoblast DEGs are also DEGs in myotubes of T2D vs controls, and identified 42 DEGs, of which 20 were the same as in myoblasts. These include also the 4 genes that have been highlighted from the beginning of this manuscript. The authors conclude that these genes likely have a role in T2D muscle pathology, and selected “the 4 genes” for functional experiments. Here they aimed to find novel muscle regulators and selected genes based on the following criteria: reduced expression in both myoblasts and myotubes in T2D, inverse correlation between expression and methylation, and known functions in other cell types suggesting a potential impact on myogenesis. How many genes fulfilled these criteria?

RESPONSE: As mentioned in our answer to comment 1, only *VPS39*, *TDPI*, *MAEA*, *FBN2* and *PCMTD2* fulfilled these criteria and were differentially expressed in both T2D myoblasts and myotubes and did also show correlations with DNA methylation in myoblasts. We did not follow up *PCMTD2* because we did not find any support in the literature that this gene may have a role in muscle cells or diabetes.

6. The authors showed that silencing of VSP39 resulted in lack of myotube formation and concluded that VPS39 is a putative regulator of myoblast differentiation, hypothesis of which they went to explore further by measuring protein levels, gene expression, DNA methylation and cell physiology in VPS39 silenced vs control myoblasts differentiating into myotubes (n=6 per group). They identified 2635 DEGs including myogenic transcription factors and muscle specific genes downregulated, DNA replication, cell cycle and autophagocytosis related gene sets upregulated, and various epigenetic enzymes showing differential expression in VSP39-deficient vs control cells in early differentiation. These epigenetic enzymes that were identified here, were there more of them identified than what would be expected by chance?

RESPONSE: We appreciate this comment and have tested if the epigenetic enzymes with differential expression in VPS39-silenced muscle cells are more than expected by chance. We added this information on page 10 of the revised ms.

“These are more than expected by chance based on the search terms epigenetics and histones and a chi-2 test ($p_{\text{chi}^2} < 0.05$).”

7. PSCAN and JASPAR were used to search for transcription factor binding motifs in the DEG identified in the VPS39 deficient vs control cells. Could the authors please describe how these tools were used in the methods section of the manuscript? This would help in interpreting the predictions. How are the Z-scores and the associated p- or q-values interpreted? These prediction tools identified 97 motifs, and the authors decided to highlight the motif at top 68th position, i.e. myogenin. Rationale for this over the more significant motifs?

RESPONSE: We now describe these tools in the method section on page 30. In the ms, we highlighted TF that were also differentially expressed in VPS39-silenced muscle cells.

“PCAN

PCAN Web Interface²⁸ together with JASPAR⁷⁴ were used to find enriched transcription binding motifs 0-1,000 bp upstream of transcription start sites of differentially expressed genes. PCAN is a software tool that scans promoter sequences from co-regulated genes, looking for over-represented motifs describing the binding specificity of known TFs, thus providing quick hints on which factors could be responsible for the patterns of expression observed. FDR-corrected significance threshold was used. We highlighted TF that were also differentially expressed in VPS39-silenced muscle cells.”

8. The authors tested if autolysosome formation is important in myogenesis and state on line 207 that this has not been studied before in muscle cells. I suppose the authors refer here to human myogenesis, as it has been studied at least in rodents (reference 37) and drosophila (Fujita et al eLife. 2017; 6: e23367, doi: 10.7554/eLife.23367). Please correct accordingly.

RESPONSE: We apologize for this and have added “*human*” at several places throughout the revised ms. We also write on page 11; “*Rodent data support that proper autophagy is important for myogenesis³¹, but this has not been verified in human muscle cells.*”

9. As a number of epigenetic enzyme gene and protein expression as well as their activity was altered in VPS39-defective cells the authors assessed DNA methylation in these cells vs controls (n=6). They observed 5045 CpG sites with differential methylation with nominal $p < 0.05$, of which almost 2/3 annotated to the DEGs observed in the same cell comparisons. Here the authors consider $p < 0.05$ as significant, although in the methods it is stated the FDR $q < 0.05$ was considered as significant. Table S4 shown that the differentially methylated genes highlighted in the manuscript body text show only tiny if any difference in the mean methylation of the CpG sites (range for all CpGs 0.4-10%, but for the highlighted genes e.g. MEF2s 1-4% difference in means, i.e these are not within the top differentially methylated CpGs). What is the authors’ argument for the p-value cut off for significance here?

RESPONSE: We appreciate this comment and agree that it should be more clearly stated that for this analysis we did not correct the individual p-values in Table S4 for multiple testing since our goal was to study general epigenetic phenomena/effects in VPS39-defective cells and show that a large number of the nominally differentially methylated sites were linked to the differentially expressed genes (for which we applied $q < 0.05$). Based on this valid comment, we have performed a chi2 test and we found the number CpG sites with “nominally” altered DNA methylation in VPS39-silenced cells to be more than expected by chance. We also changed the sentence on page 13 of the revised ms:

“We next asked whether the changes seen in epigenetic enzymes were associated with epigenetic alterations in VPS39-silenced human cells. We observed nominally altered DNA methylation at 5,045 CpG sites annotated to 72% (1,889) of genes that exhibited differential expression after VPS39 knockdown (Figure 4f and Table S4, $p < 0.05$). The number of sites with nominal methylation differences were significantly more than expected by chance ($p\text{-}\chi^2 < 0.01$).”

We also changed the methods section on page 34 based on this valid comment:

“Paired t-tests were used in siRNA knockdown experiments comparing NC vs siVPS39. FDR was applied when analyzing array expression data after VPS39-silencing, except for DNA methylation data where nominal p-values are presented and a χ^2 -test was used to examine if the number of nominal differences in methylation is more than expected by chance.”

10. I also suggest the authors mark in Table S4 those genes they refer to as “Specific genes of importance” in the body text on page 11, line 239.

RESPONSE: Based on this comment, we added this information to Table S4.

11. The authors conclude that their data support a model where low levels of VPS39 in the myoblasts impair autophagy, which results in disturbed metabolic homeostasis and is followed by alterations of epigenetic enzymes and the epigenome. This results into lower expression of key myogenic regulators and muscle specific genes, and reduced proportion of myoblasts differentiating to myotubes and increase in the rate of apoptosis. As the manuscript is very long, it would be helpful if the authors could add the main findings supporting this chain of events into Figure 4.

RESPONSE: We have added some main findings into this figure, which has now been moved to the supplement (Figure S3) based on a comment from the Editor.

12. The mouse model seems to mimic well human T2D in regards to glucose intolerance and reduced glucose uptake in their muscle tissue. The findings also nicely support the results in human cells. However, the significance of the expression findings are based on $p < 0.05$, not $q < 0.05$, and it looks like none of the findings in the human sample could be replicated if the p -values were corrected for multiple testing. It is also not clear how to interpret the Table S5 sheet B listing gene ontologies. The authors claim that the analysis revealed an overrepresentation of genes associated with epigenetics and histones, and a nominal significant enrichment of genes associated with muscle. What is this statement based on? How was the overrepresentation calculated? There are some highlighted GOs with 0 or 1 differentially expressed genes (with $p < 0.05$). How should one interpret this table? Also, the GOs highlighted in the text are not among the top DEGs or GOs, based on Table S5. Can the authors please clarify these issues?

RESPONSE: Based on this comment, we added “*nominally*” on page 15 of the revised ms: “Microarray analysis revealed 1,641 *nominally* differentially expressed genes in muscle of *Vps39*^{+/-} versus WT mice ($p < 0.05$, Table S5, Sheet A).”

As mentioned on page 15, the studied GO terms were selected based on our data in the VPS39-silenced human muscle cells, where we found alterations in autophagy, epigenetics, muscle specific genes and genes involved in oxidative phosphorylation:

“To better understand the biological relevance of these expression differences, and relate them to our human data, we searched the Gene Ontology (GO) terms for these 1,641 genes using four search-terms: autophagy, epigenetics and histones, muscle (excluding cardiac and smooth muscle), as well as oxidative phosphorylation and respiratory chain.”

Table S5 sheet B presents the occurrence of these Gene ontology (GO) biological process terms among all analyzed and the significant genes. For example, we count the number of times we find “histone” in any GO term with differentially expressed gene(s) ($p < 0.05$) and then test if the number is more than expected by chance. Moreover, as presented in the legend of **Figure 5e-f**, we used χ^2 tests to analyze overrepresentation of differentially expressed genes belonging to a GO term compared with all analyzed genes. Based on this comment, we have also added the following sentence on page 15 of the revised ms:

“We then used χ^2 -tests to examine an overrepresentation of differentially expressed genes belonging to these search-terms versus all analyzed genes.”

Moreover, based on this comment we also performed a gene set enrichment analysis (GSEA) on the complete mRNA expression data set in skeletal muscle of *Vps39*^{+/-} and wild-type mice and after FDR we found 3 significant KEGG pathways. These data are included on page 16 and in Table S5, sheet C of the revised ms:

*“We proceeded to perform a gene set enrichment analysis (GSEA) on the complete expression data set in *Vps39*^{+/-} versus WT mice³⁰. This analysis revealed 3 significant pathways, including the proteasome, ribosome and spliceosome (Table S5, Sheet C).”*

In addition, we validated some of the mRNA microarray findings found in the *Vps39*^{+/-} mice using Western Blot. These data are included on page 16 of the revised ms.

“Moreover, the mRNA and protein levels of autophagy protein 5 (ATG5) and DNMT3B correlated positively in muscle of the mice (Figure S4g).”

13. After examining the mouse model, the authors went back to the T2D individuals and identified DNA methylation differences in myoblasts vs myotubes in 14 T2D patients vs 14 controls. The authors report T2D myoblasts having higher methylation in certain gene regions with $q < 0.07$. This is again slightly confusing as the significance threshold is again different. Why is this?

RESPONSE: We agree with the reviewer that this is confusing and have changed this section on page 17-18 and Table S6 sheet A in the revised ms and only present data with $q < 0.05$.

14. Also, on page 15, line 336: “in these regions” refer to the regions that show higher mean methylation in myoblasts of T2D compared with controls, and when looking at Table S6 sheet A, this is not true. Please match the text with the results shown in Table S6 sheet A (both controls and T2D increase in some gene regions, and also in CpG island regions).

RESPONSE: Based on this and the previous comment, we have replaced “in these regions” with “*in the shelves and open sea*” on page 17 of the revised ms.

15. The number of differentially methylated CpGs in myoblast vs myotube and T2D vs controls are huge, and it is difficult to draw any other conclusions but that the methylation alterations were broader in T2D vs controls. The authors however report 39 CpG sites that showed opposite methylation change during myogenesis of T2D vs controls. It would have been interesting to see whether there was any overrepresented GO terms these genes belong to, which could give a hint on why they were with opposite methylation patterns.

RESPONSE: Based on $FDR < 0.05$, there were no overrepresented GO terms for the genes annotated to these 39 sites.

16. In the methods statistics –section it is stated for all the analyses with multiple testing the significance is based on $FDR q < 0.05$, but in certain tables it is $FDR q < 0.07$ or even $p < 0.05$ which was considered as significant (same in the text referring to these results). Could the authors please give a rationale for these inconsistencies?

RESPONSE: We agree with the reviewer that this is confusing and have changed the section on page 17 and Table S6 sheet A in the revised ms and only present data with $q < 0.05$ (not $q < 0.07$).

Moreover, we also agree that it should be more clearly stated for the analysis presented in Table S4 and on page 13, that the individual p-values were not corrected for multiple testing since our goal was to study general epigenetic phenomena/effects in VPS39-defective cells and show that a large number of the nominally differentially methylated sites were linked to the differentially expressed genes (for which we applied $q < 0.05$). Based on this valid comment, we have performed a χ^2 test and we found the number CpG sites with “nominally” altered DNA methylation in VPS39-defective cells to be more than expected by chance. We also changed the sentence on page 13 of the revised ms:

“We next asked whether the changes seen in epigenetic enzymes were associated with epigenetic alterations in VPS39-silenced human cells. We observed nominally altered DNA methylation at 5,045 CpG sites annotated to 72% (1,889) of genes that exhibited differential expression after VPS39 knockdown (Figure 4f and Table S4, $p < 0.05$). The number of sites with nominal methylation differences were significantly more than expected by chance ($p\text{-}\chi^2 < 0.01$).”

We also changed the methods section on page 34 based on this valid comment:

“Paired t-tests were used in siRNA knockdown experiments comparing NC vs siVPS39. FDR was applied when analyzing array expression data after VPS39-silencing, except for DNA methylation data where nominal p-values are presented and a χ^2 -test was used to examine if the number of nominal differences in methylation is more than expected by chance. “

17. All in all, the level of detail provided in this manuscript regarding the methodology is appropriate and will enable other researchers to reproduce their work, and the statistical analyses performed were valid, to the best of my understanding.

RESPONSE: We are happy to receive this positive feedback and want to thank the reviewer for constructive comments that helped us to improve our manuscript.

18. Finally, I fully agree with the authors that the pathway from impaired expression of VPS39 to low muscle mass and insulin resistance in T2D makes sense. However, have the authors considered any other alternative routes their results would also support?

How confident can the authors be for the causality of events they propose here?

Which results support the claim that impaired autophagy resulted in altered metabolic state?

And that this disturbed metabolic state would result in expression changes in the epigenetic enzymes and epigenetic makeup of the VPS39 deficient cells, or T2D muscle cells?

Which occurred first epigenetic or gene expression changes, any support for either?

There were long lists of differentially expressed genes and differentially methylated CpG sites identified in myoblasts vs myotubes, and T2DM vs controls as well as during myogenesis in VPS39 deficient vs control cells.

There may be other highly important genes than those highlighted throughout the manuscript among these. The authors should discuss the potential other conclusions on the chain of events their data support, as well as mention clearly the reasons for only following up certain genes (even though they were not the top significant hits) and ignoring the rest (excluding some GSEA/GO analyses these genes contributed to).

RESPONSE: We appreciate this comment and we have tried to address it in the revised ms. Our data clearly demonstrate that reduced VPS39 levels result in impaired myogenesis (**Figure 1i-j**). We then tried to dissect the mechanisms behind this deficiency. We are confident that reduced VPS39 levels changes the autophagic flux. We also found reduced AKT, TBC1D4 and GSK3 phosphorylation in VPS39-silent muscle cells further supporting insulin resistance and potentially metabolic alterations. The level of numerous epigenetic enzymes and epigenetic modifications are altered as well as the expression of key factors involved in myoblast differentiation. We do not state that there is causality and we agree with the reviewer that the exact order of these events are difficult to dissect. Based on this comment we changed the figure legend to the schematic **Figure S3** and added the following sentence to page 20 of the discussion.

*“Although we suggest an order of events in **Figure S3**, it is possible that the progression from VPS39-deficiency to impaired myogenesis in T2D is slightly different.”*

We have also performed several additional experiments and added the following text on page 12-13 of the result section:

*“The autophagic process has been linked to metabolic remodeling and myoblast differentiation in rodents ^{31,34}. We proceeded to examine whether VPS39-silenced human muscle cells have altered activation of key proteins involved in metabolic pathways i.e. glucose uptake and glycogen synthesis. VPS39-silenced cells exhibited decreased insulin-induced Akt phosphorylation, at both serine 473 (Ser473) and threonine 308 (Thr308) compared to control cells (**Figure 4a**). The phosphorylation of downstream Akt substrates TBC1D4 that controls GLUT4 translocation ³⁵, and glycogen synthase kinase 3 (GSK3) - α that regulates glycogen synthase activity ³⁶, were both decreased in insulin-stimulated VPS39-silenced cells versus control (**Figure 4b-c**). Phosphorylation of TBC1D4 is associated with increased glucose uptake and phosphorylation of GSK3 is associated with activation of glycogen synthesis. These results suggest that the overall activity of the insulin signaling pathway is perturbed in VPS39-silenced myoblasts, and that some metabolic pathways may be altered.”*

Moreover, based on this valid comment, we added the following sections to the discussion.

“We then tested if myoblasts and myotubes from T2D individuals resemble the phenotypes seen in VPS39-silenced muscle cells. In agreement, a marker of autophagy was altered in muscle cells from T2D individuals. Interestingly, the expression of several genes related to autophagy and mTOR signaling were regulated differently during myogenesis in T2D compared with control subjects. For example, MLST8, encoding a component of mTORC1 and mTORC2, which regulates lysosome function and autophagy⁵⁴, was upregulated in diabetics and downregulated in control individuals during myogenesis. Autophagy is suggested to be a leading cause of reduced muscle mass and quality during aging and metabolic diseases^{59,60}, and the myogenic potential of satellite cells decrease with age⁶¹. mTOR signaling is central for regulation of metabolic pathways, myogenesis and autophagy^{54,62} and regulated by the availability of amino acids⁶³. Additionally, mTORC1 signaling is aberrant in the skeletal muscle from T2D individuals^{64,65} and the activity of the HOPS complex was recently suggested to be regulated by mTORC1⁶⁶. Interestingly, the HOPS complex may also regulate the endocytic pathway⁶⁴. GLUT4 is the main glucose transporter in skeletal muscle cells and GLUT4 storage vesicles traffic in endocytic and exocytic compartments⁵⁶. GLUT4 translocation was found to be impaired in muscle from T2D individuals⁶⁷. Future studies may test whether lower levels of VPS39, a subunit of the HOPS complex, also affect the endocytic pathway and thereby GLUT4 translocation and glucose uptake in muscle. This could be an additional mechanism contributing to the reduced glucose uptake in muscle of T2D individuals. In support of this, we found reduced glucose uptake in muscle from Vps39^{+/-} mice, lower insulin-stimulated Akt, TBC1D4 and GSK3 phosphorylation in VPS39-silenced muscle cells as well as a positive correlation between insulin-stimulated glucose uptake and VPS39 expression in human skeletal muscle.”

“T2D is a complex disease where both numerous genetic and non-genetic factors affect the pathogenesis and subsequently most likely also some of the results in the present study. We found differential expression of several genes between the diabetic and non-diabetic groups, and although we focused this study on dissecting the role of VPS39 in muscle cells, other genes do also contribute muscle dysfunction and insulin resistance seen in T2D. Moreover, to reduce the risk of random findings due to individual variation between people we included as many individuals as technically possible in this study.”

Reviewers' Comments:

Reviewer #1:

Remarks to the Author:

my comments have been sufficiently addressed

Reviewer #2:

Remarks to the Author:

The revised manuscript by Davegardh and colleagues addresses many of the concerns raised previously. Some aspects that still require attention are detailed below:

1- The authors have conducted extensive experiments to understand the autophagy defect upon the silencing of VPS39.

a. It is not clear why a three-way ANOVA was used for statistical analysis in the autophagic flux. Can the authors justify such an analysis? It seems a two-way ANOVA is more suitable (Two factors genotype and drug treatment).

b. As mentioned in the manuscript, they have contradictory observations between HCS and WB for LAMP1 and LAMP2 data. They claim it is due to a difference in sensitivity, which is the more sensitive assay? Please provide WB data for LAMP1/2 autoflux.

c. Can the authors please provide a blot image with complete LC3 for Fig. 3d.

2- The authors claim there is a decrease in GSK3 in insulin-stimulated VPS239-silenced cells versus control, but this is not supported by the statistical analysis provided (Fig. 4c).

3- The authors claim there are higher LAMP2 levels in individuals with T2D than controls, but based on the quantification in Fig. 6a this is unclear. The same observations are extended for Fig. 6b, where the statistical analysis for DNMT3B does not seem to support the claim that there was an increase only in T2D muscle cells upon differentiation or in DNMT3B in myotubes day 7 versus control individuals. Based on these results, the claim that markers of autophagy and epigenetic enzymatic activity are altered in cells from T2D individuals lacks support.

4- The authors claim that reduced VPS39 in the VPS39^{+/-} mice resembles the situation seen in T2D individuals, nevertheless expression in muscle of T2D shown in the manuscript is not significantly different than control individuals (Fig. S6b).

Reviewer #3:

Remarks to the Author:

The authors have addressed the comments and the manuscript can be accepted for publication.

Reviewer #4:

Remarks to the Author:

I thank the authors for addressing my concerns thoroughly, and have no further comments to raise.

Response to Reviewers' comments NCOMMS-19-27809A:

Reviewer #1 (Remarks to the Author):

my comments have been sufficiently addressed

Reviewer #3 (Remarks to the Author):

The authors have addressed the comments and the manuscript can be accepted for publication.

Reviewer #4 (Remarks to the Author):

I thank the authors for addressing my concerns thoroughly, and have no further comments to raise.

RESPONSE: We are pleased that three out of four reviewers are overall positive to the improvements of the revised manuscript, and that all their comments have been sufficiently addressed. We are very happy to see that they believe that the manuscript is now suitable for publication in Nature Communications.

Reviewer #2 (Remarks to the Author):

The revised manuscript by Davegardh and colleagues addresses many of the concerns raised previously. Some aspects that still require attention are detailed below:

1- The authors have conducted extensive experiments to understand the autophagy defect upon the silencing of VPS39.

RESPONSE: We appreciate that the reviewer notices our work, and that our revised manuscript largely addressed his/her concerns. We also want to thank reviewer #2 for his/her final comments that helped us to improve our manuscript further. We have now addressed all the final aspects that required attention, and they are described in the point-to-point response below.

a. It is not clear why a three-way ANOVA was used for statistical analysis in the autophagic flux. Can the authors justify such an analysis? It seems a two-way ANOVA is more suitable (Two factors genotype and drug treatment).

RESPONSE: We thank the reviewer for this comment. For the autophagic flux experiments we used three-way ANOVA with the factors genotype, starvation and Baf-A1 to compare values that differed by one factor, and adjusted for multiple comparisons with FDR ($q < 0.05$). We chose to execute a three-way ANOVA in order to be able to perform all multiple comparisons in one analysis instead of either doing multiple two-way ANOVAs for comparisons between and within genotypes, or having to combine “starvation” and “Baf-A1” into one factor (“treatment”). The aims with the two treatments in the experimental design are opposite, one activates autophagy (starvation) and the other inhibits the process

(Baf-A1). Hence, the three-way ANOVA allows us to distinguish the effects that can be attributed to either of the two different treatments as well as genotype. Based on this comment, we have consulted a professional biostatistician affiliated with Lund University Diabetes Centre and after this consultation we still believe that a three-way ANOVA is the appropriate method for the statistical analysis of these data. We also performed a linear mixed-effects model analysis on the autophagy flux data that confirmed the statistical results from the three-way-ANOVA. In the revised version of the manuscript we have clarified the description of this analysis in the “Statistical analyses” section on page 35:

“Autophagic flux data were analyzed with a repeated measures three-way ANOVA (factors “Genotype”, “Starvation” and “Baf-A1”), followed by tests for each treatment between groups, and between treatments within each group. The three-way ANOVA enabled us to study the overall effects of two different treatments (starvation and Baf-A1) and genotype (siVPS39), as well as perform multiple comparisons between individual groups, in a single analysis.”

b. As mentioned in the manuscript, they have contradictory observations between HCS and WB for LAMP1 and LAMP2 data. They claim it is due to a difference in sensitivity, which is the more sensitive assay? Please provide WB data for LAMP1/2 autoflux.

RESPONSE: We apologize that the statement on the “sensitivity” of the two assays was unclear, and agree that the wording was unfortunate. In order to strengthen our autophagy data we used two separate methods, HCS and WB, to detect the autophagy markers LC3B, p62, LAMP1 and LAMP2. We don’t mean to argue that one of the assays is better than the other but rather that they are *different*. As such they also measure different aspects of the protein dynamics and discrepancies can be expected. HCS is based on immunostaining and detects the markers on a more detailed cellular level where both the number and the area of stained spots *per cell* is counted as well as visualizing the intracellular localization patterns of the markers. On the other hand, WB measures the protein in a bulk lysate and only distinguishes the amount of the epitope that is present in the solubilized lysate. Consequently, we meant that the HCS assay is generally able to pick up *subtle* changes between different experimental groups, compared to WB. We have now changed the wording regarding this in the manuscript on pages 11-12:

“The discrepancies in LAMP1/2 levels between the HCS and western blot assays may depend on that these two separate methods detect different aspects of the protein dynamics, and that HCS is able to detect more subtle changes in LAMP1/2 levels by measuring spot number and spot area per cell compared to western blot.”

Moreover, based on the fact that LAMP1/2 levels detected by WB were not altered in siVPS39 cells vs control cells in the basal state (Figure 3d), despite changes in spot number and spot area detected by HCS (Figure 3a-b), we found it relevant to only follow up LAMP1/2 using HCS in the autophagy flux experiments. However, based on this reviewer comment we now added both LAMP1 and LAMP2 protein levels in the autophagy flux experiments (determined by WB) in Supplementary Figure S2h-i. In summary, LAMP1/2 protein levels detected by WB were not different between the genotypes and not altered in response to any of the treatments. We also mention this in the manuscript on page 12:

“LAMP1/2 protein levels were not different between the genotypes, and not altered in response to any of the treatments (Supplementary Figure 2h-i).”

c. Can the authors please provide a blot image with complete LC3 for Fig. 3d.

RESPONSE: We thank the reviewer for pointing out that the LC3 blot in Figure 3d could be improved. We have now replaced the blot image so that both LC3B-I and LC3B-II bands are clearly visible. Additionally, all uncropped blots underlying the figures are provided in the Source data file accompanying the manuscript. For clarity, we also present the complete blots for LC3B below (Figure 1).

Figure 1. Uncropped blots underlying LC3B-II protein levels in Figure 3d.

LC3B protein was analyzed in VPS39-silenced (VPS39) and control cells (NC) at day 3 of human muscle cell differentiation. Bands at the level marked with red triangle (LC3B-II) were quantified. Bands marked with red X corresponds to unrelated samples that were not used for quantification. Values were normalized based on the total protein using the Stain-Free technology (Bio-Rad). Lane numbers in red and molecular weight bands (ladder) in blue. Samples used as representative blot in Figure 3e in the dashed, orange rectangle.

2- The authors claim there is a decrease in GSK3 in insulin-stimulated VPS239-silenced cells versus control, but this is not supported by the statistical analysis provided (Fig. 4c).

RESPONSE: We thank the reviewer for this comment. Indeed, the decrease in phosphorylated GSK3 α was only nominally significant with a p-value of 0.055 (2-way ANOVA followed by Fisher's LSD test). For GSK3 β , the factor "Genotype" was significant (*p<0.05) for the overall ANOVA and there was no interaction between the two factors "Insulin" and "Genotype" (p=0.95). The overall group means (basal and insulin combined) are 1.665 for NC and 1.276 for siVPS39, and the corresponding overall difference between the means is -0.3883 with a 95% confidence interval of -0.6926 to -0.08387. In summary, the average phosphorylated GSK3 β is overall lower in siVPS39 compared to NC. In the revised version of the manuscript, we have toned down the effect of VPS39-silencing on GSK3 function in the "Results" section on page 13, and removed it from the "Discussion" on pages 22 and 23:

Results p. 13: *"The phosphorylation of downstream Akt substrates TBC1D4 that controls GLUT4 translocation³⁵, and glycogen synthase kinase 3 (GSK3) that regulates glycogen synthase activity³⁶, was significantly and nominally decreased, respectively, in insulin-stimulated VPS39-silenced cells versus control (Figure 4b-c)."*

Discussion p. 22: *"We also found lower insulin-stimulated phosphorylation of key proteins involved in glucose uptake and metabolism i.e. Akt and TBC1D4 in VPS39-silenced human myoblasts, suggesting an altered glucose metabolism in these cells^{55,56}."*

Discussion p. 23: *“In support of this, we found reduced glucose uptake in muscle from $Vps39^{+/-}$ mice, lower insulin-stimulated Akt and TBC1D4 phosphorylation in VPS39-silenced muscle cells as well as a positive correlation between insulin-stimulated glucose uptake and VPS39 expression in human skeletal muscle.”*

3- The authors claim there are higher LAMP2 levels in individuals with T2D than controls, but based on the quantification in Fig. 6a this is unclear. The same observations are extended for Fig. 6b, where the statistical analysis for DNMT3B does not seem to support the claim that there was an increase only in T2D muscle cells upon differentiation or in DNMT3B in myotubes day 7 versus control individuals. Based on these results, the claim that markers of autophagy and epigenetic enzymatic activity are altered in cells from T2D individuals lacks support.

RESPONSE: We apologize that this was unclear. Protein levels of LAMP2 and DNMTs in T2D and controls during differentiation were analyzed with two-way ANOVA with repeated measures in the factor “Differentiation” (Day 0, 3 and 7) followed by multiple comparisons between time points (Day 0, 3 and 7) within each group (NGT or T2D), or between groups for each time point.

The statement that LAMP2 is increased in T2D is based on the result from the overall ANOVA (stated in text above the graphs). For LAMP2 the factor “T2D” is significant ($*p < 0.05$) and there was no interaction between the two factors “T2D” and “Differentiation” ($p = 0.74$). This means that there is an overall difference between the groups (NGT vs T2D) throughout differentiation. When inspecting the data closer the overall group means (Day 0, 3 and 7 combined) are 1.015 for NGT and 1.262 for T2D, and the corresponding overall difference between the means is 0.2475 with a 95% confidence interval of 0.01405-0.4809. In conclusion, the average LAMP2 protein levels are overall higher in T2D compared to NGT. We have explained these results in greater detail on page 16:

*“Based on the overall two-way ANOVA ($*p < 0.05$ for NGT vs T2D), the average LAMP2 levels were higher throughout differentiation in diabetics compared to controls (Figure 6a), which agreed with observations in VPS39-silenced cells (Figure 3a-c) and are generally in line with previous findings⁵¹.”*

For DNMT3B we write in the text that the *alternative* isoform of DNMT3B (SUMOylated) is increased during differentiation in T2D muscle cells, and that this isoform is also higher in myotubes from T2D versus NGT at day 7 of differentiation. We agree that the graph in the previous version of the manuscript did not fully support this statement as only nominally significant values were reported for the increase in SUMOylated DNMT3B during differentiation. When reviewing the DNMT3B protein data (based on this comment) we performed a normality test (Shapiro-Wilk) and found that the data did not follow a Gaussian distribution but rather a lognormal distribution. Therefore, we have now rerun the statistical analysis for DNMT3B (and DNMT1) using log₂-transformed values instead and the graphs have been updated with the corresponding statistical results. Protein levels of SUMOylated DNMT3B are significantly increased during differentiation (Day 3 and 7 vs Day 0) in the T2D muscle cells as well as between T2D and NGT at day 7.

In the revised version of the manuscript we clarified the description of this analysis in the “Statistical analyses” Methods section on page 35 and in the corresponding figure legend.

Methods p. 35: *“DNMT3B, SUMOylated DNMT3B and DNMT1 protein values in NGT/T2D were log₂-transformed before statistical analysis.”*

Figure legend p. 62: *“The effects of T2D and differentiation stated above the graphs were calculated with two-way ANOVA, or mixed-effects model (DNMT3B), with repeated measures in the factor*

“Differentiation” (Day 0, 3 and 7). DNMT3B, SUMOylated DNMT3B and DNMT1 protein values were log2-transformed before statistical analysis. Unlogged values are presented in the graphs.”

4- The authors claim that reduced VPS39 in the VPS39^{+/-} mice resembles the situation seen in T2D individuals, nevertheless expression in muscle of T2D shown in the manuscript is not significantly different than control individuals (Fig. S6b).

RESPONSE: We appreciate this comment. However, we mean that the heterozygous mice (*Vps39*^{+/-}) resemble T2D, because we found similar reduction in VPS39 mRNA expression in muscle from *Vps39*^{+/-} versus WT mice as we found in myoblasts and myotubes from T2D individuals versus controls. We have clarified this and discuss it on page 24 of the revised ms:

*“To further study the role of VPS39 in vivo, we examined a mouse model for VPS39-deficiency. Heterozygous mice were chosen because they have reduced, but not lacking, expression of VPS39, which resembles the situation seen in **the muscle cells** of T2D individuals. *Vps39* expression was reduced by 26% in muscle from *Vps39*^{+/-} mice, while it was reduced by 21.4% and 23.6% in myoblasts and myotubes from T2D individuals, respectively. Indeed, *Vps39*^{+/-} mice showed glucose intolerance and decreased glucose uptake in muscle. These mice also exhibited differential expression of genes encoding proteins that affect autophagy, epigenetic programming, muscle function and oxidative phosphorylation. In agreement, expression of OXPHOS genes is reduced in T2D muscle⁴. Together, we demonstrate for the first time that *Vps39*^{+/-} mice have metabolic phenotypes similar to **a human muscular model of T2D.**”*

Moreover, as mentioned in the Abstract, in the Introduction and in the Results section of our manuscript, our main goal was to explore alterations in expression in cultured myoblasts and myotubes from individuals with T2D versus non-diabetic controls, as we are specifically interested in any dysregulations within the muscle cells during the disease (see the text below). Our main goal was not to study skeletal muscle biopsies.

Abstract p. 4: *“Lower muscle quality and insulin resistance are hallmarks of type 2 diabetes (T2D). We explored whether alterations in muscle stem cells (myoblasts) from T2D individuals contribute to these phenotypes.”*

Introduction p. 6: *“To identify novel candidates that contribute to abnormalities seen in muscle from T2D patients, we analyzed genome-wide expression and DNA methylation in primary human myoblasts and myotubes from T2D individuals and controls.”*

Results p. 7: *“We then tested whether T2D is associated with altered expression of previously unrecognized and known regulators of muscle regeneration in human myoblasts. We performed genome-wide expression analysis of myoblasts obtained from 13 controls and 13 T2D individuals (**Figure 1b**).”*

We identified reduced *VPS39* expression in both myoblasts and myotubes from T2D versus controls. We then performed follow-up experiments by silencing *VPS39* in both human myoblasts *in vitro* and in a mouse model *in vivo*. Indeed, *Vps39*^{+/-} mice had altered glucose uptake in muscle and other alterations that mimic the situation seen in T2D patients as well as in human muscle cells where VPS39 had been silenced.

During the review process we were asked to include *VPS39* expression data from skeletal muscle biopsies from individuals with T2D and controls. Here, we used available microarray data from a previous study (V Mootha et al Nature Genetics 2003) and in line with our data in myoblast and myotubes, *VPS39* expression was also reduced in muscle biopsies from T2D versus control individuals, although only

nominally significant (see Figure S6b). Interestingly, *VPS39* expression in muscle biopsies correlated positively with glucose uptake (M-value, $r=0.64$, $p=0.017$) in diabetics (see Figure S6a). Here, it is important to consider that the muscle biopsies were taken exclusively from men who were ≈ 66 years old, while myoblasts and myotubes were from both women and men who were ≈ 56 years old. Also, the myoblast and myotubes are more “pure” cell fractions, while the muscle biopsies include other cells types such as endothelial cells, smooth muscle cells, immune cells, nerve cells and fibroblasts which may influence the overall mean *VPS39* expression. Importantly, the Human Protein Atlas (<https://www.proteinatlas.org/ENSG00000166887-VPS39>) states low tissue specificity and demonstrate expression in several different cell types, underscoring the importance of studying loss of function in well-defined cell systems and providing a possible explanation for the expression in human muscle biopsies. Based on this comment, we now discuss this on page 25 of the revised manuscript:

“We found nominally reduced VPS39 expression in muscle biopsies from T2D versus control individuals. This result could possibly be explained by that these biopsies were taken exclusively from elderly men (≈ 66 years) and that muscle biopsies contain several other cells types such as endothelial cells, smooth muscle cells, immune cells, nerve cells and fibroblasts apart from myoblasts and myotubes.”

Reviewers' Comments:

Reviewer #2:

Remarks to the Author:

The authors have satisfactorily addressed my additional concerns.